# Feeding strategy as a key driver of the bioaccumulation of MeHg in megabenthos

David J. Amptmeijer<sup>1</sup>, Andrea Padilla<sup>2</sup>, Sofia Modesti<sup>2</sup>, Corinna Schrum<sup>1, 2</sup>, and Johannes Bieser<sup>1</sup>

Correspondence: David J. Amptmeijer (davidamptmeijer@gmail.com)

Abstract. The bioaccumulation of methylmercury (MeHg) in the marine food chain poses a neurotoxic risk to human health, especially through the consumption of seafood. Although MeHg bioaccumulation at higher trophic levels is relatively well understood, MeHg bioaccumulation at the base of the food web remains underexplored. Given the neurotoxic effects of MeHg on human health, it is essential to understand the drivers of bioaccumulation at every level of the food chain. We coupled six megabenthos functional groups in the ECOSMO end-to-end ecosystem model to the MERCY v2.0 Hg cycling model. We investigated how various feeding strategies influence the bioaccumulation of both inorganic Hg (iHg) and MeHg in marine ecosystems. We show that feeding strategy significantly influences bioaccumulation and correlates more strongly with iHg than does trophic level. In particular, suspension feeders have elevated iHg levels, while filter feeders have elevated MeHg levels compared to other megabenthos. Additionally, we show that feeding strategies alone allow us to accurately model the bioaccumulation of both iHg and MeHg in low-trophic-level megabenthos. However, when modeling higher trophic levels, incorporating the allometric scaling law substantially improves model performance. These results demonstrate the need for a holistic approach in which iHg, MeHg, and the trophic level of organisms are evaluated at both high and low trophic levels to identify what food web structures drive high MeHg concentrations in seafood.

## 1 Introduction

Mercury (Hg) is a naturally occurring element. In addition to its natural occurrence, it is also emitted through various anthropogenic activities, such as the burning of fossil fuels, small-scale artisanal gold mining, and the production of cement and ferrous metals (Pacyna et al., 2006). These anthropogenic emissions have significantly raised environmental Hg levels, with 78%, 85%, and 50% of atmospheric, upper ocean, and deep ocean Hg, respectively, originating from anthropogenic emissions (Geyman et al., 2025).

When elemental Hg (Hg0) is emitted, it can undergo long-range atmospheric transport, allowing global dispersion and subsequent deposition in the oceans, thus increasing Hg levels in the marine environment (Durnford et al., 2010). Marine Hg<sup>0</sup> is volatile and can return to the atmosphere or be oxidized into dissolved Hg (Hg<sup>2+</sup>) (Sommar et al., 2020). This Hg<sup>2+</sup> can be reduced back to volatile elemental Hg<sup>0</sup>, or it can be methylated to the dangerous neurotoxin methylmercury (MeHg), which can be present as monomethylmercury (MMHg<sup>+</sup>) and dimethylmercury (DMHg) (Jensen and Jernelov, 1969; Lin et al., 2021).

<sup>&</sup>lt;sup>1</sup>Matter Transport and Ecosystem Dynamics, Helmholtz-Zentrum Hereon, Geesthacht, Germany

<sup>&</sup>lt;sup>2</sup>Universität Hamburg, Institute for Marine Sciences, Mittelweg 177, 20146 Hamburg, Germany

In this paper, we examine the bioaccumulation of three groups of Hg; total Hg (tHg) refers to all Hg, methylmercury (MeHg) refers to both MMHg<sup>+</sup> and DMHg, and inorganic Hg (iHg) refers to all Hg that is not MeHg.

There are two key processes involved in bioaccumulation: bioconcentration and biomagnification. When animals absorb Hg directly from their environment, this is called bioconcentration. Both iHg and MeHg bioconcentrate. Since iHg is generally present in higher concentrations than MeHg, and its bioconcentration rate is higher, iHg is usually bioconcentrated to higher levels than MeHg (Mason et al., 1996). The bioconcentration process can result in high concentrations in aquatic organisms. This process is commonly quantified using the Volume Concentration Factor (VCF), a unitless ratio between the Hg concentration in phytoplankton and that in the surrounding water:

$$VCF = \frac{C_{\text{phytoplankton}}}{C_{\text{water}}} \tag{1}$$

where both  $C_{\text{phytoplankton}}$  and  $C_{\text{water}}$  have the same units, for example, ng Hg  $\mu\text{m}^{-3}$ , and the VCF is unitless. For MeHg, very high volume concentration factors of up to  $6.4 \times 10^6$  have been reported in the literature (Lee and Fisher, 2016; Schartup et al., 2018).

MeHg concentrations that are elevated due to bioconcentration can be further increased by biomagnification along the aquatic food web. Biomagnification refers to the increase in Hg with each successive trophic level in the food chain. The trophic transfer efficiency of MeHg (66-80%) is higher than that of iHg (7-46%); this is a key reason why MeHg accumulates at much higher levels in the food chain, especially in high-trophic-level animals (Metian et al., 2020; Wang and Wong, 2003; Dutton and Fisher, 2012). MeHg is a neurotoxin whose overconsumption can decrease IQ points and raise the risk of heart attacks, and consumption of MeHg-contaminated seafood is the primary pathway of Hg exposure in humans, with elevated risk among coastal and seafood-reliant populations (Sheehan et al., 2014; Zhang et al., 2021; Genchi et al., 2017; Trasande et al., 2006).

The risk associated with consuming seafood contaminated with MeHg gained significant attention after over 1000 fatalities occurred in Japan in 1956 due to the consumption of contaminated seafood from Minamata Bay (Harada, 1995). Although this MeHg outbreak was a unique event linked to industrial waste disposal containing Hg, it highlighted the dangers of MeHg exposure. In order to reduce the risk of further outbreaks of MeHg intoxications, the Minamata Convention on Mercury was founded. A total of 151 countries have pledged to reduce their Hg emissions in support of the Minamata Convention and 128 countries have signed and ratified the convention (UNEP, 2013). The global state of Hg as a pollutant and the effect of the Minamata Convention is periodically reviewed in the Minamata Convention Effectiveness Evaluation (Outridge et al., 2018).

45

While there is considerable understanding of MeHg bioaccumulation in high-trophic-level animals, less is known about the bioaccumulation drivers at the base of the food web where Hg concentrations tend to be lower, resulting in reduced risk to humans. As such, these organisms are not prioritized in the current monitoring strategies under the ongoing effectiveness evaluation of the Minamata Convention, which focuses primarily on fish, humans, and predatory wildlife (Evers et al., 2016). Additionally, the Minamata Convention effectiveness evaluation has shown that Hg and MeHg concentrations in water and sediment do not correlate well with levels in biota, leading to greater emphasis on biological monitoring over abiotic compartments.

Once Hg is bioconcentrated in primary producers, a strong link appears between the trophic level and Hg bioaccumulation (Madgett et al., 2021). This indicates that our understanding of Hg bioaccumulation in high trophic levels is greatly limited by our understanding of Hg bioaccumulation at the base of the food web.

The benthic food web is highly complex, making it challenging to improve our understanding of bioaccumulation within it (Silberberger et al., 2018). There are several distinct groups of megabenthos with different feeding strategies, such as bivalves that filter feed, lugworms that feed on sediment carbon particles, active hunters and scavengers such as shrimps and crabs, and sponges that feed on suspended dissolved material. These different feeding strategies allow them to exploit a variety of food sources, but different food sources can have different Hg concentrations, and Hg originating from different food sources can have different assimilation efficiencies. In this study, we hypothesize that the low-trophic-level biota feeding strategy has a significant impact on their Hg content.

We focus this study on the benthic food web. Although primary production in the North Sea can be highly variable due to factors such as wind (Daewel and Schrum, 2017), tidal mixing (Zhao et al., 2019) and nutrient availability (Richardson et al., 1998), primary production in coastal areas is generally dominated by pelagic phytoplankton, with the exception of extremely shallow areas that are dominated by benthic macroalgae (Krause-Jensen et al., 2012; Cibic et al., 2022). In well-mixed areas where pelagic phytoplankton dominate primary production, they can be consumed by megabenthos and there is a strong coupling between the benthic and the pelagic, called the bentho-pelagic coupling. In these well-mixed areas, megabenthos can reach high biomass since food is abundant in several ways, resulting in megabenthos with different feeding strategies in the same ecosystem (Ghodrati Shojaei et al., 2016).

We hypothesize that differences in feeding strategies among low-trophic-level megabenthos play an important role in the observed disconnect between Hg concentrations in the water and sediment and those at the base of the food web. We investigated whether the feeding strategy impacts bioaccumulation and hypothesized that feeding strategies influence the bioaccumulation of iHg and MeHg differently, contributing to the high variation in Hg levels at the base of the benthic food web.

80

To test our hypotheses, we employed three approaches. First, we conducted a literature review in which we collected field observations of tHg, MeHg, and iHg concentrations, together with trophic level and megabenthos feeding strategy. We then performed statistical analyses on these data to examine if we could find a relationship between feeding strategy and trophic level. Second, we carried out a modeling experiment in which megabenthos with different feeding strategies competed under physical drivers in idealized scenarios representative of megabenthos-rich coastal oceans. The megabenthos groups were designed to differ only in their feeding strategies, allowing us to isolate this effect. This experiment tested whether observed effects from our literature review could be reproduced in a fully coupled model.

Finally, we analyzed data from a single study to evaluate whether the same dynamics observed in the model and the global dataset were also present in a single geographical location. While none of these individual tests is conclusive on its own, consistent evidence across all three approaches would support the conclusion that feeding strategy is an important driver of Hg bioaccumulation and would warrant further empirical studies to investigate this role in more detail.

## 2 Materials and methods

#### 2.1 The models

To assess the importance of the feeding strategy, we modeled bioaccumulation in megabenthos, with the feeding strategy being the only distinction between different groups of megabenthos. Then we compared our model to observations to evaluate whether this approach allows us to accurately model bioaccumulation or if additional drivers should be taken into account. We used a fully coupled 1D water column model run in two setups that resemble typical hydrological regimes found in coastal oceans. We coupled the Generalized Ocean Turbulence Model (GOTM) (Burchard et al., 1999) with the ECOSMO E2E ecosystem model (Daewel et al., 2019) and the MERCY v2.0 Hg speciation and bioaccumulation model (Bieser et al., 2023).

## 100 2.1.1 The hydrodynamical model

The hydrodynamics of the model are simulated using the GOTM, which is a 1D hydrodynamic model (Bolding et al., 2021). GOTM calculates the turbulence of a vertical 1D water column set-up by computing the solutions to the one-dimensional version of the transport equation of momentum, salinity, and temperature. The model is nudged to observational data sets for temperature and salinity. The setups are based on gridded bathymetry data with 1/240° resolution (GEBCO Bathymetric Compilation Group, 2020), ECMWF ERA5 dataset for meteorological data (Wouters et al., 2021), Ocean Atlas for salinity and temperature profiles (Garcia H.E. et al., 2019), and the TPOX-9 atlas for tides (Egbert and Erofeeva, 2002), which is combined using the iGOTM tool (https://igotm.bolding-bruggeman.com). The GOTM model is coupled via the Framework for Aquatic Biogeochemical Modeling (FABM) (Bruggeman and Bolding, 2014). The biogeochemical models are encoded in FABM. The FABM interfaces communicate the state variables between the GOTM model and the biogeochemical models.

## 110 2.1.2 The physical setups

115

120

The model runs in two setups, the first is a 41.5 m deep permanently mixed Southern North Sea setup and the second is a seasonally mixed 110 m Northern North Sea setup. These setups are described in more detail in Amptmeijer et al. (2025). The Southern North Sea setup is located at  $(54^{\circ}15'00.0"N\ 3^{\circ}34'12.0"E)$ . It is a shallow station that is permanently mixed, meaning that megabenthos can feed directly from the phytoplankton and zooplankton bloom. The setup is chosen because it resembles perfect growth conditions for megabenthos, and most megabenthos in the observations are sampled from similar circumstances. Because of this, most samples are from shallow well-mixed coastal areas, and we used this setup to evaluate the performance of the models.

The Northern North Sea setup is located at  $(57^{\circ}42'00.0"N\ 2^{\circ}42'00.0"E)$  and is only mixed in winter. This means that megabenthos cannot feed directly from the bloom, but are rather dependent on the sinking of detritus particles. In nature, these deeper areas typically have lower overall biomass. This setup is used to evaluate whether the models predict a difference in the bioaccumulation of iHg and MeHg under a different hydrodynamic regime.

### 2.1.3 The MERCY v2.0 model

Hg cycling and speciation is modeled using the MERCY v2.0 model (Bieser et al., 2023). The MERCY v2.0 model is a comprehensive Hg cycling model that includes speciation between seven forms of Hg and partitioning to both Dissolved Organic Matter (DOM) and detritus. It was originally developed as a 3D Hg cycling model of the North and Baltic Seas. However, in this study, we use the 1D version of this model, which is driven using the GOTM model. This configuration is used, described, and evaluated in more detail in (Amptmeijer et al., 2025).

## **2.1.4 ECOSMO E2E**

125

130

135

150

The ecosystem model is based on the ECOSMO E2E (ECOSystem Model End-to-End) ecosystem model (Daewel et al., 2019). This model extends the ECOSMO II model to include higher trophic levels while preserving consistency at lower trophic levels (Daewel et al., 2019). The version used in this study is the same as the version used and evaluated in (Amptmeijer et al., 2025). In this version, small modifications have been made, such as lowering the mortality rate of zooplankton and decreasing the efficiency of carbon uptake to make the model more suitable for bioaccumulation compared to the version published by (Daewel et al., 2019). We implemented bioaccumulation to account for bioconcentration in all trophic levels and biomagnification in all consumers. Phytoplankton have a size-dependent Hg uptake and release rate, based on observations by Pickhardt et al. (2006), which found higher MeHg in smaller phytoplankton but consistent iHg levels. As such, different phytoplankton groups are implemented with a size-dependent uptake and release rate for iHg but only size-dependent uptake rates with constant release rates for MeHg. This results in diatoms and flagellates bioaccumulating similar amounts of iHg, while smaller flagellates accumulate more MeHg compared to larger diatoms. The uptake and release rates of iHg and MeHg in zooplankton are based on Tsui and Wang (2004) and on Wang and Wong (2003) for fish. An essential component of the ecosystem that interacts with bioaccumulation in megabenthos that was not overhauled for this study is the interactions between detritus and DOM and iHg and MeHg. The only Hg species assumed to partition to DOM and detritus are Hg<sup>2+</sup> and MMHg<sup>+</sup>, and this partitioning is assumed to be an equilibrium that is instantaneous and is reestimated on every time step. The equilibrium is based on the particle partitioning coefficient for organic matter K<sub>d</sub> values which are based on Allison et al. (2005) and Tesán Onrubia et al. (2020). This value is  $\log_{10}(6.4)$  and  $\log_{10}(6.6)$  for the partitioning of  $\mathrm{Hg}^{2+}$  and  $\log_{10}(5.9)$  and  $\log_{10}(6.0)$  for the binding of MMHg<sup>+</sup> to detritus and DOM respectively. This is the same approach that is used and evaluated in Bieser et al. (2023).

## 2.2 Model development

To use the model to study bioaccumulation in megabenthos, the higher trophic level of the ECOSMO E2E model is altered. We replaced the macrobenthos, fish 1, and fish 2 functional groups with six megabenthos functional groups, as shown in Fig. 1. The megabenthos groups are separated by their feeding strategy: filter feeder, deposit feeder, generalist feeder, suspension feeder, predator, and top predator.

**Filter feeders** filter suspended particles from the water column. In our model, they can eat phytoplankton, zooplankton, and detritus. Examples of filter feeders are mussels, tubeworms, and barnacles. The second group is **deposit feeders**. These animals

Figure 1. The overview of the modeled megabenthos functional groups and how they interact with each other and functional groups in the ECOSMO E2E model. There are six megabenthic functional groups. The filter feeder feeds on pelagic detritus, zooplankton, and phytoplankton. The suspension feeders feed on pelagic detritus, phytoplankton, zooplankton, and DOM. The generalist feeds on phytoplankton, zooplankton, pelagic detritus, and sediment organic carbon. The deposit feeder feeds on sediment organic carbon. The benthic predator feeds on the other four megabenthos functional groups and the top predator solely feeds on the benthic predator. The arrows indicate trophic interactions where the arrow goes from the prey to the predator and the arrows have the same colour as the prey. The black lines represent loss of organic material due to mortality. When megabenthos die, their organic carbon is transferred to pelagic DOM and detritus, as well as the sediment, shown by the solid black arrow. In contrast, when pelagic organisms die, their organic carbon is transferred to DOM and detritus, indicated by the dotted black arrow. Several sub-images have been used in this image. Sources of the images: Filter feeder: *Sabella spallanzanii* (photo by Diego Delso, CC BY-SA 4.0, via Wikipedia), Suspension feeder: *Aplysina fistularis* (photo by Twilight Zone Expedition Team 2007, NOAA-OE, CC BY 2.0, via Flickr), Generalist feeder: *Crangon crangon* (photo by Etrusko25, Public Domain, via Wikipedia), Deposit feeder: *Buccinum undatum* (photo by Oscar Bos / Ecomare, CC BY 4.0, via Wikipedia), Benthic predator: *Hommarus gammarus* (photo by Bart Braun, Public Domain, via Wikipedia), Top predator: *Sepia officinalis* (photo by Nick Hobgood, CC BY-SA 3.0, via Wikipedia).

consume organic carbon from the sediment; in our model, they exclusively feed on organic carbon deposited in the sediment. This group includes gastropods and polychaete worms, such as the lugworm (*Arenicula marina*). The **generalist feeder** includes animals such as brown shrimp (*Crangon crangon*), which can utilize various feeding strategies. In our model, this group feeds on phytoplankton, zooplankton, detritus, and deposited material. We also include a **suspension feeder**. Suspension feeders, such as sponges, can consume detritus and DOM. The consumption of DOM, which is too small to be consumed by filter feeders, differentiates suspension and filter feeders. A common mechanism for consuming DOM involves symbiotic bacteria, as seen in chemosymbiotic bivalves from the families Lucinidae, Solemyidae, and Thyasiridae, and microbial biomes of high microbial assemblage sponges (Dufour, 2018; Olinger et al., 2021). Finally, we included two predators. The first predator is referred to as **the predator**. The predator feeds on the four benthic groups mentioned above and has an equal preference and grazing rate in all groups, but it will prioritize abundant groups. This preference is implemented by making the food available for predation by the predators not linearly related to the abundance of the prey, but calculated as:

165 
$$b_{\text{available}} = \begin{cases} b_{\text{biomass}}, & \text{if } b_{\text{biomass}} \ge b_{\text{protected}}, \\ b_{\text{biomass}} \frac{b_{\text{biomass}}}{b_{\text{protected}}}, & \text{if } b_{\text{biomass}} < b_{\text{protected}}. \end{cases}$$

in which,

155

- $b_{\text{available}}$ : Portion of prey biomass in g C m<sup>-2</sup> accessible to predators.
- $b_{\text{protected}}$ : Level of prey biomass in g C m<sup>-2</sup> below which hunting becomes less optimal or energetically inefficient.
- $b_{\text{biomass}}$ : Total prey biomass in g C m<sup>-2</sup> in the environment.

The megabenthos in the North Sea are estimated to have between 1.1 and 35.5 gC m<sup>-2</sup> (Heip et al., 1992; Daan and Mulder, 2001). The value for B<sub>Protected</sub> is chosen as 1 gC m<sup>-2</sup> for all megabenthos except for the benthic predator where B<sub>Protected</sub> is 0.5 gC m<sup>-2</sup>. These values are chosen to protect megabenthos functional groups from extinction due to predation when their values are below the expected range. This relationship models two real-world interactions. First, when the concentration of prey is low, the small number of individuals can more likely survive under ideal circumstances and, therefore, may be less exposed to predation (Campanella et al., 2019). Secondly, several predators, such as the shore crab, adapt their behaviors to the density of the prey and learn to be more efficient in the hunting of more common prey (Chakravarti and Cotton, 2014). Our model is resolved in carbon content, while measurements are often in dry weight. The carbon fraction of dry weight generally ranges from 0.4 to 0.6, but can vary between different taxa (Gorokhova and Hansson, 2000; Jurkiewicz-Karnkowska, 2005). To ensure consistency across different functional groups with diverse feeding strategies, we maintain a 1:2 conversion ratio for carbon to dry weight for all megabenthos functional groups.

### 2.2.1 Assimilation efficiency of iHg and MeHg

The assimilation efficiency (AE) of iHg and MeHg is a key parameter in correct biomagnification modeling. AE is based on laboratory experiments that analyze AE in phytoplankton (Metian et al., 2020; Wang and Wong, 2003). An assimilation

efficiency of 0.95 for MeHg and 0.31 for iHg is chosen for everything except deposit feeding, which has a lower feeding efficiency of 0.07 for iHg and 0.43 for MeHg according to Dutton and Fisher (2012).

### 2.2.2 Semi-labile DOM




In the ECOSMO E2E model, only labile-DOM is resolved. This means that there is very little DOM. In our model, we want to incorporate a suspension feeder that would utilize DOM as a food source. Because of this, we added a DOM component referred to as semi-labile DOM. This semi-labile DOM has the same bacterial degradation rate as that of the detritus, and it has the same Hg partitioning behavior as labile DOM. When organic carbon (detritus + labile-DOM + semi-labile-DOM) is formed, 5% is formed as semi-labile DOM, and there is a breakdown of the detritus into semi-labile DOM of 0.001 d<sup>-1</sup> (per day). Since the categorization of DOM is very complex, these rates are estimated to create a low maximum of 50 mg C m<sup>-3</sup>. This is lower than the DOM concentrations typically found in the North Sea, but because it is unclear which fraction of DOM can be consumed by suspension feeders, this amount provides suspension feeders a unique food source that they can utilize while not outcompeting other megabenthos (Lønborg et al., 2024).

## 2.2.3 Allometric scaling model

Finally, we ran the model incorporating additional drivers of MeHg bioaccumulation to evaluate whether they improved model performance. There are three interactions that we take into account for this second model. First, the allometric scaling law, which states that larger animals have a lower base metabolic rate when normalized to body weight (da Silva et al., 2006). Secondly, we account for the observations that MeHg bioaccumulation in fish increases as the water temperature increases, indicating that increased activity does not increase MeHg excretion while it increases MeHg uptake due to a higher grazing rate (Dijkstra et al., 2013). Finally, we assume that predators need to spend more energy on active metabolism to hunt their prey. Because of this, for predators and top predators we assumed a MeHg excretion rate (0.002 d<sup>-1</sup>), which is equal to the respiration rate used for fish in Amptmeijer et al. (2025), while keeping the overall carbon cycle identical between the two models. This leads to a higher bioaccumulation of MeHg at higher trophic levels. The bioaccumulation of iHg is not altered between the two models. In the evaluation, the second model is referred to as the allometric scaling (AS) model.

# 2.3 Literature research and statistics

## 2.3.1 Literature research

To compare the findings with the literature, we collected field studies measuring Hg in megabenthos. The studies we used are shown in Table S1. We categorized the megabenthos into the same feeding categories, "deposit feeder", "filter feeder", "suspension feeder", "grazer", and "predator". To better assess the effect of the trophic level, we also added "primary producers" as the base of the food web, and "seabird" and "benthic fish" as top predators. We analyzed whether trophic level and feeding strategy influence megabenthos iHg, MeHg, and/or tHg content. The total and partial R<sup>2</sup> of the linear regression of the trophic level and the feeding strategy were compared to analyze the effect of both drivers on bioaccumulated iHg, MeHg, and tHg.

We compared our model to observations in two ways. First, we compared it to all the data available in our global dataset. We acknowledge the limitation of this approach, as different geographical regions may have different Hg baselines, but it can provide insight into whether certain feeding strategies are consistently higher or lower in iHg, MeHg, or tHg. The most comprehensive dataset of MeHg bioaccumulation that we could find was published by McClelland et al. (2024); we used this single dataset to verify if patterns observed in the model and the global dataset are also present in a single dataset. If certain patterns are present in our model, in globally aggregated data, and in a single large dataset, it becomes a compelling argument to form a hypothesis for further targeted empirical studies.

## 2.3.2 Model evaluation using a global dataset





The goal of the model is to evaluate how well we can represent the bioaccumulation of iHg and MeHg while only taking into account the feeding strategy and trophic interactions. To this extent, the model's performance is its result. If the model performs well, we can conclude that only accounting for feeding strategies and trophic interactions explains a large amount of the variability in Hg bioaccumulation. Initially, we performed this comparison between observations and the modeled Southern North Sea setup. This was done because most samples are collected from shallow areas with high megabenthic biomass, which the well-mixed Southern North Sea setup better resembles. Afterwards, the models were compared to the Northern North Sea models and the AS model to evaluate the effect of hydrodynamics and increased bioaccumulation in higher trophic level animals on our conclusions. The grazer feeding strategy was omitted, as the ECOSMO E2E model does not include benthic algae to graze on. The modeled generalist was compared to the sum of the deposit and filter feeders from the observations, and the modeled top predator to the benthic fish and seabird feeding strategies.

Model performance was evaluated using normalized bias, RMSE, NRMSE, and the  $R^2$  (Pearson and residual) (see Table S2 for details). Normalized bias and NRMSE values below 0.5 indicate low bias and a good fit.  $R^2_{\rm Pearson}$  quantifies how well differences between feeding strategies are captured, while  $R^2_{\rm Residual}$  reflects agreement with absolute observed values.

#### 2.3.3 Evaluation of the model using a single dataset

We used MeHg bioaccumulation and trophic level data from 476 individuals across 53 taxa of benthic invertebrates as published by McClelland et al. (2024) to verify if the interactions that occur in both our model and the global dataset are consistent when data from one geographical location are studied. These data were selected as they are the largest study we could find with both trophic level and MeHg concentrations. Unfortunately, this study did not sample iHg or tHg, so this component of the model can not be evaluated using this study. When several animals of the same group were sampled, the dataset presents mean values per group per location, which we use as one datapoint in our analyses. Although feeding strategies in the dataset were broadly aligned with our classifications, we reassigned them to match the functional groups in our model. For example, shrimps were categorized as generalist feeders, which group is not present in McClelland et al. (2024), and isopods, which can be small benthic predators, were labeled as deposit feeders because their prey type is not represented in our model.

The data is sampled from two locations in the Canadian Arctic, Cape Bathurst (CB), which has a depth of 22 m and is located at 70°41′42.79″ N, 128°50′21.34″ W, and the eastern coast of Herschel Island in the Mackenzie Trough (MT), which

has a depth of 116 m and is located at 69°36′44.96″ N, 138°33′45.25″ W. Note that this dataset is selected as it is extensive, but the region does have notable differences to the North Sea, where our model is run. It has extensive ice cover in winter, it is colder, and is geographically distant from the model location. It does, however, provide us with an opportunity to test if our model conclusions can be verified using field observations from a single study.

To isolate the effect of the feeding strategy on MeHg bioaccumulation, we first transformed MeHg concentrations to their natural logarithm and fit a linear model with trophic level as predictor using the base R 1m() function. The significance of the deviation from the predicted MeHg concentration at the trophic level was assessed using a one-sample t test. To improve interpretability, we calculated the percentage differences using Percentage difference =  $100 \times \left(\frac{\text{MeHg}_{\text{obs}}}{\text{MeHg}_{\text{pred}}} - 1\right)$  based on the residuals of the linear fit. This is visualized on a bar graph showing the percentage difference in MeHg concentration caused by the feeding strategy. The error bars represent one Standard Error (SE). The same analysis was then performed to estimate differences in MeHg bioaccumulation related to phylum.

As a final test, linear models were fitted on the natural logarithm of bioaccumulated MeHg concentrations using trophic level, phylum, and feeding strategy as predictor variables (using the lm() function in R). Estimated marginal means (EMM) for each feeding strategy were calculated with the emmeans () function of the emmeans package and compared against the overall mean to assess deviations. This analysis was also performed separately for the MT and CB locations to verify the consistency of the effects of the feeding strategy. The EMMs were transformed to a percentage difference with the earlier used equation and the estimated percentage difference due to feeding strategy and its significance are shown.

### 265 3 Results





## 3.1 Model evaluation

### 3.1.1 Evaluation of the Hg cycling and pelagic bioaccumulation

The marine cycling and speciation of Hg, in addition to its bioaccumulation in phytoplankton and zooplankton, are essential drivers of the bioaccumulation of iHg and MeHg in the benthic food web. Observed and modeled dissolved tHg concentrations, the percentage of tHg that is MeHg, and the Hg content of phytoplankton and zooplankton are shown in Table 1. The concentration of dissolved tHg and the percentage of MeHg of dissolved tHg are compared to observations by Coquery and Cossa (1995), while the bioaccumulation of tHg in phytoplankton and zooplankton is compared to observations by Nfon et al. (2009). Note that the observations by Nfon et al. (2009) are not from the North Sea itself but from the better-studied nearby Baltic Sea. The average dissolved tHg concentration is 1.7 and 2.1 pM in the Northern and Southern North Sea, respectively. This is well within one standard deviation of the 1.7±0.7 pM observed by Coquery and Cossa (1995). The MeHg concentration was observed to be between 0.5 and 4.3% of tHg, with an average of 3% in the North Sea. The percentage MeHg in our model is 2.3% and 2.0% on average, which falls well within that range.

For bioaccumulation, we could not find separate reliable measurements of MeHg and iHg in phytoplankton and zooplankton in the North Sea, and therefore evaluated the tHg content instead. The mean bioaccumulation in our model is lower, with 5.8

ng Hg mg<sup>-1</sup> and 9.0 ng Hg mg<sup>-1</sup> in the Northern and Southern North Sea, respectively, but still within one standard deviation of the measurements. Observations labeled as zooplankton and mysis were compared to our modeled microzooplankton and mesozooplankton, respectively. All modeled values fall within one standard deviation of the observed tHg concentration, with one exception: mesozooplankton in the Northern North Sea, which is 13.5% more than one standard deviation above the observations. This is mostly driven by a high iHg content, as the MeHg content is similar in microzooplankton and mesozooplankton.

This similarity in MeHg content between microzooplankton and mesozooplankton in our model arises because, even though mesozooplankton occupy a higher trophic level, they preferentially feed on larger diatoms. These diatoms have a lower MeHg bioconcentration rate than the smaller flagellates preferred by microzooplankton. The high iHg content, especially in the Northern North Sea, is caused by the consumption of detritus by zooplankton in the model. While there is a shortage of data on bioaccumulation at the base of the food web, especially in the North Sea, which complicates model evaluation, the dissolved tHg concentration, the percentage of MeHg, and the tHg content of phytoplankton and zooplankton agree well with observations. With the exception of the 13.5% elevated tHg content in Northern North Sea mesozooplankton, all modeled values fall within one standard deviation of the observations. Because of this, we conclude that the model replicates marine Hg cycling and bioaccumulation at the base of the food web in line with observations, with the caveat that we do not have measurements of zooplankton in the Northern North Sea to verify or reject the elevated levels in that setup.

**Table 1.** Dissolved tHg (pM), MeHg (% of tHg), and tHg concentrations in biota (ng Hg mg<sup>-1</sup> d.w.) in observations and the modeled setups. The observations of aquatic tHg and % MeHg are from Coquery and Cossa (1995) and the bioaccumulation in biota is compared to observations by Nfon et al. (2009).

|                                                  | Observed        | NNS             | SNS             |
|--------------------------------------------------|-----------------|-----------------|-----------------|
| tHg <sub>dissolved</sub> (pM)                    | $1.7\pm0.7$     | $1.7 \pm 0.26$  | $2.0\pm0.28$    |
| MeHg (% of tHg)                                  | 3 (0.5–4.3)     | $2.3 \pm 0.23$  | $2.0 \pm 0.31$  |
| Diatoms tHg (ng Hg mg <sup>-1</sup> )            | 10 ± 5          | $7.0 \pm 1.1$   | $8.3 \pm 1.6$   |
| Flagellates tHg (ng Hg mg <sup>-1</sup> )        |                 | $13.9 \pm 3.0$  | $14.3 \pm 3.0$  |
| Microzooplankton tHg (ng Hg mg <sup>-1</sup> )   | $37.5 \pm 31.3$ | $67.4 \pm 29.3$ | $40.3 \pm 11.4$ |
| Microzooplankton MeHg (ng Hg mg <sup>-1</sup> )  |                 | $7.1\pm 2.1$    | $10.5 \pm 2.7$  |
| Mesozooplankton tHg (ng Hg mg <sup>-1</sup> )    | $62.5 \pm 12.5$ | $86.7 \pm 15.1$ | $72.3 \pm 19.6$ |
| Mesozooplankton MeHg (ng Hg $\mathrm{mg}^{-1}$ ) |                 | $6.9 \pm 2.6$   | $10.5 \pm 1.7$  |

### 3.1.2 Megabenthic biomass


While our megabenthos groups only differ in their feeding strategies and lack direct real-world counterparts, it is important to ensure that all functional groups have consistent biomass in the model. This guarantees that the results originate from the modeled interactions rather than being influenced by unrealistically high or low biomass. We show the yearly progression of

the megabenthos biomass in Fig. 2. Filter feeders have the highest biomass, which is up to 10 g C m<sup>-2</sup> followed by deposit feeders with up to 5 g C m<sup>-2</sup>, generalist feeders with up to 3 g C m<sup>-2</sup>, and suspension feeders with up to 1 g C m<sup>-2</sup>. Higher trophic levels have lower biomass, with up to 0.2 g C m<sup>-2</sup> for the predator and 0.5 g C m<sup>-2</sup> for the top predator. This shows that after a simulation period of 20 years, all megabenthos have a stable population, while biomass is highest at the base of the food web.

**Figure 2.** Megabenthos biomass in the modeled Southern North Sea, dominated by filter feeders, followed by deposit feeders, generalist feeders, suspension feeders, predators, and top predators. Biomass fluctuates between 10 and 15 gC m<sup>-2</sup> and all functional groups have stable populations

#### 3.2 Bioaccumulation in the model


Figure 3 shows the modeled bioaccumulation in the AS model in the Southern North Sea; note that the values are expressed in ng Hg mg  $C^{-1}$ , as this is the best proxy in our model to show the dietary uptake of Hg per unit of energy and nutrients consumed. There is a high concentration of iHg in the sediment, detritus, and DOM. These values are 0.60, 1.1, and 2.6 ng Hg mg  $C^{-1}$  for iHg and 0.089, 0.0067, and 0.012 ng Hg mg  $C^{-1}$  for MeHg respectively. The high amount of iHg in organic carbon is in line with observations that found values of up to 0.114-1.192 ng Hg mg d.w. in sediments from the Scheldt estuary and that DOM strongly binds up to 1.0 ng Hg mg<sup>-1</sup> (Zaferani and Biester, 2021; Haitzer et al., 2002; Muhaya et al., 1997), which would approximate our modeled 2.6 ng Hg mg  $C^{-1}$  if we assume a carbon-to-weight ratio of 1:2. These high iHg values in DOM lead to high values in suspension feeders in both setups. The bioaccumulation of MeHg is different from that of iHg and has the highest bioaccumulation in the top predators and predators, followed by deposit feeders and suspension feeders. In Fig. 4a, c, and e, we show the relationship between the trophic level and the bioaccumulation of iHg, MeHg, and tHg in megabenthos. There is an increase in the MeHg content with trophic level, which is not present for iHg. There is a weak anti-correlation  $(R^2 = 0.20)$  between the bioaccumulation of iHg and the trophic level, which is mainly caused by the high iHg content of the

**Figure 3.** Modeled bioconcentration and biomagnification of iHg and MeHg. Partitioning to detritus and DOM is colored as bioconcentration. The y-axis is cut to show the high and low values. Notable is the high iHg to mgC ratio associated with detritus and DOM, leading to elevated iHg in suspension feeders. Additionally, higher trophic level animals have higher biomagnified MeHg.

low-trophic-level suspension feeders. There is no positive relationship between the bioaccumulation of tHg and the trophic level ( $R^2 = 0.02$ ), while there is a positive relationship present in the AS model ( $R^2 = 0.50$ ).

### 3.3 Bioaccumulation in the global dataset



In Table 2, we show the results of a linear regression using the global dataset while accounting for both the trophic level and the feeding strategy; each model's relative fit explains Hg bioaccumulation based on both factors. The regressions are based on the natural logarithms of iHg, tHg, and MeHg as dependent variables. These linear regressions show that the bioaccumulation of MeHg can be predicted very well ( $R^2$ =0.72) with a linear model that takes both drivers into account, while iHg is poorly explained ( $R^2$ =0.11) and tHg shows intermediate explanatory power ( $R^2$ =0.46). Furthermore, we show the unique contributions of the fit of each driver, the partial  $R^2$ . Note that feeding strategy and trophic level can sometimes co-correlate, especially in the case of high MeHg bioaccumulation in predators, benthic fish, and seabirds, as predators naturally occupy higher trophic levels than their prey. The feeding strategy has greater explanatory power than that of the trophic level for tHg and iHg, while it is similar for MeHg. Despite the limitations mentioned above, this shows that the partial  $R^2$  for the feeding strategy is twice

that of the trophic level for tHg, demonstrating the importance of feeding strategy in Hg bioaccumulation at the base of the food web.

**Table 2.** R-squared and Partial R-squared Results for ln(THg), ln(iHg), and ln(MeHg)

| Model                                | ln(tHg) | ln(iHg) | ln(MeHg) |
|--------------------------------------|---------|---------|----------|
| Full Model R-squared                 | 0.46    | 0.11    | 0.72     |
| Partial R-squared (Feeding Strategy) | 0.22    | 0.089   | 0.32     |
| Partial R-squared (Trophic Level)    | 0.10    | 0.012   | 0.31     |

## 3.4 The allometric scaling law in high-trophic-level animals

In Table 3, we show the performance of the base and AS models against the global dataset. Table 3 shows that if we take the allometric scaling law into account, the model results for high-trophic-level animals improve considerably. In Fig. 4b, d, and e, we show the relationship between the natural logarithm of bioaccumulation and the trophic level of the AS model in the Southern North Sea setup. The normalized bias in the predator and the top predator decreased from -0.37 and -0.80 to -0.32 and -0.12, respectively. Our base model does agree well with both observed iHg ( $R^2$ =0.61) and MeHg ( $R^2$ =0.86) in the Southern North Sea setup, but this is mostly driven by accurate predictions in the lower trophic levels while there is a normalized bias of -0.80 in MeHg bioaccumulation in the top predator. The model performance in MeHg bioaccumulation is improved substantially in the AS model with the reduction of the normalized bias of the top predator to -0.22, which improves the overall  $R^2$  of the model to >0.99.

## 3.5 Comparing model and observations




## 3.5.1 The effect of feeding strategy on bioaccumulation

The annual mean and range of modeled bioaccumulated iHg and MeHg, along with the observed ranges and means, are shown in Table 4. We visualized the modeled values of the AS model in the Southern North Sea compared to the observations in Fig. 5. In Fig. 5a, the bioaccumulation of MeHg, and in Fig. 5b the bioaccumulation of iHg are visualized. All values fall within the range of observations, except for the modeled top predator in the base model. In the AS model, the top predator has values for both iHg and MeHg in both the Southern North Sea and the Northern North Sea that are within the range of observations. The most notable observation for iHg bioaccumulation is that, although the variation in measured iHg is considerable, suspension feeders consistently have high iHg values. In both the Southern North Sea setup and the observations, the mean MeHg is lowest in suspension feeders (17 and 8 ng Hg g<sup>-1</sup> d.w. respectively), while it is similar for deposit feeders (22 and 35 ng Hg g<sup>-1</sup> d.w. respectively), filter feeders (28 and 39 ng Hg g<sup>-1</sup> d.w. respectively), and generalist feeders (26 and 40 ng Hg g<sup>-1</sup> d.w. respectively). Observed MeHg is higher in predators (77 ng Hg g<sup>-1</sup> d.w.) and highest in top predators (381 ng Hg g<sup>-1</sup> d.w.).

Figure 4. The influence of trophic level on the bioaccumulation of MeHg, iHg, and tHg in both the AS model (panels a, c, e) and the base model (panels b, d, f). In the AS model, the relationship with trophic level is stronger, where ln(MeHg) = 1.24TL - 0.03, compared to the base model, which is ln(MeHg) = 0.64TL + 1.42. TL represents trophic level, and MeHg is expressed in ng Hg g<sup>-1</sup> d.w. For iHg, the bioaccumulation patterns are nearly identical, with ln(MeHg) = -0.19TL + 5.11 for the AS model and ln(MeHg) = -0.18TL + 5.11 for the base model, both showing a weak inverse correlation with trophic level, largely due to higher iHg levels in low-trophic-level feeders. In terms of tHg, there is a higher increase in bioaccumulation in the AS model (ln(MeHg) = 0.43TL + 3.76) compared to the base model (ln(MeHg) = 0.04TL + 4.75), driven by the stronger association between MeHg and trophic level in the AS model.

**Table 3.** Statistical analysis of model performance for iHg and MeHg levels by feeding strategy for the Southern North Sea (SNS) and Northern North Sea (NNS). The predator and top predator of both the default setup and the Allometric Scaling (AS) model are shown. For all individual feeding strategies, we show the normalized bias, and for the full model, the RMSE, NRMSE,  $R^2_{Pearson}$ , and  $R^2_{Residual}$ .

|                                         | s      | NS    | NNS   |       |  |
|-----------------------------------------|--------|-------|-------|-------|--|
|                                         | iHg    | МеНд  | iHg   | МеНд  |  |
| Suspension                              | 0.18   | 1.09  | -0.18 | 0.24  |  |
| Filter                                  | 1.48   | -0.28 | 1.45  | -0.69 |  |
| Deposit                                 | 1.01   | -0.36 | 0.34  | -0.75 |  |
| Generalist                              | 1.31   | -0.35 | 1.23  | -0.73 |  |
| Predator                                | 0.41   | -0.37 | 0.07  | -0.77 |  |
| Top predator                            | -0.22  | -0.80 | -0.46 | -0.92 |  |
| Predator (AS)                           | 0.41   | -0.31 | 0.07  | -0.75 |  |
| Top predator (AS)                       | -0.22  | -0.12 | -0.46 | -0.67 |  |
| Overall Model Perfo                     | rmance |       |       |       |  |
| RMSE                                    | 40     | 132   | 40    | 146   |  |
| NRMSE                                   | 0.36   | 0.35  | 0.35  | 0.39  |  |
| $R^2_{Pearson}$                         | 0.61   | 0.86  | 0.24  | 0.94  |  |
| R <sup>2</sup> Residual                 | <0     | <0    | <0    | <0    |  |
| RMSE (AS)                               | 40     | 22.8  | 40    | 108   |  |
| NRMSE (AS)                              | 0.36   | 0.061 | 0.35  | 0.29  |  |
| R <sup>2</sup> Pearson (AS)             | 0.61   | >0.99 | 0.24  | 0.99  |  |
| R <sup>2</sup> <sub>Residual</sub> (AS) | <0     | 0.96  | <0    | <0    |  |

These values closely match the modeled MeHg values of 54 and 337 ng Hg g<sup>-1</sup> d.w. in the AS model in the Southern North Sea.

## 3.5.2 The statistical performance of the model


Our model predicts that suspension feeders have the highest iHg values, which is in line with observations. In our model, the high iHg values are caused by the efficient Hg scavenging of small DOM particles. These small particles have the highest Hg/C ratio (as shown in Fig. 3) and can only be consumed by suspension feeders. This leads to very high iHg and low MeHg in suspension feeders. The partial replication of high iHg values in suspension feeders suggests that our model may have underestimated the role of DOM or that additional factors contributed to the observed elevated iHg levels. Orani et al. (2020) demonstrated that the extremely low MeHg/Hg ratio in suspension-feeding sponges may be caused by the demethylation of MeHg by symbiotic bacteria. Our study supports this finding, showing that high iHg and low MeHg values may partially result from DOM consumption by suspension feeders. However, bacterial demethylation may further explain why the model

**Table 4.** Comparison of modeled and observed Hg and MeHg bioaccumulation in different feeding strategies for the Southern North Sea (SNS), Northern North Sea (NNS), and field observations. Values are presented as ranges with means in parentheses. Units are ng Hg g d.w. for iHg and MeHg, and % for MeHg percentage. The bottom two rows are the predator and top predator from the AS model (AS).

|                   | l I           | Model (SNS)   |        | Model (NNS)  |               | Observations |              |              |        |
|-------------------|---------------|---------------|--------|--------------|---------------|--------------|--------------|--------------|--------|
|                   | iHg           | МеНд          | % MeHg | iHg          | МеНд          | % MeHg       | iHg          | МеНд         | % MeHg |
| Suspension        | 141-213 (180) | 14-20 (17)    | 9      | 72-186 (125) | 6-14 (10)     | 7            | 58-515 (152) | 1-26 (8)     | 5      |
| Filter            | 85-109 (97)   | 23-32 (28)    | 22     | 80-120 (96)  | 10-15 (12)    | 11           | 3-82 (39)    | 2-173 (39)   | 50     |
| Deposit           | 73-93 (83)    | 19-26 (22)    | 21     | 41-71 (55)   | 7-12 (9)      | 14           | 9-113 (41)   | 2-231 (35)   | 46     |
| Generalist        | 82-105 (94)   | 21-29 (26)    | 22     | 71-114 (90)  | 8-13 (11)     | 11           | 3-113 (40)   | 2-231 (40)   | 50     |
| Predator          | 62-67 (65)    | 47-50 (49)    | 43     | 45-51 (49)   | 16-19 (18)    | 27           | 9-329 (46)   | 4-367 (77)   | 63     |
| Top predator      | 83-91 (88)    | 69-76 (73)    | 45     | 51-71 (61)   | 26-39 (32)    | 34           | 69-266 (113) | 77-895 (381) | 77     |
| Predator (AS)     | 45-48 (47)    | 52-55 (54)    | 54     | 45-51 (49)   | 18-20 (19)    | 28           | 9-329 (46)   | 4-367 (77)   | 63     |
| Top predator (AS) | 62-66 (64)    | 320-348 (337) | 84     | 51-71 (61)   | 109-147 (127) | 68           | 69-266 (113) | 77-895 (381) | 77     |

**Figure 5.** Comparison of bioaccumulation across feeding strategies between the Southern North Sea AS model and observations. The bars represent the mean with the error bar showing 1 Standard Error (SE). Figure 5a) shows MeHg bioaccumulation. Notably is that top predators show the highest MeHg levels, followed by predators, with generalists and filter feeders at intermediate levels and deposit feeders at lower levels while suspension feeders have the lowest MeHg. 5b) shows iHg Bioaccumulation. Suspension feeders show increased iHg, while all other categories except top predators are overestimated by the model. Top predators have high observed iHg not fully captured by the model.

underestimates observed patterns. Based on this, it is likely that the unique bioaccumulation patterns in suspension feeders are caused by a combination of their ability to feed on DOM and biochemical processes that occur in their symbiotic bacteria. Notably, our model overestimates the mean iHg values with a normalized bias of 1.48 and 1.45 for filter feeders, and 1.31 and 1.23 for generalist feeders in the Southern North Sea and Northern North Sea, respectively. In Fig 3, we see that the majority of this iHg originates from bioconcentration. This discrepancy is described in more detail later in the paper.

The  $R^2_{Pearson}$  is high (>0.86) for MeHg in all setups, and exceeds 0.99 in the AS model, indicating that the model captures the relative differences between feeding strategies well. For iHg, the performance is lower, particularly in the Northern North Sea ( $R^2_{Pearson} = 0.24$ ). The ability to reproduce absolute bioaccumulation is more limited. Only the AS model in the Southern North Sea shows good agreement ( $R^2_{Residual} = 0.96$ ), while all other setups yield  $R^2_{Residual} 

**Figure 6.** The natural logarithm of bioaccumulation for a) MeHg, b) iHg, and c) tHg in the permanently mixed Northern North Sea model shows that while the slope for MeHg bioaccumulation is comparable in the Northern and Southern North Sea with a slope of 0.55 and 0.64 respectively, its overall level of MeHg bioaccumulation is lower than in the Southern North Sea. Similar as in the Southern North Sea, there's no significant correlation between trophic level and iHg bioaccumulation.

## 3.6 The role of the feeding strategy on MeHg bioaccumulation in a single case study




In addition to using the global bioaccumulation dataset to evaluate our hypothesis that the feeding strategy is a key driver of bioaccumulation, we also assess whether this hypothesis holds when analyzing a comprehensive published dataset from a single study. The fit of a linear model to the natural log of bioaccumulated MeHg based on the data published by McClelland et al. (2024) is shown in Fig. 7. The R<sup>2</sup> is similar at 0.43 and 0.45 in the CB and MT, respectively, while bioaccumulation is slightly lower in the CB  $(ln(MeHg_{BA}) = 0.137 + 1.14TL)$  compared to the MT  $(ln(MeHg_{BA}) = 0.256 + 1.39TL)$ . Here, MeHg<sub>BA</sub> is the bioaccumulated MeHg in ng Hg mg<sup>-1</sup> d.w., and TL is the trophic level. The influence of the feeding strategy on MeHg bioaccumulation based on the results of McClelland et al. (2024) is shown in Table 5. While the only significant effect is that deposit feeders in the MT have less MeHg than would be expected given their trophic positions, some other effects are consistent, although not significant, in both locations. The strongest effect is that filter feeders consistently have higher MeHg (residuals are 0.7 in the CB and 0.8 in the MT), while deposit feeders have lower MeHg (residuals are -0.2 in the CB and -0.5 in the MT). The effects of phylum on MeHg bioaccumulation are shown in Table 6. Here we see two consistent significant effects. Molluscs have elevated MeHg levels (residuals are 0.61 in the CB and 0.51 in the MT), while arthropods have reduced MeHg values (residuals are -0.35 in the CB and -0.30 in the MT). The percentage difference in MeHg bioaccumulation per feeding strategy is visualized in Fig. 8 and per phylum in Fig. 9. The average percentage difference between observed values and the expectation based on trophic level is 102% and 128% in the CB and MT, respectively, for filter feeders. Deposit feeders have 19% and 37% less MeHg than predicted based on trophic level alone in the CB and MT, respectively. In the phylum-level analysis, we see that molluscs have highly elevated MeHg levels with an increase of 66% (CB) and 85% (MT). The largest reduction in observed MeHg compared to the predicted values based on trophic level is in arthropods; here there is a decrease of 29% and 26% in the CB and MT, respectively.

**Figure 7.** The linear fitted model between the natural logarithm of the bioaccumulated MeHg in ng Hg mg<sup>-1</sup> d.w. and the Trophic Level in the data presented by McClelland et al. (2024). For extra clarity the different Phyla shown with different colors while the different feeding strategies are marked with different symbols. In both the CB and MT setups there positive relationship between trophic level and the bioaccumulation of MeHg, but R<sup>2</sup> is only 0.43 and 0.45 in the CB and MT respectively, so it does not explain the full variation in bioaccumulation.

**Table 5.** Mean residuals ( $\pm$ SE) of ln(MeHg) by feeding strategy and region, after trophic level correction. Significant deviations (p 

Figure 8. Percentage difference from trophic level predicted MeHg concentrations by feeding strategy, with error bars showing  $\pm 1$  SE. In both CB and MT regions, filter feeders have elevated MeHg levels relative to trophic level based expectations, while deposit feeders are reduced. Predators display higher MeHg than predicted, though the effect is smaller than in filter feeders; in CB, this increase does not exceed one SE. Generalist feeders have a slight reduction compared to expectations, but this is well within one SE, and were not present in CB for cross-region comparison.

Figure 9. Percentage difference from the predicted MeHg bioaccumulated based on trophic level per phylum, the error bars represent  $\pm 1$  SE. The notable phyla are Mollusca and Arthropoda, while Mollusca have a notable increase in MeHg bioaccumulation compared to the prediction of 85% and 66% respectively in the CB and MT, there is a reduction of 26% and 29% in Arthropoda in the CB and MT respectively. Annelida are inconsistent with an increase in the CB and decrease in the MT compared to the predictions. Echinodermata have a mean reduction compared to the prediction in both the CB and the MT, but the SE is much larger than the mean effect.

**Table 6.** Mean residuals ( $\pm$ SE) of ln(MeHg) by phylum and region, after trophic level correction. Significant deviations (p < 0.05) are marked with \*.

| Region | Phylum        | n  | Mean Residual $\pm$ SE | p-value |
|--------|---------------|----|------------------------|---------|
| СВ     | Annelida      | 6  | $0.229 \pm 0.424$      | 0.612   |
| CB     | Arthropoda    | 11 | $-0.349 \pm 0.137$     | 0.0294* |
| CB     | Echinodermata | 3  | $-0.198 \pm 0.584$     | 0.767   |
| CB     | Mollusca      | 5  | $0.611 \pm 0.211$      | 0.0446* |
| MT     | Annelida      | 5  | $-0.405 \pm 0.377$     | 0.343   |
| MT     | Arthropoda    | 12 | $-0.304 \pm 0.111$     | 0.0196* |
| MT     | Echinodermata | 5  | $-0.188 \pm 0.509$     | 0.730   |
| MT     | Mollusca      | 13 | $0.509 \pm 0.231$      | 0.0482* |

and -40% in filter and deposit feeders, respectively. In the CB, the changes are smaller, with 7.2% and -14.8% in filter and deposit feeders, respectively. It must be stated that this final analysis is included to address potential concerns regarding the co-correlation between phyla and feeding strategy, although the reduced sample size remains a limitation. In the CB, where the increase in MeHg in filter feeders is rather low after correcting for both trophic level and feeding strategy, there are only three filter feeders, which are molluscs. These three make up 3/5 of the mollusc samples in this location, so these results should be interpreted with caution, as filter feeders and molluscs overlap strongly. On the other hand, in the MT, there are five filter feeders from multiple phyla (Mollusca and Echinodermata), and the effect is considerably stronger, with filter feeders having 118% more MeHg than would be expected based on their trophic level and phyla.

**Table 7.** The effect of feeding strategy on MeHg bioaccumulation per Region compared to the prediction accounting for both trophic level and feeding strategy. Significant (p < 0.05) is marked with \*. There is still a consistent increase in filter feeders and a consistent decrease in deposit feeders. This is effect is larger in the MT with a relative percentage increase of 118% in filter feeders and a decrease of 40% in deposit feeders.


| Feeding Strategy | % Diff (MT) | p-value (MT) | % Diff (CB) | p-value (CB) |
|------------------|-------------|--------------|-------------|--------------|
| Deposit feeder   | -40.0       | 0.034*       | -14.8       | 0.888        |
| Filter feeder    | 118.0       | 0.034*       | 7.2         | 0.888        |
| Generalist       | -25.9       | 0.563        | _           | _            |
| Predator         | 3.0         | 0.895        | 9.4         | 0.888        |

## 4 Discussion

## 4.1 The role of feeding strategy on the bioaccumulation of MeHg

Overall we find that the feeding strategy plays an important role in the bioaccumulation of MeHg in our model, the global dataset, and the single dataset published by McClelland et al. (2024). Because of this, we find it convincing that the role of the feeding strategy in MeHg bioaccumulation deserves further attention in both modeling and empirical studies.

## 4.2 The AS model





While the base model underestimates MeHg in top predators, this improvement is observed in the AS model. The normalized bias for MeHg bioaccumulation in top predators in the Southern North Sea decreased from -0.80 to -0.12. In the AS model, we obtain a linear relationship of  $\ln(\text{MeHg}) = 1.24x - 0.03$  ( $R^2 = 0.93$ ). This slope is similar to the linear relationships found in the CB and MT stations of the McClelland et al. (2024) dataset, which are 1.14x + 0.387 and 1.39x + 0.256, respectively. The improvement in the AS model compared to the base model indicates that lower MeHg release rates in high-trophic-level animals should be accounted for when modeling MeHg bioaccumulation in higher trophic levels. We also tested the model with the lower MeHg release rate applied to all megabenthos, but this resulted in unrealistically high values both at the base and top of the food web. Therefore, implementing the allometric scaling law is preferable to lowering the MeHg release rate at every trophic level. We conclude that, in addition to feeding strategy, differences in MeHg release rates associated with body size, metabolism, or activity likely contribute significantly to the high MeHg values in high-trophic-level animals.

#### 4.3 Bioconcentration of iHg

The largest bias in our model, which remains uncorrected in the AS model, is the overestimation of iHg in filter and generalist feeders. Although the modeled iHg values are not outside the observed range, the consistently high normalized bias indicates that the model overestimates iHg bioaccumulation. In Fig. 3, we can see that the vast majority of iHg in filter and generalist feeders originates from bioconcentration. The most important driver of bioconcentration is the ratio between uptake and release rates, or the uptake-release ratio. Our model uses an uptake-release ratio of 210 1 g<sup>-1</sup> d.w., derived from Tsui and Wang (2004), as it represents the lowest ratio reported in the literature. The exact rate was obtained by subtracting the modeled carbon excretion rate from the measured iHg release rate to obtain an iHg-specific release rate, which was found to be 0.04 d<sup>-1</sup>, as presented in Amptmeijer et al. (2025). Other studies, such as Pan and Wang (2011), found higher uptake-release ratios between 424 and 781 1 g<sup>-1</sup> d.w.

The discrepancy between modeled and observed iHg may stem from several factors. First, iHg concentrations in North Sea megabenthos could be higher than those reported in other coastal zones. However, there are currently no empirical data to confirm or refute this. Second, translating experimentally obtained uptake and release rates to observations of iHg may depend on drivers that are not captured in the model. In either case, it is difficult to verify the cause of this high normalized bias, as iHg bioaccumulation is comparatively understudied relative to MeHg, both in models and empirical studies.

### 4.4 Model structural limitations

The GOTM-MERCY-ECOSMO coupled system captures the influence of feeding strategy on MeHg bioaccumulation, but performance differs between regions. The Southern North Sea setup performs well in pelagic Hg cycling and benthic bioaccumulation, whereas the Northern North Sea setup underestimates MeHg in all benthic groups and shows unexpectedly high mesozooplankton tHg. While this cannot be directly validated due to a lack of data, the model predicts lower MeHg bioaccumulation in deeper water, which does not match observations by McClelland et al. (2024). This suggests that MeHg fluxes from the pelagic to the benthic system are underestimated.

In shallow waters, megabenthos can feed directly on phytoplankton and zooplankton blooms, which leads to strong benthopelagic exchange of organic carbon and Hg. In deeper waters, megabenthos mainly rely on detritus that sinks from the euphotic zone, which, in our model, carries less MeHg. The higher performance in shallow conditions combined with the reduced performance in deeper conditions indicates that the model could be improved in processes controlling deep-water MeHg bioaccumulation, such as sediment Hg chemistry, deep-water Hg speciation, bentho-pelagic coupling, or transport of Hg due to sinking organic material.

## 480 4.5 Data-related limitations





Combining the results of the model and literature studies is difficult due to the high uncertainty in most drivers, including the organic material content of dry weight. Therefore, the results should be viewed with caution. For example, the data analyses by McClelland et al. (2024) were prepared to mimic consumption by predators: for small arthropods, their skins were not removed, but for gastropods and bivalves, the shells were not included in the weight, as predators typically would not eat them.

The concentration of MeHg per unit energy is arguably the key measure in bioaccumulation. Predators need to ingest a specific amount of energy, so if a prey is composed of half organic material and half non-organic components, such as shell, its MeHg concentration per dry weight is halved. However, predators would consume double the dry weight to obtain the same energy, and thus the MeHg intake remains unchanged. In general, energy content is best approximated by Ash Free Dry Weight (AFDW). Ideally, all MeHg bioaccumulation measurements should be normalized to AFDW (Weil et al., 2019). Unfortunately, performing this conversion reliably on published data is not possible, as AFDW varies with the age and body size of animals, information that is not always reported or made available (Eklöf et al., 2017).

## 4.6 Potential improvements

The model uses the same rates for all megabenthos groups. This allows us to isolate the effect of feeding strategy, but it also limits the model's ability to predict bioaccumulation of iHg or MeHg in specific animals. Our simulations are run for the North Sea, whereas most field observations come from different regions. Therefore, this study should be seen as hypothesis-generating, identifying the role of feeding strategies in iHg and MeHg bioaccumulation as a potential direction for further empirical studies, rather than providing a robust quantification.

Based on our results, the inclusion of megabenthos with different feeding strategies could improve the performance of MeHg bioaccumulation models. At the same time, our analyses demonstrate the underperformance of the model in simulating deep-water bentho-pelagic coupling. This indicates that the ECOSMO-MERCY-GOTM coupled system should be critically evaluated before being used for predictive bioaccumulation modeling in deeper waters.

#### 5 Summary and conclusion






In this study, we analyze the role of the trophic level and the feeding strategy on the bioaccumulation of iHg and MeHg. We did this by performing a literature study and running a fully coupled 1D model in two idealized setups representing two different hydrodynamic regimes in which megabenthic communities can live. Our study estimates that trophic level alone explains up to 32% of the variability in MeHg concentrations in the benthic food web. Including both trophic level and feeding strategy increases this explained variability to 72%, highlighting significant differences between feeding strategies.

Additionally, we demonstrate that there are notable differences between feeding strategies. iHg is higher in suspension feeders and MeHg is low in suspension feeders and grazers, while filter feeders have the highest MeHg followed by deposit feeders. Our model expands on this by demonstrating that we can accurately model the bioaccumulation of iHg and MeHg at the base of the food web by only taking the feeding strategy into account.

Our model results, together with both aggregated and single-study literature analyses, consistently suggest that feeding strategy is a key driver of bioaccumulation at the base of the food web. While these findings should be interpreted cautiously, due to the inherent uncertainty in the model, the consistent indication that feeding strategy is a key driver in MeHg bioaccumulation does suggest it is a potential target for further empirical studies. In the Southern North Sea setup, feeding strategy in our base model correlates well with observed iHg ( $R^2$ =0.61) and MeHg ( $R^2$ =0.86), suggesting it is a key driver of bioaccumulation at the base of the food web. This strong performance largely reflects the fact that four of our six megabenthos groups are low-trophiclevel non-predators. The model underperforms in predicting MeHg bioaccumulation in higher-trophic-level organisms. This is improved upon by accounting for the allometric scaling law and assuming that MeHg removal from the organism is not linked to the total but rather to the base metabolic rate. Because of this, we conclude that our hypothesis that the feeding strategy is an essential driver of the bioaccumulation of iHg and MeHg in low-trophic-level animals is supported by both our model and observations, but our model shows that other differences in the organisms between high- and low-trophic-level animals should also be taken into account when predicting MeHg values in high-trophic-level fish. Our model and observation focus on lower-trophic-level benthic invertebrates, with some high-trophic-level animals added to create context. The importance of this for the bioaccumulation of MeHg in high-trophic-level animals is that all biomagnification is an exponential function starting at the base of the food web. Therefore, a change in MeHg at the base of the food web will correspond to a similar relative increase at the top of the food chain. Because the feeding strategy has such a large impact on the base of the food web, high-trophic-level animals could have considerably different MeHg values depending on the species composition of the base of the food web.

Interestingly, although iHg has a lower biomagnification potential, its high concentrations in some low-trophic-level animals can result in higher tHg levels in these organisms than in higher-trophic-level animals, demonstrating the need to measure different Hg species to not misidentify the toxicity of biota.

### 5.1 Societal relevance & future work





Our study highlights the critical role of benthic diversity in driving MeHg bioaccumulation. Both trophic interactions and the feeding strategy significantly influence MeHg bioaccumulation, which has important implications for seafood safety and fisheries management. Understanding these processes can help explain the spatial and temporal variability in the MeHg content of fish, which is crucial for policymakers to develop effective regulations that safeguard human health and marine ecosystems.

Filter feeders and molluscs typically accumulate more MeHg than other organisms at similar trophic levels. This pattern is consistent not only in our models but also in available data. This raises a hypothesis that expanding bivalve populations, as seen in mussel or oyster farming, might affect MeHg bioaccumulation in higher trophic levels. This is supported by the observations that fish in lakes invaded by zebra mussels have higher Hg levels than fish in lakes without zebra mussels (Blinick et al., 2024). However, such ecological alterations also impact other bioaccumulation factors like biomass distribution and trophic interactions. While our findings support the role of filter feeders and molluscs in MeHg dynamics and higher bioaccumulation in top predators, the complexity of ecological situations requires further case-specific studies to understand if and when bivalve communities lead to increased MeHg transfer.

Modeling studies can help our understanding of the factors influencing MeHg bioaccumulation, but the ability to accurately predict MeHg bioaccumulation needs to be carefully validated. Our findings reveal that filter-feeding molluscs and DOM-utilizing suspension feeders have different Hg bioaccumulation patterns compared to other megabenthos. Modeling bivalve aquaculture or DOM-consuming suspension feeders can help explore their potential role in altering MeHg bioaccumulation. Understanding how functional traits, such as feeding strategy, influence MeHg transfer is essential for improving predictive models and environmental risk assessments. Based on our results, we strongly recommend targeted field studies that systematically measure iHg, MeHg, and trophic level in diverse marine communities. Such studies would clarify how food web structure affects MeHg bioaccumulation in seafood.

#### 6 Acknowledgments

Readability suggestions for this paper were generated using rAI tools such as ChatGPT (OpenAI), while AI-based spell checks such as Grammarly and Writefull were used to correct spelling. In addition, rAI tools helped optimize the R and Python scripts and provide coding suggestions. All suggestions were implemented only after critical manual evaluation. Finally, Google Scholar and Perplexity were used to find sources for literature research, which were consequently manually read, verified, and cited.

## 560 Author contributions

The contributions per author are listed in Table 8.

**Table 8.** Contributions per Author. Authors are: David Johannes Amptmeijer (DA), Andrea Padilla (AP), Sofia Modesti (SM), Prof. Dr. Corinna Schrum (CS), and Dr. Johannes Bieser (JB).

| Contributor role                                                | Role definition                                   | Authors        |
|-----------------------------------------------------------------|---------------------------------------------------|----------------|
| Conceptualisation                                               | Conceptualized the study                          | DA, JB, CS     |
|                                                                 | Developed the research objectives                 | DA, JB, CS     |
| Methodology                                                     | Implementation of the model into FABM             | DA             |
| outodology                                                      | Compiled the database of megabenthos iHg and MeHg | DA, AP         |
|                                                                 | observations                                      |                |
| Evaluation Evaluated the model performance against observations |                                                   | JB, DA, AP, SM |
|                                                                 | Performed statistical tests on the observations   | DA, AP, SM     |
| Writing of the original draft                                   |                                                   | DA             |
|                                                                 | Review of the original draft and quality control  | AP, SM, JB, DA |
| Supervision                                                     | Supervised the development of the work            | CS, JB, DA     |
| Funding acquisition                                             | Acquired funding via the GMOS-Train ITN           | JB             |

# **Conflict of interest**

None of the authors declare any conflict of interest.

# **Funding**

This research has been funded by the European Union's Horizon 2020 research and innovation programme under the Marie Sklodowska-Curie grant agreement no. 860497.

# Code availability

The model code is publicly available on Zenodo (DOI: 10.5281/zenodo.17372353) under the GPL 3.0 License.

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
