# Peer review of "Feeding strategy as a key driver of the bioaccumulation of MeHg in megabenthos"

_EGUsphere, 2025_

## Author Comment (AC1)

**1 Answers to reviewer 2**

**Author Response**

Dear reviewer,

Thank you for your comments and taking the time to review the paper. Before I get to answering your specific comments, I want to respond to what I agree is a very general criticism of the paper that should be properly addressed, which is related to using data from different geographic areas. The issue is that there is very little data on MeHg bioaccumulation at the base of the food web, and almost nothing in the North Sea. This complicates model validation and is the reason why the decision was made to simply try to use all data available. However, 2 recent papers came out with a lot of data, of which one paper, by McClelland et al., 2024 samples 476 benthic animals from two locations in the Canadian Arctic. While this is a different location, it does allow us to evaluate if certain patterns between feeding strategies are consistent in samples from the same geographic location. Because of this, I suggest to expand the paper by a third component, where I evaluate if we see an effect of feeding strategy on MeHg bioaccumulation. I suggest to add the analyses below:

**Suggested edit**

A third test of our hypothesis using field data from a single study to see if the same dynamics as occured in our model are present in these observations. We used MeHg bioaccumulation and trophic level data from 476 individuals across 53 taxa of benthic invertebrates as published by McClelland et al. (2024). These data were selected as it is the laregest study we could find with both trophic level and MeHg concentrations. When several animals of the same group where sampled, the dataset presents mean values per group per location, which we use as one datapoint in our analyses. Although feeding strategies in the dataset were broadly aligned with our classifications, we reassigned them to match the functional groups in our model. For example, shrimps were categorized as generalist feeders, which group is not present in McClelland et al. (2024), and isopods, which can be small benthic predators, were labeled as deposit feeders because their prey type is not represented in our model.

The data is sampled from two locations in the Canadian Arctic, Cape Bathurst (CB), which has a depth of 22 m and is located at 70°41′42.79″ N, 128°50′21.34″ W, and the eastern coast of Herschel Island in the Mackenzie Trough (MT), which has a depth of 116 m and is located at 69°36′44.96″ N, 138°33′45.25″ W. It must be noted this dataset is selected as it is extensive, but the region does have notable differences to the North Sea, where our model is run. It has extensive ice cover in winter, it is colder and is geographically distant from the model location. While the Arctic ecosystem differs from the North Sea, the use of a single, internally consistent dataset allows us to isolate feeding strategy effects better than using the global dataset and verify our conclusions.

**Author Response**

I will add this to the methods section at line 218

**Suggested edit**

**Evaluation of the model using a single dataset**

To isolate the effect of the feeding strategy on MeHg bioaccumulation, we first transformed MeHg concentrations to their natural logartitm and fit a linear model with trophic level as predictor using the base R `lm()` function. The significance of the deviation from the predicted MeHg concentration at the trophic level was assessed using a one-sample $t$ tests. To improve interpretability, we calculated the percentage differences using Percentage difference $= 100 \times \left( \frac{\text{MeHg}_{\text{obs}}}{\text{MeHg}_{\text{pred}}} - 1 \right)$ based on the residuals of the linear fit. This is visualized on a bar graph showing the percentage difference in MeHg concentration caused by the feeding strategy. The errors bars represent the mean $\pm$ 1 Standard Error (SE). The same analysis was then performed to estimate differences in MeHg bioaccumulation related to phylum.

As a final test, linear models were fitted on the natural logarithm of bioaccumulated MeHg concentrations using trophic level, phylum, and feeding strategy as predictor variables (using the `lm()` function in R). Estimated marginal means (EMM) for each feeding strategy were calculated with the `emmeans()` function of the emmeans package and compared against the overall mean to assess deviations. This analysis was also performed separately for the MT and CB locations to verify the consistency of the effects of the feeding strategy. The EMMs where transformed to a percentage difference with the earlier used equation and the estimated percentage difference due to feeding strategy and its signifance is shown.

**Author Response**

I will add this to the result and discussion about the literature study result section (3.1). At line 351.

**Suggested edit**

The fit of the linear model against the natural logarithm of the bioaccumulated MeHg is shown in Fig. 1. The $R^2$ is similar with 0.43 and 0.45 in the CB and MT respectively, while the bioaccumulation is a bit lower in the CB (log(MeHg$_{\text{BA}}$)=0.137+1.14*TL) compared to that in the MT (log(MeHg$_{\text{BA}}$)=0.256+1.39*TL), where MeHg$_{\text{BA}}$ is the bioaccumulated MeHg in ng Hg mg$^{-1}$ d.w. and TL is the trophic level. The influence of the feeding strategy on MeHg bioaccumulation based on the results of McClelland et al. (2024) is shown in Table 1. While the only significant effect is deposit feeders in the MT having less MeHg than would be expected on their trophic positions, some other effects are consistent, albeit not significant in both locations. The strongest effect is that filter feeders have consistently higher MeHg (residuals are 0.7 in the CB and 0.8 in the MT), while deposit feeders have lower MeHg (residuals are -0.2 in the CB and -0.5 in the MT). The results of the same analyses for phyla are shown in Table 2. Here we see two consistent significant effects. Molluscs have elevated MeHg levels (residuals are 0.61 in the CB and 0.51 in the MT) while arthropods have reduced MeHg values (residuals are -0.35 in the CB and -0.30 in the MT). The percentage difference in MeHg bioaccumulation per feeding strategy is visualised in Fig. 2 and per phyla in Fig. 3. The average percentage difference between observed and the expectation based on trophic level is 102% and 128% in the CB and MT respectively for filter feeders while deposit feeders have 19 and 37% less MeHg than would be predicted based on trophic level alone in the CB and MT respectively. If we repeat the analysis per Phylum we see that molluscs have highly elevated MeHg levels with an increase of

[Figure]

Figure 1: The linear fitted model between the natural logarithm of the bioaccumulated MeHg in ng Hg mg$^{-1}$ d.w. and the Trophic Level in the data presented by McClelland et al. (2024). For extra clarity the different Phyla shown with different colors while the different feeding strategies are marked with different symbols. In both the CB and MT setups there positive relationship between trophic level and the bioaccumulation of MeHg, but R$^2$ is only 0.43 and 0.45 in the CB and MT respectively, so it does not explain the full variation in bioaccumulation.

66% and 85% respectively in the CB and MT. The largest reduction in observed MeHg compared to the predicted values based on trophic level is in Arthropoda, here there is a decrease compared to the predicted values of 29% and 26% in the CB and MT respectively.

The results of the final analyses are shown in Table 3. Despite the lower sample size, which reduces statistical power, the results indicate that filter feeders consistently have higher MeHg levels than predicted based on their trophic position and phyla, while deposit feeders tend to have lower MeHg concentrations. These results are stronger in the MT with a change of 118% and -40% in filter and deposit feeders respectively than in the CB with a change of 7.2% and -14.8% in filter and deposit feeders respectively. It must be stated that this final analysis is included to address potential concern between the co-correlation of phyla and feeding strategy, but the problem of the reduced sample size has to be addressed. In the CB, where the increase in MeHg in filter feeders is rather low after correcting for both trophic level and feeding strategy, there are only 3 filter feeders, which are molluscs, and they make up 3/5 mollusc samples in this location, meaning that results should be seen with skepticism as filter feeders and molluscs have too much overlap. On the other hand, in the MT, there are 5 filter feeders from multiple phyla (Mollusca and Echinodermata) and the effect is considerably stronger with filter feeders having 118% more MeHg than would be expected based on their trophic level and phyla.

[Figure]

Figure 2: Percentage difference from trophic level predicted MeHg concentrations by feeding strategy, with error bars showing 1 SE. In both CB and MT regions, filter feeders have elevated MeHg levels relative to trophic level based expectations, while deposit feeders are reduced. Predators display higher MeHg than predicted, though the effect is smaller than in filter feeders; in CB, this increase does not exceed one SE. Generalist feeders have a slight reduction compared to expectations, but this is well within one SE, and were not present in CB for cross-region comparison.

[Figure]

Figure 3: Percentage difference from the predicted MeHg bioaccumulated based on trophic level per phyla, the error bars represent the mean 1 SE. The notable phyla are Mollusca and Arthropoda, while Mollusca have a notable increase in MeHg bioaccumulation compared to the prediction of 85% and 66% respectively in the CB and MT, there is a reduction of 26% and 29% in Arthropoda in the CB and MT respectively. Annelida are inconsistent with an increase in the CB and decrease in the MT compared to the predictions. Echinodermata have a mean reduction compared to the prediction in both the CB and the MT, but the SE is much larger than the mean effect.

Table 1: Mean residuals (±SE) of log(MeHg) by feeding strategy and region, after trophic level correction. Significant deviations (p < 0.05) are marked with *.

| Region | Feeding Strategy | $n$ | Mean Residual ± SE | p-value |
|--------|------------------|-----|--------------------|---------|
| CB | Deposit feeder | 16 | $-0.208 \pm 0.181$ | 0.268 |
| CB | Filter feeder | 3 | $0.704 \pm 0.286$ | 0.133 |
| CB | Predator | 6 | $0.203 \pm 0.331$ | 0.568 |
| MT | Deposit feeder | 15 | $-0.467 \pm 0.159$ | 0.011* |
| MT | Filter feeder | 5 | $0.824 \pm 0.395$ | 0.105 |
| MT | Generalist | 3 | $-0.143 \pm 0.319$ | 0.698 |
| MT | Predator | 12 | $0.277 \pm 0.226$ | 0.247 |

Table 2: Mean residuals (±SE) of log(MeHg) by phylum and region, after trophic level correction. Significant deviations (p < 0.05) are marked with *.

| Region | Phylum | $n$ | Mean Residual ± SE | p-value |
|--------|--------|-----|--------------------|---------|
| CB | Annelida | 6 | $0.229 \pm 0.424$ | 0.612 |
| CB | Arthropoda | 11 | $-0.349 \pm 0.137$ | 0.0294* |
| CB | Echinodermata | 3 | $-0.198 \pm 0.584$ | 0.767 |
| CB | Mollusca | 5 | $0.611 \pm 0.211$ | 0.0446* |
| MT | Annelida | 5 | $-0.405 \pm 0.377$ | 0.343 |
| MT | Arthropoda | 12 | $-0.304 \pm 0.111$ | 0.0196* |
| MT | Echinodermata | 5 | $-0.188 \pm 0.509$ | 0.730 |
| MT | Mollusca | 13 | $0.509 \pm 0.231$ | 0.0482* |

Table 3: The effect of feeding strategy on MeHg bioaccumulation per Region compared to the prediction accounting for both trophic level and feeding strategy. Significant (p < 0.05) is marked with *. There is still a consistent increase in filter feeders and a consistent decrease in deposit feeders. This is effect is larger and significant in the MT with a relative percentage increase of 118% in filter feeders and a decrease of 40% in deposit feeders.

| Feeding Strategy | % Diff (MT) | p-value (MT) | % Diff (CB) | p-value (CB) |
|------------------|-------------|--------------|-------------|--------------|
| Deposit feeder | -40.0 | 0.034* | -14.8 | 0.888 |
| Filter feeder | 118.0 | 0.034* | 7.2 | 0.888 |
| Generalist | -25.9 | 0.563 | – | – |
| Predator | 3.0 | 0.895 | 9.4 | 0.888 |

**Author Response**

**Updated statistics due to extra data in the analyses**

In addition to the extra papers added, a final small change was made: the assimilation efficiency of the deposit feeder was set to 0.12, as this is more in line with the literature presented by Dutton and Fisher (2012). To clarify, this and the addition of two more papers, of course, change the statistics and also highlight your concern about different base levels of Hg. The new statistical analyses are presented in Table 4. Because the new data has a higher baseline of MeHg, it indeed changes (in this case reduces) the agreement between the model and the observations in absolute terms. It does, however, not mean that the correlation between feeding strategies and elevated MeHg levels is different, as this remains consistent. To increase transparency of model fit, I will both show the $R^2$ based on the Pearson correlation coefficient and the Residual-Based $R^2$.

**Suggested edit**

Finally, the goodness of fit of the model is shown both based on the Pearson correlation coefficient and based on the residuals. It is important to note the difference between the $R^2$ based on the Pearson correlation coefficient ($R^2_{Pearson}$) and the Residual-Based $R^2$ ($R^2_{Residual}$). They are defined as $R^2_{Pearson} = (corr(y_{mod}, y_{obs}))^2$ and the $R^2_{Residual} = 1 - \frac{\sum(y_{obs} - y_{mod})^2}{\sum(y_{obs} - \bar{y}_{obs})^2}$, where $y_{obs}$ are the mean observed values and $y_{mod}$ are the mean modeled value per feeding strategy. The $R^2_{Pearson}$ describes how well a linear relationship between our model and the data holds, thus, if this is high, the models explain the observed differences between feeding strategies well, but it is not effected by the absolute concentrations. This means that the $R^2_{Pearson}$ evaluates if our model accurately represents differences between feeding strategies, but it is not affected by a bias in the data due to mismatch between the baseline of MeHg bioaccumulation. The $R^2_{Residual}$ measures how well the modeled values match the observations, including absolute values, which is both affected by the ability of the model to predict the effect of feeding strategy on bioaccumulation, and by overal concentrations of iHg and MeHg in biota. Both $R^2_{Pearson}$ and $R^2_{Residual}$ give a value below 1. Closer to 1 means a better fit while below 0 means that the the model does not outperform estimating the data based on the mean.

**Suggested edit**

The $R^2_{Pearson}$ is high (>0.86) for MeHg in all setups and exceeds 0.99 in the AS model, indicating that the model captures the relative differences between feeding strategies well. For iHg, performance is lower, particularly in the Northern North Sea ($R^2_{Pearson} = 0.24$). The ability to reproduce absolute bioaccumulation is more limited. Only the AS model in the Southern North Sea shows good agreement ($R^2_{Residual} = 0.96$), while all other setups yield $R^2_{Residual} < 0$, suggesting that using the mean of the observations would outperform the model.

This can be explained, as baseline MeHg levels vary between sampling regions. Notably, the AS model in the Southern North Sea performs well both in reproducing overall MeHg levels and in explaining variability across feeding strategies. Even when excluding predators and top predators, $R^2_{Pearson}$ remains high (0.80), suggesting that feeding strategy effects are captured across trophic levels and are not just driven by high MeHg levels in predatorial feeding strategies. In contrast, the Northern North Sea has a high $R^2_{Pearson}$ (=0.94) but low $R^2_{Residual}$ (<0) so it captures the effect of feeding strategies while not being able to resplicate absolute MeHg concentrations.

Table 4: Statistical analysis of model performance for iHg and MeHg levels by feeding strategy for Southern North Sea (SNS) and Northern North Sea (NNS).

| | SNS | | | | NNS | | | |
|---|---|---|---|---|---|---|---|---|
| | iHg | | MeHg | | iHg | | MeHg | |
| | N. Bias | BF10 | N. Bias | BF10 | N. Bias | BF10 | N. Bias | BF10 |
| **Suspension** | 0.18 | 0.11 | 0.85 | 0.011 | -0.18 | 0.11 | 0.17 | 0.0072 |
| **Filter** | 1.48 | 0.47 | -0.33 | 0.046 | 1.45 | 0.069 | -0.71 | 0.054 |
| **Deposit** | 1.01 | 0.078 | -0.44 | 0.046 | 0.34 | 0.034 | -0.77 | 0.052 |
| **Generalist** | 1.31 | 0.18 | -0.39 | 0.047 | 1.23 | 0.14 | -0.75 | 0.055 |
| **Predator** | 0.41 | 0.081 | -0.42 | 0.10 | 0.07 | 0.079 | -0.71 | 0.074 |
| **Top predator** | -0.22 | 0.089 | -0.80 | 0.51 | -0.46 | 0.10 | -0.93 | 0.59 |
| **Predator (AS)** | 0.41 | 0.081 | -0.31 | 0.093 | 0.07 | 0.079 | -0.75 | 0.11 |
| **Top predator (AS)** | -0.22 | 0.089 | -0.12 | 0.35 | -0.46 | 0.10 | -0.67 | 0.45 |
| **Overall Model Performance** | | | | | | | | |
| **RMSE** | 40 | | 132 | | 40 | | 147 | |
| **NRMSE** | 0.36 | | 0.35 | | 0.35 | | 0.39 | |
| **$R^2_{Pearson}$** | 0.61 | | 0.86 | | 0.24 | | 0.94 | |
| **$R^2_{Residual}$** | <0 | | <0 | | <0 | | <0 | |
| **RMSE (AS)** | 40 | | 22.8 | | 27 | | 108 | |
| **NRMSE (AS)** | 0.36 | | 0.061 | | 0.26 | | 0.29 | |
| **$R^2_{Pearson}$ (AS)** | 0.61 | | >0.99 | | 0.24 | | >0.99 | |
| **$R^2_{Residual}$ (AS)** | <0 | | 0.96 | | <0 | | <0 | |

**2 Direct answers on reviewer comments**

**Reviewer Comment**

Since the primary objective of the study is to model Hg bioaccumulation, I recommend that the model evaluation be presented as part of the Results and Discussion rather than the Methods. This change would strengthen the narrative and reduce redundancy—many of the points currently discussed in Section 3 could be streamlined. I suggest restructuring Section 3.1 to serve as the model evaluation, followed by subsequent sections explaining key discrepancies between model output and observations (currently in Section 4).

**Author Response**

I agree and I will give the manuscript a restructuring of the result sections. As you mentioned, the model evaluation should be seen as results. We will expand the evaluation to expand on the evaluation of the Hg cycling and pelagic bioaccumulation, a suggested edit is presented below, as the manuscript was very low on the evaluation of pelagic Hg cycling and bioaccumulation. Then I would merge the model and result section together as section 3. After this we will then discuss the results of how the model and literature study agree or disagree.

The benefit of doing this is that we can than state the AS model is better, and therefore focus the rest of the paper on using this model.

1. **Introduction**

2. **Methods**

   (a) The model (Current Sections 2.2–2.10)

   (b) Explain the model evaluation using all data (Current Sections 2.1 and 2.11)

   (c) Explain the Evaluation of the model conclusion using a single dataset (based on the use of the McClelland et al. (2024) dataset)

3. **Results**

   (a) Model evaluation: Hg cycling and pelagic bioaccumulation (described below)

   (b) Discuss patterns that occur in the model (Current Sections 3.1–3.4)

   (c) Evaluate the observed patterns using all data (Current Sections 4 and 5)

   (d) Evaluation of the observed patterns using a single dataset (based on McClelland et al. (2024), as described above)

4. Discuss the observed rol of the feeding strategy based on the results (Discussed below).

5. **Model limitations** (Current Section 6)

6. **Future work**: Further improvement of this model (Discussed below)

7. **Summary and conclusion (Current section 7)**

[Figure]

Figure 4: Modeled bioconcentration and biomagnification of iHg and MeHg. Partitioning to detritus and DOM is colored as bioconcentration. The y-axis is cut to show the high and low values. Notably is the high iHg to mgC ratio of detritus and DOM, leading to elevated iHg in suspension feeders. Additionally, higher trophic level animals have higher biomagnified MeHg.

**Author Response**

We will also update Fig. 6 to show the bioaccumulation in the AS model, as shown in Fig. 4. This because this is the better performing version of the model and we can therefore better anlyse this version.

**Author Response**

I would also add this section ad the end of the model evaluation segment

**Suggested edit**

**2.1 Model Limitations and Potential Improvements**

Evaluation of the GOTM-MERCY-ECOSMO setup reveals performance strengths and weaknesses of the model. There is a strong correlation between the modeled and observed effect of the feeding strategy on the bioaccumulation of MeHg in both the Northern and Southern North Sea setups. However, the Southern North Sea setup outperforms the Northern setup in both pelagic Hg cycling and megabenthos bioaccumulation assessments. In the Northern North Sea setup, the mesozooplankton tHg levels are higher than expected, although this could not be properly validated due to a lack of measurements. What is more notable is our model results show a reduction in MeHg bioaccumulation in the Northern North Sea compared to the Southern North Sea, while this is not present in the observations presented by McClelland et al. (2024). Since the Northern North Sea does have a good agreement with observations in the correlation between the feeding strategy and the bioaccumulation of MeHg, but the overall model performance is lower due to lower than expected MeHg concentrations in all benthic groups, it is likely that the model underestimates the flux of MeHg from the pelagic to the benthic system. In the

Southern North Sea setup, macrobenthos can feed directly on the phytoplankton bloom which results in a major exchange of organic material and MeHg between the benthic and pelagic food webs. In the Northern North Sea this is not possible, and the consumption of detritus, either by filtering it from the water or feeding on it from the sediment, is the main food source for megabenthos. The higher performance of the model in the Southern North Sea compared to the Northern North Sea indicates that this model could be further improved by enhancing the MeHg dynamics in deeper water. Further model evaluation is needed to determine whether the observed underperformance is due to limitations in the representation of sediment Hg chemistry, pelagic Hg speciation in deep waters, interactions between megabenthos, detritus, and MeHg, or a combination of these factors. A potential improvement here would be to implement a variable partioning coefficient of iHg and MeHg to detritus to account for the particle-concentration effect as is discussed by Coquery and Cossa (1995) for the Hg in the North Sea.

**Author Response**

I will add this to the beginning of the model evaluation section to show the Hg cycling and bioaccumulation in the pelagic is in line with observations.

**Suggested edit**

**2.2 Evaluation of the Hg cycling and pelagic bioaccumulation**

The marine cycling and speciation of Hg, in addition to the bioaccumulation in phytoplankton and zooplankton, is an essential driver of the bioaccumulation of iHg and MeHg in the benthic food web. Observed and modelled dissolved tHg concentration, the percentage of tHg that is MeHg, and the Hg content of phytoplankton and zooplankton is shown in Table 5. The concentration of dissolved tHg and the percentage of MeHg of dissolved tHg are compared to observations by Coquery and Cossa (1995), while the bioaccumulation of tHg in phytoplankton and zooplankton is compared to observations by Nfon et al. (2009). It must be noted that the observations by Nfon et al. (2009) are not from the North Sea itself, but from the better-studied nearby Baltic Sea. The average dissolved tHg concentration is 1.7 and 2.1,pM in the Northern and Southern North Sea, respectively. This is well within 1 standard deviation of the $1.7\pm0.7$ pM observed by Coquery and Cossa (1995). The MeHg concentration was observed to be between 0.5 and 4.3% of tHg, with an average of 3% in the North Sea. The percentage MeHg in our model is 2.3% and 2.0% on average, which falls well within that range.

For bioaccumulation, we could not find separate reliable measurements of MeHg and iHg in phytoplankton and zooplankton in the North Sea, and we therefore evaluated the tHg content. This was measured in diatoms to be $10\pm5$ ng Hg mg$^{-1}$. This means that the mean bioaccumulation in our model in diatoms is lower, with 5.8 ng Hg mg$^{-1}$ and 9.0 ng Hg mg$^{-1}$ in the Northern and Southern North Sea, respectively, but still within 1 standard deviation of the measurements. Observations labeled as zooplankton and mysis were compared to our modeled microzooplankton and mesozooplankton, respectively. All modeled values fall within 1 standard deviation of the observed tHg concentration, with one exception: mesozooplankton in the Northern North Sea, which is 13.5% more than 1 standard deviation above the observations. This is mostly driven by a high iHg content, as the MeHg content is similar in microzooplankton and mesozooplankton.

This similarity in MeHg is caused because, even though mesozooplankton have a higher trophic level, they prefer to feed on larger diatoms which have less MeHg than smaller flagellates, which are preferred by microzooplankton. The high iHg content, especially

in the Northern North Sea, is caused by the consumption of detritus by zooplankton in the model. While there is a shortage of data on bioaccumulation at the base of the food web, especially in the North Sea, which complicates model evaluation, the dissolved tHg concentration, the percentage of MeHg, and the tHg content of phytoplankton and zooplankton agree well with observations. With the exception of the 13.5% elevated tHg content in Northern North Sea mesozooplankton, all modeled values fall within 1 standard deviation of the observations. Because of this, we conclude that the model replicates marine Hg cycling and bioaccumulation at the base of the food web in line with observations, with the caveat that we do not have measurements of zooplankton in the Northern North Sea to verify or reject the elevated levels in that setup.

Table 5: Dissolved total Hg (pM), MeHg (% of tHg), and Hg concentrations in biota (ng Hg mg$^{-1}$ d.w.) across North Sea regions.

|  | Observed | NNS | SNS |
|---|---|---|---|
| tHg$_{dissolved}$ (pM) | $1.7 \pm 0.7$ | $1.7 \pm 0.26$ | $2.0 \pm 0.28$ |
| MeHg (% of tHg) | 3 (0.5–4.3) | $2.3 \pm 0.23$ | $2.0 \pm 0.31$ |
| Diatoms (ng Hg mg$^{-1}$) | $10 \pm 5$ | $7.0 \pm 1.1$ | $8.3 \pm 1.6$ |
| Flagellates (ng Hg mg$^{-1}$) |  | $13.9 \pm 3.0$ | $14.3 \pm 3.0$ |
| Microzooplankton tHg (ng Hg mg$^{-1}$) | $37.5 \pm 31.3$ | $67.4 \pm 29.3$ | $40.3 \pm 11.4$ |
| Microzooplankton MeHg (ng Hg mg$^{-1}$) |  | $7.1 \pm 2.1$ | $10.5 \pm 2.7$ |
| Mesozooplankton tHg (ng Hg mg$^{-1}$) | $62.5 \pm 12.5$ | $86.7 \pm 15.1$ | $72.3 \pm 19.6$ |
| Mesozooplankton MeHg (ng Hg mg$^{-1}$) |  | $6.9 \pm 2.6$ | $10.5 \pm 1.7$ |

**Author Response**

I would suggest to add the below expansion of the discussion about what drives the role in feeding strategy.

**Suggested edit**

Combining the results of the model and the literature studies is difficult due to the high uncertainty in most drivers, including the organic material content of dry weight and result should be viewed with skepticism. The data used in this study was prepared by McClelland et al. (2024) to mimic consumption by predators: for small arthropods, their skin was not removed, but for gastropods and bivalves, the shell was not taken into account for the weight as predators would typically not eat this. The concentration of MeHg per unit energy is arguably the key measure in bioaccumulation. Predators need to ingest a specific energy amount, so if a prey is composed of half organic material and half non-organic components, such as shell, its MeHg content per dry weight is halved. However, predators would consume double the dry weight to obtain the energy, and thus the same MeHg. In general, the energy appears to be consistent with Ash Free Dry Weight (AFDW), as such ideally we would normalize all measurement of MeHg bioaccumulation per AFDW (Weil et al., 2019).

Unfortunately, doing this conversion reliably on published data is not possible as AFDW varies with the age and body size of animals, which information is not generally available (Eklöf et al., 2017). That being said, we find it convincing that both our model results, the literature study in which we aggregate all measurements, and the literature study where we take samples from a single study all suggest that filter feeders have elevated MeHg.

The difference in the uptake of MeHg and organic carbon is also the main driver in our model of the difference caused between the bioaccumulation in filter and deposit feeders. In the model this is mostly driven by the lower feeding efficiency of MeHg presented by (Dutton & Fisher, 2012). While the lower assimilation efficiency of MeHg from sediment is understudied, it is supported by the limited available literature. In addition to the direct work by Dutton and Fisher (2012) analysing transfer efficiency of MeHg in the sediment to biota, it is also demonstrated that the concentration of MeHg in the sediment correlates poorly with the MeHg concentration in all biota, except infauna (Buckman et al., 2019; Chen et al., 2009). If MeHg would easily be transferred from the sediment to deposit feeders, there would be a much stronger correlation between sediment and deposit feeder MeHg concentrations.

The elevated iHg levels in suspension feeders are not related to this. There is no literature on the assimilation efficiency of MeHg when DOM is consumed, but the parameter is not altered in our model compared to when other food is consumed. The increased level of iHg in our model must, therefore, be caused by another driver. In our model, this is because the partitioning of iHg to DOM is stronger than that to detritus. As mentioned earlier, the partitioning of iHg to detritus and DOM is based on the $K_{ow}$ which has the values of log10(6.4) and log10(6.6) for detritus and DOM respectively. So while biota bioaccumulates MeHg stronger than iHg, DOM has a stronger binding to iHg than to MeHg. Since iHg is generally much more common (about 98% of tHg in our model) this results in a much higher bioaccumulation of iHg than MeHg. This aligns with the observations by Orani et al. (2020), which found MeHg to be only between 1% and 29% in DOM consuming sponges.

**Author Response**

Additionally I would add the following component to the concluding remarks

**Suggested edit**

Filter feeders and molluscs typically accumulate more MeHg than other organisms at similar trophic levels. This pattern is consistent not only in our models but also in available data. This raises a hypothesis that expanding bivalve populations, as seen in mussel or oyster farming, might affect MeHg bioaccumulation in higher trophic levels. This is supported by the observations that fish in lakes invaded by zebra mussels have higher Hg levels than fish in lakes without zebra mussels Blinick et al. (2024). However, such ecological alterations also impact other bioaccumulation factors like biomass distribution and trophic interactions. While our findings support the role of filter feeders and molluscs in MeHg dynamics and higher bioaccumulation in top predators, the complexity of ecological situations requires further case specific studies to understand if and when bivalve communities lead to increased MeHg transfer.

Modeling studies can help our understanding of the factors influencing MeHg bioaccumulation, but ability to accurately predicht MeHg bioaccumulations needs to be carefully validated. Our findings reveal that filter-feeding molluscs and DOM-utilizing suspension feeders have different Hg bioaccumulation patterns compared to other megabenthos. Modeling bivalve aquaculture or DOM-consuming suspension feeders can help explore their potential role in altering MeHg bioaccumulation. Understanding how functional traits like feeding strategy influence MeHg transfer remains key to improving both predictive models and environmental risk assessments.

**Reviewer Comment**

The purpose of Figure 3 is unclear. It is not evident why the authors chose to use Hg data from different ecosystems and plot them against trophic level (referred to as feeding strategy in the figure). Since ecosystems differ in baseline inorganic Hg and MeHg concentrations, the MeHg–trophic level relationship should be examined within each ecosystem independently.

**Author Response**

The reason we showed this was to be able to compare the relationship between Trophic Level and bioaccumulation between the base model and the AS model with the relationship in observations. While the original intention was the show that while MeHg bioaccumulates, iHg does not, I agree that this is thus standard accepted that showing this is not necessary. Because the figures does allow to quickly compare the slope of the bioaccumulation with the model I would put the figure in the Supplementary information. Here an interested reader can see how the slope of the bioaccumulation of MeHg and iHg compare to the global dataset while not making it a core part of the paper. Instead we performed the ecosystem specific analyses based on the McClelland et al. (2024) dataset of which Fig. 1 will be shown in the paper.

**Reviewer Comment**

For model evaluation, I strongly suggest plotting modeled versus observed concentrations of speciated Hg (inorganic and MeHg) for each modeled feeding strategy. This would provide a clearer and more direct assessment of model performance.

[Figure]

Figure 5: Mean bioaccumulation of MeHg in both model and observations is shown, with error bars representing 1 SE. The model's predictions do not consistently match observations within 1 SE, yet they display a comparable trend: Top Predators have the highest MeHg levels, followed by predators, with generalists and filter feeders showing similar MeHg, which are higher than those found in deposit feeders. Both model and observations show that suspension feeders have the lowest MeHg levels among the feeding strategies.

**Author Response**

We will do this by making a barplot for both MeHg and iHg with a broken axis for MeHg so we can show both higher and lower trophic levels. I would also suggest to only do this for the MeHg so it can show both lower and higher trophic levels. We will show Fig. 5 for MeHg and Fig. 6 for iHg. Based on your comment another suggestion would be to show it as is shown in Fig. 7, but I think the bar plots give the same information in this case in a way that is very easy to undestand for readers, even readers that are not used to seeing model data.

[Figure]

Figure 6: Mean bioaccumulation of iHg in both the model and observations. Error bars represent 1 SE. The model accurately shows increased iHg levels in suspension feeders. However, for all other feeding categories, except top predators, the iHg content is overestimated. In top predators, observed iHg levels are higher and not reflected in the model, though these elevated iHg levels have a very high SE.

[Figure]

Figure 7: Direct plot of modeled and obseved MeHg (a) and iHg (b).

**Reviewer Comment**

Section 3.2.3, which addresses the effect of feeding strategy on bioaccumulation, is central to the manuscript's aims, yet it is not discussed in sufficient depth. In contrast, the manuscript devotes substantial space to explaining Hg vs. trophic level patterns (Section 3.2.4), which are already well-established in the literature. I recommend condensing the discussion in 3.2.4 and focusing more on how feeding strategies influence MeHg and inorganic Hg transfer, particularly in benthic food webs.

**Author Response**

I addition to the suggestion to restructure made above, I would further adress this comment by merging section 3.2.4 about the effect of trophic level and 3.2.5 about the allometric scaling. The whole reason why this model performs better is because it better captures the trophic dynamics. These dynamcis are, as you mention already well understood and thus not a novel result of this model. This combined with the removal of Fig. 3 would shorten the part of this paper dedicated to this. Combined with the above discussed expansion of the drivers behind the importance of feeding strategy

**Reviewer Comment**

Figure 4 is difficult to interpret. It is unclear whether the data are empirical or simulated. A more straightforward approach might be to present Hg concentrations across feeding strategies as a bar chart with error bars. If the intent is to show correlations between feeding strategies, a correlation coefficient would be more appropriate.

**Author Response**

I will remove this plot. The bar plots that you suggested convay the same message in a clearer way, that there are differences in iHg and MeHg bioaccumulation. Combining this with a plot that compares it to the model saves space and a reducent figure.

**Reviewer Comment**

Section 3.3, on allometric scaling, should appear earlier in the manuscript. When reading Sections 3.2.3 and 3.2.4, I repeatedly found myself wondering about the effects of allometric scaling on the results. Figures 7 and 8 could be consolidated to allow readers to compare model performance with and without allometric scaling more clearly.

**Author Response**

I agree and I hope the suggested restructering would improve the manuscript in your opinion. Here I will show Fig. 7 and Fig. 8 in the first part of the Result section. Then the difference between the 2 models can be seen. This can be compared in to observations in the third part of the result section. This would allow for a good comparison between the previous version of the model and the AS model, and an easy comparison to field observations that the AS model is an improvement in higher trophic leel compared to the previous version of the model. I will update the images as shown belowin Fig. 8

[Figure]

Figure 8: The inluence of trophic level on the bioaccumulation of MeHg, iHg, and tHg in both the AS (panels a, c, e) and the base model (panels b, d, f). In the AS model, the relationship with trophic level is stronger, where ln(MeHg)=1.24TL-0.03, compared to the base model, which is ln(MeHg)=0.64TL+1.42. TL represents trophic level, and MeHg is expressed in ng Hg g$^{-1}$ d.w. For iHg, the bioaccumulation patterns are nearly identical, with ln(MeHg)=-0.19TL+5.11 for the AS model and ln(MeHg)=-0.18TL+5.11 for the base model, both showing a weak inverse correlation with trophic level, largely due to higher iHg levels in low trophic level feeders. In terms of tHg, there is a higher increase in bioaccumulation in the AS model (ln(MeHg)=0.43TL+3.76) compared to the base model (ln(MeHg)=0.04TL+4.175), driven by the stronger association between MeHg and trophic level in the AS model.

**Reviewer Comment**

Lines 340–350: This content would be better integrated into the allometric scaling section.

**Author Response**

Agreed, I will move this to the evaluation (section 3(a) as suggested above.)

**Reviewer Comment**

As the authors note, the model is implemented for the North Sea, yet many of the empirical datasets used for comparison originate from other regions. This mismatch raises concerns about the validity of the model evaluation. Comparing model output to observations from ecologically distinct systems—each with different baseline Hg and MeHg levels, food web structures, and biogeochemical conditions—complicates interpretation and undermines the credibility of the evaluation. I strongly recommend either (1) limiting the model evaluation to observed data from the North Sea, or (2) running separate models parameterized for the specific ecosystems from which the empirical data are drawn.

**Author Response**

I agree with this concern of data being drawn from different locations. Unfortunately, there is not enough data from the North Sea to evaluate the model purely on data from the North Sea. I think the biggest concern is that there might be a cocorrelation between certain areas with high MeHg and the feeding strategies that are commen there. I hope that involving the evaluation component purely based on the McClelland et al. (2024) data adresses this concern to some degree. The model is not purely designed to be a predictive model for different megabenthos species in the North Sea, rather it is aimed to analyse if feeding strategy is a significant driver of MeHg bioaccumulation. The aim is to show that running a model in an idealized 1D setup resembling coastal conditions results in differences in the bioaccumulation caused by feeding strategies. Notably, our model shows higher MeHg in filter feeders compared to deposit feeders and extremely elevated levels of iHg in suspension feeders consuming DOM. We find it convincing the same pattern occurs in our model, the global dataset and the McClelland et al. (2024) dataset, but I fully agree that better measurements are necessary to validate models before they can be used in a predictive capacity. Our hope is that this manuscripts helps the message that studies to measrurements of biaoccumulation in the benthic food web can be relevant to increase our understandig of bioaccumulation in higher trophic levels.

**References**

Blinick, N. S., Link, D., Ahrenstorff, T. D., Bethke, B. J., Fleishman, A. B., Janssen, S. E., Krabbenhoft, D. P., Nelson, J. K. R., Rantala, H. M., Rude, C. L., & Hansen, G. J. A. (2024). Increased mercury concentrations in walleye and yellow perch in lakes invaded by zebra mussels.

Buckman, K. L., Seelen, E. A., Mason, R. P., Balcom, P., Taylor, V. F., Ward, J. E., & Chen, C. Y. (2019). Sediment organic carbon and temperature effects on methylmercury concentration: a mesocosm experiment. *The Science of the total environment*, *666*, 1316.

Chen, C. Y., Dionne, M., Mayes, B. M., Ward, M., Darron, Sturup Stefan, & Brian, J. P. (2009). Mercury Bioavailability and Bioaccumulation in Estuarine Food Webs in the Gulf of Maine. *Environ. Sci. Technol.*, *43*, 1804–1810.

Coquery, M., & Cossa, D. (1995). Mercury speciation in surface waters of the north sea. *Netherlands Journal of Sea Research*, *34*(4), 245–257.

Dutton, J., & Fisher, N. S. (2012). Bioavailability of sediment-bound and algal metalsto killifish Fundulus heteroclitus. *Aquatic biology*, *16*, 85–96.

Eklöf, J., Austin, Å., Bergström, U., Donadi, S., Eriksson, B. D., Hansen, J., & Sundblad, G. (2017). Size matters: Relationships between body size and body mass of common coastal, aquatic invertebrates in the Baltic Sea. *PeerJ*, *2017*(1), e2906.

McClelland, C., Chételat, J., Conlan, K., Aitken, A., Forbes, M. R., & Majewski, A. (2024). Methylmercury dietary pathways and bioaccumulation in Arctic benthic invertebrates of the Beaufort Sea. *Arctic Science*, *10*(2), 305–320.

Nfon, E., Cousins, I. T., Järvinen, O., Mukherjee, A. B., Verta, M., & Broman, D. (2009). Trophodynamics of mercury and other trace elements in a pelagic food chain from the Baltic Sea.

Orani, A. M., Vassileva, E., Azemard, S., & Thomas, O. P. (2020). Comparative study on Hg bioaccumulation and biotransformation in Mediterranean and Atlantic sponge species. *Chemosphere*, *260*, 127515.

Weil, J., Trudel, M., Tucker, S., Brodeur, R. D., & Juanes, F. (2019). Percent ash-free dry weight as a robust method to estimate energy density across taxa. *Ecology and Evolution*, *9*(23), 13244–13254.

---

## Author Comment (AC2)

**1 General answer**

**Author Response**

Thank you for your great comments. Before I go into the individual comments in detail, I want to acknowledge a limitation in the paper that you flagged in several of your comments. This is related to the evaluation of the model output and the problem that the combining of data from various locations brings. This was done because, although data on Hg bioaccumulation can be found more often, studies measuring both MeHg and Trophic Level are less common. That being said, two big studies came out in 2024 that after I did the original data analyzes and I can include these papers. Although 2 papers sounds small, they have a lot of samples and drastically improve the sample size. Especially the paper by McClelland et al. (2024) can really supplement the paper. They sampled 476 benthic animals from two locations in the canadian Arctic. I would suggest that in addition to pooling all the data to see for general patterns, I use these data to verify if the feeding strategy also plays a significant role within the same dataset. I would add these additional analyses on several locations in the manuscript, so I will first describe my suggested addition here as I believe it is relevant for several suggestions you made.

**Suggested edit**

**Evaluating the role of feeding strategy using a single large dataset**

A third test of our hypothesis using field data from a single study to see if the same dynamics as occured in our model are present in these observations. We used MeHg bioaccumulation and trophic level data from 476 individuals across 53 taxa of benthic invertebrates as published by McClelland et al. (2024). These data were selected as it is the laregest study we could find with both trophic level and MeHg concentrations. When several animals of the same group where sampled, the dataset presents mean values per group per location, which we use as one datapoint in our analyses. Although feeding strategies in the dataset were broadly aligned with our classifications, we reassigned them to match the functional groups in our model. For example, shrimps were categorized as generalist feeders, which group is not present in McClelland et al. (2024), and isopods, which can be small benthic predators, were labeled as deposit feeders because their prey type is not represented in our model.

The data is sampled from two locations in the Canadian Arctic, Cape Bathurst (CB), which has a depth of 22 m and is located at $70°41'42.79''$ N, $128°50'21.34''$ W, and the eastern coast of Herschel Island in the Mackenzie Trough (MT), which has a depth of 116 m and is located at $69°36'44.96''$ N, $138°33'45.25''$ W. It must be noted this dataset is selected as it is extensive, but the region does have notable differences to the North Sea, where our model is run. It has extensive ice cover in winter, it is colder and is geographically distant from the model location. It does however, provide us with an opportunity to test if our model conclusions can be verified using field observations from a single study.

**Author Response**

I will add this to the methods section at line 218

**Suggested edit**

**Evaluation of the model using a single dataset**

To isolate the effect of the feeding strategy on MeHg bioaccumulation, we first transformed MeHg concentrations to their natural logartitm and fit a linear model with trophic level as predictor using the base R `lm()` function. The significance of the deviation from the predicted MeHg concentration at the trophic level was assessed using a one-sample $t$ tests. To improve interpretability, we calculated the percentage differences using $\text{Percentage difference} = 100 \times \left( \frac{\text{MeHg}_{\text{obs}}}{\text{MeHg}_{\text{pred}}} - 1 \right)$ based on the residuals of the linear fit. This is visualized on a bar graph showing the percentage difference in MeHg concentration caused by the feeding strategy. The errors bars represent the 1 Standard Error (SE). The same analysis was then performed to estimate differences in MeHg bioaccumulation related to phylum.

As a final test, linear models were fitted on the natural logarithm of bioaccumulated MeHg concentrations using trophic level, phylum, and feeding strategy as predictor variables (using the `lm()` function in R). Estimated marginal means (EMM) for each feeding strategy were calculated with the `emmeans()` function of the emmeans package and compared against the overall mean to assess deviations. This analysis was also performed separately for the MT and CB locations to verify the consistency of the effects of the feeding strategy. The EMMs where transformed to a percentage difference with the earlier used equation and the estimated percentage difference due to feeding strategy and its signifance is shown.

**Author Response**

I will add this to the result and discussion about the literature study result section (3.1). At line 351.

**Suggested edit**

The fit of the linear model against the natural logarithm of the bioaccumulated MeHg is shown in Fig. 1. The $R^2$ is similar with 0.43 and 0.45 in the CB and MT respectively, while the bioaccumulation is a bit lower in the CB (log(MeHg$_{\text{BA}}$)=0.137+1.14*TL) compared to that in the MT (log(MeHg$_{\text{BA}}$)=0.256+1.39*TL), where MeHg$_{\text{BA}}$ is the bioaccumulated MeHg in ng Hg mg$^{-1}$ d.w. and TL is the trophic level. The influence of the feeding strategy on MeHg bioaccumulation based on the results of McClelland et al. (2024) is shown in Table 1. While the only significant effect is deposit feeders in the MT having less MeHg than would be expected on their trophic positions, some other effects are consistent, albeit not significant in both locations. The strongest effect is that filter feeders have consistently higher MeHg (residuals are 0.7 in the CB and 0.8 in the MT), while deposit feeders have lower MeHg (residuals are -0.2 in the CB and -0.5 in the MT). The results of the same analyses for phyla are shown in Table 2. Here we see two consistent significant effects. Molluscs have elevated MeHg levels (residuals are 0.61 in the CB and 0.51 in the MT) while arthropods have reduced MeHg values (residuals are -0.35 in the CB and -0.30 in the MT). The percentage difference in MeHg bioaccumulation per feeding strategy is visualised in Fig. 2 and per phyla in Fig. 3. The average percentage difference between observed and the expectation based on trophic level is 102% and 128% in the CB and MT respectively for filter feeders while deposit feeders have 19 and 37% less MeHg than would be predicted based on trophic level alone in the CB and MT respectively. In the the analysis per phylum we see that molluscs have highly elevated MeHg levels with an increase of 66%

and 85% respectively in the CB and MT. The largest reduction in observed MeHg compared to the predicted values based on trophic level is in arthropods, here there is a decrease compared to the predicted values of 29% and 26% in the CB and MT respectively.

The results of the final analyses are shown in Table 3. Despite the lower sample size, which reduces statistical power, the results indicate that filter feeders consistently have higher MeHg levels than predicted based on their trophic position and phyla, while deposit feeders tend to have lower MeHg concentrations. These results are stronger in the MT with a change of 118% and -40% in filter and deposit feeders respectively than in the CB with a change of 7.2% and -14.8% in filter and deposit feeders respectively. It must be stated that this final analysis is included to address potential concern between the co-correlation of phyla and feeding strategy, but the problem of the reduced sample size has to be addressed. In the CB, where the increase in MeHg in filter feeders is rather low after correcting for both trophic level and feeding strategy, there are only three filter feeders, which are molluscs, and they make up 3/5 mollusc samples in this location, meaning that results should be seen with skepticism as filter feeders and molluscs have too much overlap. On the other hand, in the MT, there are five filter feeders from multiple phyla (Mollusca and Echinodermata) and the effect is considerably stronger with filter feeders having 118% more MeHg than would be expected based on their trophic level and phyla.

**Author Response**

I will add the following component to the discussion to evaluate the uncertainty of these results.

**Suggested edit**

Combining the results of the model and the literature studies is difficult due to the high uncertainty in most drivers, including the organic material content of dry weight and result should be viewed with skepticism. The data used in this study was prepared by McClelland et al. (2024) to mimic consumption by predators: for small arthropods, their skin was not removed, but for gastropods and bivalves, the shell was not taken into account for the weight as predators would typically not eat this. The concentration of MeHg per unit energy is arguably the key measure in bioaccumulation. Predators need to ingest a specific energy amount, so if a prey is composed of half organic material and half non-organic components, such as shell, its MeHg content per dry weight is halved. However, predators would consume double the dry weight to obtain the energy, and thus the same MeHg. In general, the energy appears to be consistent with Ash Free Dry Weight (AFDW), as such ideally we would normalize all measurement of MeHg bioaccumulation per AFDW (Weil et al., 2019).
Unfortunately, doing this conversion reliably on published data is not possible as AFDW varies with the age and body size of animals, which information is not generally available (Eklöf et al., 2017). That being said, we find it convincing that both our model results, the literature study in which we aggregate all measurements, and the literature study where we take samples from a single study all suggest that filter feeders have elevated MeHg.

The difference in the uptake of MeHg and organic carbon is also the main driver in our model of the difference caused between the bioaccumulation in filter and deposit feeders. In the model this is mostly driven by the lower feeding efficiency of MeHg presented by (Dutton & Fisher, 2012). While the lower assimilation efficiency of MeHg from sediment is

[Figure]

Figure 1: The linear fitted model between the natural logarithm of the bioaccumulated MeHg in ng Hg mg$^{-1}$ d.w. and the Trophic Level in the data presented by McClelland et al. (2024). For extra clarity the different Phyla shown with different colors while the different feeding strategies are marked with different symbols. In both the CB and MT setups there positive relationship between trophic level and the bioaccumulation of MeHg, but $R^2$ is only 0.43 and 0.45 in the CB and MT respectively, so it does not explain the full variation in bioaccumulation.

understudied, it is supported by the limited available literature. In addition to the direct work by Dutton and Fisher (2012) analysing transfer efficiency of MeHg in the sediment to biota, it is also demonstrated that the concentration of MeHg in the sediment correlates poorly with the MeHg concentration in all biota, except infauna (Buckman et al., 2019; Chen et al., 2009). If MeHg would easily be transferred from the sediment to deposit feeders, there would be a much stronger correlation between sediment and deposit feeder MeHg concentrations.

More work is needed to analyse if molluscs also have elevated MeHg levels, especially when normalised to AFDW. While molluscs do appear to have more MeHg in our analyses, it is possible that the increase in MeHg in molluscs compared to other animals of the same trophic level is caused by a combination of co-correlation with the filter feeding strategy and an increase in AFDW percentage of dry weight in the mollusc samples as the weight of the shell was not taken into account. Focusing Hg monitoring on using a holistic approach in which ideally tHg, MeHg, trophic level and AFDW are sampled in fully identified species would help clarify whether bioaccumulation patterns associated with feeding strategy and phylum are robust and relevant for Hg risk management.

[Figure]

Figure 2: Percentage difference from trophic level predicted MeHg concentrations by feeding strategy, with error bars showing ±1 SE. In both CB and MT regions, filter feeders have elevated MeHg levels relative to trophic level based expectations, while deposit feeders are reduced. Predators display higher MeHg than predicted, though the effect is smaller than in filter feeders; in CB, this increase does not exceed one SE. Generalist feeders have a slight reduction compared to expectations, but this is well within one SE, and were not present in CB for cross-region comparison.

[Figure]

Figure 3: Percentage difference from the predicted MeHg bioaccumulated based on trophic level per phyla, the error bars represent ±1 SE. The notable phyla are Mollusca and Arthropoda, while Mollusca have a notable increase in MeHg bioaccumulation compared to the prediction of 85% and 66% respectively in the CB and MT, there is a reduction of 26% and 29% in Arthropoda in the CB and MT respectively. Annelida are inconsistent with an increase in the CB and decrease in the MT compared to the predictions. Echinodermata have a mean reduction compared to the prediction in both the CB and the MT, but the SE is much larger than the mean effect.

Table 1: Mean residuals (±SE) of log(MeHg) by feeding strategy and region, after trophic level correction. Significant deviations (p < 0.05) are marked with *.

| Region | Feeding Strategy | $n$ | Mean Residual ± SE | p-value |
|--------|------------------|-----|--------------------|---------|
| CB | Deposit feeder | 16 | $-0.208 \pm 0.181$ | 0.268 |
| CB | Filter feeder | 3 | $0.704 \pm 0.286$ | 0.133 |
| CB | Predator | 6 | $0.203 \pm 0.331$ | 0.568 |
| MT | Deposit feeder | 15 | $-0.467 \pm 0.159$ | 0.011* |
| MT | Filter feeder | 5 | $0.824 \pm 0.395$ | 0.105 |
| MT | Generalist | 3 | $-0.143 \pm 0.319$ | 0.698 |
| MT | Predator | 12 | $0.277 \pm 0.226$ | 0.247 |

Table 2: Mean residuals (±SE) of log(MeHg) by phylum and region, after trophic level correction. Significant deviations (p < 0.05) are marked with *.

| Region | Phylum | $n$ | Mean Residual ± SE | p-value |
|--------|--------|-----|--------------------|---------|
| CB | Annelida | 6 | $0.229 \pm 0.424$ | 0.612 |
| CB | Arthropoda | 11 | $-0.349 \pm 0.137$ | 0.0294* |
| CB | Echinodermata | 3 | $-0.198 \pm 0.584$ | 0.767 |
| CB | Mollusca | 5 | $0.611 \pm 0.211$ | 0.0446* |
| MT | Annelida | 5 | $-0.405 \pm 0.377$ | 0.343 |
| MT | Arthropoda | 12 | $-0.304 \pm 0.111$ | 0.0196* |
| MT | Echinodermata | 5 | $-0.188 \pm 0.509$ | 0.730 |
| MT | Mollusca | 13 | $0.509 \pm 0.231$ | 0.0482* |

Table 3: The effect of feeding strategy on MeHg bioaccumulation per Region compared to the prediction accounting for both trophic level and feeding strategy. Significant (p < 0.05) is marked with *. There is still a consistent increase in filter feeders and a consistent decrease in deposit feeders. This is effect is larger in the MT with a relative percentage increase of 118% in filter feeders and a decrease of 40% in deposit feeders.

| Feeding Strategy | % Diff (MT) | p-value (MT) | % Diff (CB) | p-value (CB) |
|------------------|-------------|--------------|-------------|--------------|
| Deposit feeder | -40.0 | 0.034* | -14.8 | 0.888 |
| Filter feeder | 118.0 | 0.034* | 7.2 | 0.888 |
| Generalist | -25.9 | 0.563 | – | – |
| Predator | 3.0 | 0.895 | 9.4 | 0.888 |

**Author Response**

**Updated statistics due to extra data in the analyses**

In addition to the extra papers added, a final small change was made: the assimilation efficiency of the deposit feeder was set to 0.12, as this is more in line with the literature presented by Dutton and Fisher (2012). To clarify, this and the addition of two more papers, of course, change the statistics and also highlight your concern about different base levels of Hg. The new statistical analyses are presented in Table 4. Because the new data has a higher baseline of MeHg, it indeed changes (in this case reduces) the agreement between the model and the observations in absolute terms. It does, however, not mean that the correlation between feeding strategies and elevated MeHg levels is different, as this remains consistent. To increase transparency of model fit, I will both show the $R^2$ based on the Pearson correlation coefficient and the Residual-Based $R^2$.

**Suggested edit**

Finally, the goodness of fit of the model is shown both based on the Pearson correlation coefficient and based on the residuals. It is important to note the difference between the $R^2$ based on the Pearson correlation coefficient ($R^2_{\text{Pearson}}$) and the Residual-Based $R^2$ ($R^2_{\text{Residual}}$). They are defined as $R^2_{Pearson} = (\text{corr}(y_{\text{mod}}, y_{\text{obs}}))^2$ and the $R^2_{Residual} = 1 - \frac{\sum(y_{\text{obs}} - y_{\text{mod}})^2}{\sum(y_{\text{obs}} - \bar{y}_{\text{obs}})^2}$, where $y_{\text{obs}}$ are the mean observed values and $y_{\text{mod}}$ are the mean modeled value per feeding strategy. The $R^2_{\text{Pearson}}$ describes how well a linear relationship between our model and the data holds, thus, if this is high, the models explain the observed differences between feeding strategies well, but it is not effected by the absolute concentrations. This means that the $R^2_{\text{Pearson}}$ evaluates if our model accurately represents differences between feeding strategies, but it is not affected by a bias in the data due to mismatch between the baseline of MeHg bioaccumulation. The $R^2_{\text{Residual}}$ measures how well the modeled values match the observations, including absolute values, which is both affected by the ability of the model to predict the effect of feeding strategy on bioaccumulation, and by overal concentrations of iHg and MeHg in biota. Both $R^2_{\text{Pearson}}$ and $R^2_{\text{Residual}}$ give a value below 1. Closer to 1 means a better fit while below 0 means that the the model does not outperform estimating the data based on the mean.

**Suggested edit**

The $R^2_{\text{Pearson}}$ is high (>0.86) for MeHg in all setups and exceeds 0.99 in the AS model, indicating that the model captures the relative differences between feeding strategies well. For iHg, performance is lower, particularly in the Northern North Sea ($R^2_{\text{Pearson}}$ = 0.24). The ability to reproduce absolute bioaccumulation is more limited. Only the AS model in the Southern North Sea shows good agreement ($R^2_{\text{Residual}}$ = 0.96), while all other setups yield $R^2_{\text{Residual}}$ < 0, suggesting that using the mean of the observations would outperform the model.

This can be explained, as baseline MeHg levels vary between sampling regions. Notably, the AS model in the Southern North Sea performs well both in reproducing overall MeHg levels and in explaining variability across feeding strategies. Even when excluding predators and top predators, $R^2_{\text{Pearson}}$ remains high (0.80), suggesting that feeding strategy effects are captured across trophic levels and are not just driven by high MeHg levels in predatorial feeding strategies. In contrast, the Northern North Sea has a high $R^2_{\text{Pearson}}$ (=0.94) but low $R^2_{\text{Residual}}$ (<0) so it captures the effect of feeding strategies while not being able to resplicate absolute MeHg concentrations.

Table 4: Statistical analysis of model performance for iHg and MeHg levels by feeding strategy for Southern North Sea (SNS) and Northern North Sea (NNS).

| | SNS | | | | NNS | | | |
| | iHg | | MeHg | | iHg | | MeHg | |
| | N. Bias | BF10 | N. Bias | BF10 | N. Bias | BF10 | N. Bias | BF10 |
|---|---|---|---|---|---|---|---|---|
| **Suspension** | 0.18 | 0.11 | 0.85 | 0.011 | -0.18 | 0.11 | 0.17 | 0.0072 |
| **Filter** | 1.48 | 0.47 | -0.33 | 0.046 | 1.45 | 0.069 | -0.71 | 0.054 |
| **Deposit** | 1.01 | 0.078 | -0.44 | 0.046 | 0.34 | 0.034 | -0.77 | 0.052 |
| **Generalist** | 1.31 | 0.18 | -0.39 | 0.047 | 1.23 | 0.14 | -0.75 | 0.055 |
| **Predator** | 0.41 | 0.081 | -0.42 | 0.10 | 0.07 | 0.079 | -0.71 | 0.074 |
| **Top predator** | -0.22 | 0.089 | -0.80 | 0.51 | -0.46 | 0.10 | -0.93 | 0.59 |
| **Predator (AS)** | 0.41 | 0.081 | -0.31 | 0.093 | 0.07 | 0.079 | -0.75 | 0.11 |
| **Top predator (AS)** | -0.22 | 0.089 | -0.12 | 0.35 | -0.46 | 0.10 | -0.67 | 0.45 |
| **Overall Model Performance** | | | | | | | | |
| **RMSE** | 40 | | 132 | | 40 | | 147 | |
| **NRMSE** | 0.36 | | 0.35 | | 0.35 | | 0.39 | |
| **$R^2_{Pearson}$** | 0.61 | | 0.86 | | 0.24 | | 0.94 | |
| **$R^2_{Residual}$** | <0 | | <0 | | <0 | | <0 | |
| **RMSE (AS)** | 40 | | 22.8 | | 27 | | 108 | |
| **NRMSE (AS)** | 0.36 | | 0.061 | | 0.26 | | 0.29 | |
| **$R^2_{Pearson}$ (AS)** | 0.61 | | >0.99 | | 0.24 | | >0.99 | |
| **$R^2_{Residual}$ (AS)** | <0 | | 0.96 | | <0 | | <0 | |

**2  Specific Comments**

**Reviewer Comment**

Data on mercury concentrations in marine megabenthos were compiled and examined for differences in bioaccumulation by feeding strategy. It appears a relatively small number of studies were used (n = 12, Table 5) compared to available published data on mercury in marine megabenthos. What criteria were used for the literature review and selection of papers?

**Author Response**

The papers were selected with a focus on having both trophic level and MeHg concentrations estimated in marine megabenthos. These studies are indeed less common than studies sampling only metal concentration, including total Hg. While verifying new literature, I did find two new papers that were published last year that I would now include in the analyses. One of these papers has a very substantial dataset that allows us to verify our hypothesis purely based on this data at the beginning of this answer.

**Reviewer Comment**

More information on the measurement of mercury burdens in the megabenthos studies seems important to include for interpretation and standardization. Sometimes megabenthos tissues cannot be sampled consistently due to differences in exoskeleton and body form. What tissue types were measured for mercury? (e.g., whole body [including exoskeleton], internal viscera, muscle). How was inorganic mercury concentration determined? (i.e., the studies in Table 5 do not include inorganic mercury). Are the modelled concentrations for whole body of megabenthos (e.g., Figure 6)?

**Author Response**

The bioaccumulation in the model and the output in Fig. 6 is shown to be in ng Hg per mg carbon. In the model we tried to isolate the effect of the feeding strategy on bioaccumulation, so the conversion to dry weight assumes a 1:2 ratio. Most measurements indeed express bioaccumulation in dry weight and the carbon content of biota is seldom measured alongside Hg. The conversion from carbon to dry weight does indeed introduce uncertainty, but it is within reasonable levels. For example. the soft part of molluscs are for example found to have between 36% and 48% carbon per dry weight while the carbon content of artrophods (mysis mixta) was found to be 51.4% (Gorokhova & Hansson, 2000; Jurkiewicz-Karnkowska, 2005). The complications is that the study aims to look at the feeding strategy, which does very between phyla. Filter feeders can, for example, be either arthropods in the form of barnacles, molluscs in the form of bivalves, annelids in the form of of fan worms, or echinoderms in the form of brittle stars, because of this a standard 1:2 conversion ratio between carbon and dry weight was kept consistent over all feeding strategies. I will add these citations to the manuscript at the end of the model development section (line 152)

**Suggested edit**

Our model is resolved in carbon content, while measurements are often in dry weight. The carbon fraction of dry weight generally ranges from 0.4 to 0.6, but can vary between different taxa (Gorokhova & Hansson, 2000; Jurkiewicz-Karnkowska, 2005). To ensure

consistency across different functional groups with diverse feeding strategies, we maintain a 1:2 conversion ratio for carbon to dry weight for all megabenthos functional groups.

**Reviewer Comment**

The empirical mercury data for megabenthos were pooled across geographic locations where environmental mercury exposure may have differed. How were potential confounding effects of geographic variation and feeding strategy resolved? Were the findings of feeding strategy influence on mercury burdens consistent with individual studies from specific geographic areas?

**Author Response**

I agree that this is an issue. We tried to validate our model findings using data, but data is not sufficiently available for all geographic locations to robust individual analyses. In order to address your concerns, we added the analyses solely based on the data presented (McClelland et al., 2024). Based on this, we can see that the results of our studies are consistent between the model, by pooling all geopgraphical locations and by analysing two individual locations.

**Reviewer Comment**

Organism bioaccumulation is described as involving two key processes: bioconcentration and biomagnification (lines 26-34). A more nuanced discussion is suggested here on exposure pathways and also clarification on the mechanistic processes that were modelled. Uptake of aqueous inorganic mercury and methylmercury into the food web occurs via bioconcentration in primary producers. However, consumers are typically exposed to mercury primarily through their diet and not via bioconcentration from water.

**Author Response**

I will add the following expansion to section 2.5 on line 121. I would replace the part where I discuss the evaluation of the previous version of the model. The changes in megabenthos can have a feedback on Hg cycling so I added a new paragraph on the evaluation of pelagic Hg cycling and bioaccumulation later in the review. But then I can discuss the implementation of bioaccumulation in more detail here.

**Suggested edit**

Bioaccumulation is implemented to account for bioconcentration in all trophic levels and biomagnification in all consumers. Phytoplankton have a size-dependent uptake and release rate for the uptake and release of iHg. Based on observations by Pickhardt et al. (2006) which found higher MeHg in smaller phytoplankton but consistent iHg levels, phytoplankton have a size-dependent uptake rate and constant release rates. This means that diatoms and flagellates bioaccumulate similar amounts of iHg, while the smaller flagellates accumulate more MeHg. The uptake and release rates of iHg and MeHg in zooplankton are based on Tsui and Wang (2004) and on W. Wang and Wong (2003) for fish. An essential component of the ecosystem that interacts with bioaccumulation in megabenthos that was not overhauled for this study are the interactions between detritus and DOM and iHg and MeHg. The only $Hg^{2+}$ and $MMHg^+$ are assumed to partition to detritus and DOM, and this partitioning is assumed to be an equilibrium that is instantaneous and is reestimated on every time step. The equilibrium is based on the $K_{ow}$ values which are

based on Allison et al. (2005) and Tesán Onrubia et al. (2020). This value is log10(6.4) and log10(6.6) for the partitioning of $Hg^{2+}$ and log10(5.9) and log10(6.0) for the binding of $MMHg^+$ to detritus and DOM respectively. This is the same approach that is used and evaluated in Bieser et al. (2023) and Amptmeijer et al. (2025).

**Reviewer Comment**

Figure 6 shows modelled concentrations in biota, where bioconcentrated and biomagnified mercury are differentiated. These model results are not consistent with known trophic transfer processes of mercury. In higher trophic level biota, very little of the total mercury burden is inorganic mercury (e.g., in contrast with modelled result for a top predator) and most mercury is obtained from diet rather than water (bioconcentration). In Figure 6, much of the bioaccumulated mercury is attributed to bioconcentration. E.g., see Wang and Wang, Environmental Pollution 2019, Volume 252, Part B, September 2019, Pages 1561-1573, and other studies cited therein.

**Author Response**

This image stems from an admittedly awkward design choice in the paper. As stated in the paper, we aimed to enhance the model's performance over its previous version by modifying how iHg and MeHg are released during respiration. While some species have relatively high iHg concentrations, it is typically much higher and is better depicted in the Allometric Scaling (AS) model. The image presented is of the base model, which shows MeHg levels that, while in the observed range, are below average observations. Additionally, while some experimental studies such as presented by W. Wang and Wong (2003) find that bioconcentration can be a major exposure route in fish, I agree that it is typically expected to be smaller, as is the case in the AS model.

To improve this in the manuscript we will display the bioaccumulation per group per source for the AS model, as depicted in Fig. 4. This is the better performing model and focusing the visualization on the base model creates unnecessary ambiguity. In the AS model, top predators have 80-90% of tHg from MeHg from biomagnification, which is more in line with the observations by W. X. Wang and Tan (2019). Bioconcentration remains a significant route in the model, but based on W. Wang and Wong (2003) this is in line with obervations. I will move the image to the end of section 3.3 (line 301). So it can be discussed after the evaluation of the base setup and AS model is presented.

**Reviewer Comment**

The focus of this study is on megabenthos, i.e. consumer organisms. However, a key process that warrants more modelling attention is the process of methylmercury entry in the food web via primary producers. Figure 6 shows modelling results for diatoms and dinoflagellates. How do those mercury concentrations compare with empirical data for phytoplankton? How can the inorganic mercury and methylmercury in primary producers be a result of biomagnification, as indicated in the figure?

**Author Response**

About the labeling in Fig. 6, my apology that is indeed mislabeled. I will correct this as shwon in Fig. 4. I think the question surrounding the comparison with phytoplankton aligns with your question about the model validation in the technical comments section "Line 195. Provide more detail on how the model performance was evaluated.". Since

[Figure]

Figure 4: Modeled bioconcentration and biomagnification of iHg and MeHg. Partitioning to detritus and DOM is colored as bioconcentration. The y-axis is cut to show the high and low values. Notably is the high iHg to mgC ratio of detritus and DOM, leading to elevated iHg in suspension feeders. Additionally, higher trophic level animals have higher biomagnified MeHg.

the model is quite extensively evaluated for the bioaccumulation in megabenthos, I will add the following paragraph about the evaluation of the model performance in the pelagic before the evaluation of the bioaccumulation in benthos. So this would be section 2.1.1 on line 193, and the current section about the evaluation of megabenthos would be on line 2.1.2.

**Suggested edit**

**2.1 Evaluation of the Hg cycling and pelagic bioaccumulation**

The marine cycling and speciation of Hg, in addition to the bioaccumulation in phytoplankton and zooplankton, is an essential driver of the bioaccumulation of iHg and MeHg in the benthic food web. Observed and modelled dissolved tHg concentration, the percentage of tHg that is MeHg, and the Hg content of phytoplankton and zooplankton is shown in Table 5. The concentration of dissolved tHg and the percentage of MeHg of dissolved tHg are compared to observations by Coquery and Cossa (1995), while the bioaccumulation of tHg in phytoplankton and zooplankton is compared to observations by Nfon et al. (2009). It must be noted that the observations by Nfon et al. (2009) are not from the North Sea itself, but from the better-studied nearby Baltic Sea. The average dissolved tHg concentration is 1.7 and 2.1,pM in the Northern and Southern North Sea, respectively. This is well within 1 standard deviation of the 1.7±0.7 pM observed by Coquery and Cossa (1995). The MeHg concentration was observed to be between 0.5 and 4.3% of tHg, with an average of 3% in the North Sea. The percentage MeHg in our model is 2.3% and 2.0% on average, which falls well within that range.

For bioaccumulation, we could not find separate reliable measurements of MeHg and iHg in phytoplankton and zooplankton in the North Sea, and we therefore evaluated the tHg content. This was measured in diatoms to be 10±5 ng Hg mg⁻¹. This means that the mean bioaccumulation in our model in diatoms is lower, with 5.8 ng Hg mg⁻¹ and 9.0

ng Hg mg$^{-1}$ in the Northern and Southern North Sea, respectively, but still within 1 standard deviation of the measurements. Observations labeled as zooplankton and mysis were compared to our modeled microzooplankton and mesozooplankton, respectively. All modeled values fall within 1 standard deviation of the observed tHg concentration, with one exception: mesozooplankton in the Northern North Sea, which is 13.5% more than 1 standard deviation above the observations. This is mostly driven by a high iHg content, as the MeHg content is similar in microzooplankton and mesozooplankton.

This similarity in MeHg is caused because, even though mesozooplankton have a higher trophic level, they prefer to feed on larger diatoms which have less MeHg than smaller flagellates, which are preferred by microzooplankton. The high iHg content, especially in the Northern North Sea, is caused by the consumption of detritus by zooplankton in the model. While there is a shortage of data on bioaccumulation at the base of the food web, especially in the North Sea, which complicates model evaluation, the dissolved tHg concentration, the percentage of MeHg, and the tHg content of phytoplankton and zooplankton agree well with observations. With the exception of the 13.5% elevated tHg content in Northern North Sea mesozooplankton, all modeled values fall within 1 standard deviation of the observations. Because of this, we conclude that the model replicates marine Hg cycling and bioaccumulation at the base of the food web in line with observations, with the caveat that we do not have measurements of zooplankton in the Northern North Sea to verify or reject the elevated levels in that setup.

Table 5: Dissolved total Hg (pM), MeHg (% of tHg), and Hg concentrations in biota (ng Hg mg$^{-1}$ d.w.) across North Sea regions.

|  | Observed | NNS | SNS |
|---|---|---|---|
| tHg$_{\text{dissolved}}$ (pM) | $1.7 \pm 0.7$ | $1.7 \pm 0.26$ | $2.0 \pm 0.28$ |
| MeHg (% of tHg) | 3 (0.5–4.3) | $2.3 \pm 0.23$ | $2.0 \pm 0.31$ |
| Diatoms (ng Hg mg$^{-1}$) | $10 \pm 5$ | $7.0 \pm 1.1$ | $8.3 \pm 1.6$ |
| Flagellates (ng Hg mg$^{-1}$) |  | $13.9 \pm 3.0$ | $14.3 \pm 3.0$ |
| Microzooplankton tHg (ng Hg mg$^{-1}$) | $37.5 \pm 31.3$ | $67.4 \pm 29.3$ | $40.3 \pm 11.4$ |
| Microzooplankton MeHg (ng Hg mg$^{-1}$) |  | $7.1 \pm 2.1$ | $10.5 \pm 2.7$ |
| Mesozooplankton tHg (ng Hg mg$^{-1}$) | $62.5 \pm 12.5$ | $86.7 \pm 15.1$ | $72.3 \pm 19.6$ |
| Mesozooplankton MeHg (ng Hg mg$^{-1}$) |  | $6.9 \pm 2.6$ | $10.5 \pm 1.7$ |

**3 Technical Corrections**

**Reviewer Comment**

Line 8. Is the inorganic mercury in filter feeders elevated, or more specifically, is it found as a higher proportion of total mercury compared to other megabenthos?

**Author Response**

No, filter feeders do not have elevated iHg levels. The iHg levels in filter feeders are large part of tHg, but filter feeders have similar iHg as most other macrobenthos but elevated MeHg levels, thus a reduced proportion of iHg of tHg. The main megabenthos group that has notably increase in iHg are the suspension feeders. They are defined by their ability to eat DOM (resembling sponges), where filter feeders cannot filter out dissolved particles. Suspension feeders on the other hand, have both elevated iHg levels and an elavated iHg to tHg ratio. This is true both in the model and in the observation in Meditereanean Sea sponges by Orani et al. (2020).

**Reviewer Comment**

Line 18. Cite a reference for this statement.

**Author Response**

I will expand on this sentence and add the reference as below:

**Suggested edit**

Anthropogenic emissions have significantly raised environmental Hg levels, with 78%, 85%, and 50% of atmospheric, upper ocean, and deep ocean Hg, respectively, originating from anthropogenic emissions (Geyman et al., 2025).

'

**Reviewer Comment**

Line 26-27. Does bioconcentration only occur in a polluted environment? Is the model then only relevant for polluted environments?

**Author Response**

I will update that as below.

**Suggested edit**

There are two key processes involved in bioaccumulation: bioconcentration and biomagnification. When animals absorb Hg directly from their environment; this is called bioconcentration.

**Reviewer Comment**

Line 45. The first effectiveness evaluation of the Minamata Convention has not been completed. Rephrase this text.

**Suggested edit**

Mercury concentrations tend to be lower in organisms at the base of the aquatic food web, resulting in reduced risk to humans. As such, these organisms are not prioritized in the current monitoring strategies under the ongoing effectiveness evaluation of the Minamata Convention, which focuses primarily on fish, humans, and predatory wildlife (Evers et al., 2016). Additionally, the evaluation to date has shown that Hg and MeHg concentrations in water and sediment do not correlate well with levels in biota, leading to greater emphasis on biological monitoring over abiotic compartments.

**Reviewer Comment**

Line 51-52. Some megabenthos are not lower trophic level biota (e.g., are secondary consumers) and thus are not necessarily representative of processes at the base of the food web.

**Author Response**

That is true and the paper does also look at higher trohic level benthos. I will refrase this as follows:

**Suggested edit**

The benthic food web is highly complex, making it challenging to improve our understanding of bioaccumulation within it (Silberberger et al., 2018).

**Reviewer Comment**

Line 74. Perhaps change "in silico" to "a modelling experiment"

**Suggested edit**

Afterward, we conducted a modeling experiment in which megabenthos with various feeding strategies compete under physical drivers in idealized scenarios that are typical of megabenthos-rich coastal oceans.

**Reviewer Comment**

Line 86-89 and elsewhere. Use past tense to describe work that was completed.

**Author Response**

I will change that anywhere in the manuscript. I will update the paragraph with line 86-89 as below to correct this writing style:

**Suggested edit**

To compare the findings with the literature, we collected field studies measuring Hg in megabenthos. The studies we used are shown in Table 6. We categorized the megabenthos into the same feeding categories, "deposit feeder", "filter feeder", "suspension feeder", "grazer", and "predator". To better look at the effect of the trophic level, we also added "primary producers" as the base of the food web, and "seabird" and "benthic fish" as top predators. We analyzed whether trophic level and feeding strategy influence megabenthos iHg, MeHg, and/or tHg content. The observations were analyzed by visualizing the data, performing linear regression, and plotting a correlation matrix of the differences in bioaccumulation between different feeding strategies. The total and partial $R^2$ of the linear regression of the trophic level and the feeding strategy were compared to analyze the effect of both drivers on bioaccumulated iHg, MeHg, and tHg.

**Reviewer Comment**

Figure 1. What is the black line that connects biota to detritus, DOM and sediment organic carbon?

**Author Response**

I will update the caption of the figure as stated in Fig. 5. I also added a dotted black line to show mortality from phytoplankton and zooplankton.

**Reviewer Comment**

Line 186. Rephrase "samples are sampled"

**Suggested edit**

Because of this, most samples are from shallow, well-mixed coastal areas, and we used this setup to evaluate the performance of the models.

**Reviewer Comment**

Section 3.1. How do the results of this analysis compare with published findings reported in the literature?

**Author Response**

In the literature there is relatively little attention to direct role of feeding strategy or phyla on bioaccumulation. Several individual papers remark things such as that mussels have higher values than crustaceans but I could not find our main connclusion supported in the literature. That being said, I hope the expansian of the paper by doing the further analyses with the data presented by McClelland et al. (2024) is convincing that while this interaction itself is not supported as a conclusion in literature studies, published data does support our conclusions.

**Reviewer Comment**

Line 255. Unclear meaning – "validate that they survive in the model"

**Suggested edit**

While our megabenthos groups only vary in their feeding strategies and lack a direct real-world counterpart, it is important to ensure that all functional groups have consistent biomass in the model and thus the results originate from the modeled interactions, and are not altered due to unrealistically high or low biomass in the model.

[Figure]

Figure 5: The overview of the modeled megabenthos functional groups and how they interact with each other and functional groups in the ECOSMO E2E model. There are 5 macrobenthic functional groups. The filter feeder feeds on pelagic detritus, zooplankton, and phytoplankton. The suspension feeders feed on pelagic detritus, phytoplankton, zooplankton, and DOM. The generalist feeds on phytoplankton, zooplankton, pelagic detritus, and sediment organic carbon. The deposit feeder feeds on sediment organic carbon. The benthic predator feeds on the other 4 megabenthos functional groups and the top predator solely feeds on the benthic predator. The arrows indicate trophic interactions where the arrow goes from the prey to the predator and the arrows have the same colour as the prey. The black lines represent loss of organic material due to mortality. When megabenthos die, their organic carbon is transferred to pelagic DOM and detritus, as well as the sediment, shown by the solid black arrow. In contrast, when pelagic organisms die, their organic carbon is transferred to DOM and detritus, indicated by the dotted black arrow.

**Reviewer Comment**

Figure 6. How do mercury concentrations per unit carbon (reported in Figure 6) compare to literature reported values that do not take carbon content into account?

**Author Response**

As described in above, the conversion factor of carbon to dry weight is assumed to be a 1:2 ratio for all functional groups. This conversion is indeed a source of uncertainty. I expanded on this in the suggested expansion of the paper at the beginning of the revieuw, additionally as mentioned above I will ad a brief explanation of the conversion factor on line 152. I hope that this expansion provides enough depth there.

**Reviewer Comment**

Line 310-313. Are there published empirical studies that support this model prediction regarding filter feeders?

**Author Response**

There is published emperical data, as described at the beginning of these answer, that supports our conclusion. But no studies direclty comparing filter feeders in the way we did in this study with other feeding strategies. So our conclusion comes from the combination of having measurements that show elevated MeHg levels in filter feeders combined with a modeling explanation as why this might be because they are filter feeders.

**Reviewer Comment**

Line 361. Does the bioaccumulated inorganic mercury originate from water or dietary exposure?

**Author Response**

The majority of this in all catagories except suspension feeders is via bioconcentration, thus uptake from the water. I would expand the sentence at line 361 to clarify this better.

**Suggested edit**

In Fig. 6 we can see that the vast majority of iHg in filter, deposit, and generalist feeders originates from bioconcentration, thus direct uptake from the water is the dominant pathway of iHg bioaccumulation in our model in these feeding strategies.

**Reviewer Comment**

Line 394. Explain further what is meant by "in vivo mercury speciation".

**Author Response**

I will expand by the explanation in line 194 as follows:

**Suggested edit**

A final interaction that we did not take into account is *in vivo Hg speciation.* With in vivo Hg speciation we refer to the transfer of Hg from one form to another inside organisms. Examples of this are the earlier mentioned demethylation in sponges by Orani et al. (2020) and additional studies demonstrating Hg speciation in cuttlefish by Gente et al. (2023) and Hg-methylating bacteria in copepods by Gorokhova et al. (2020). This indicates that the bioaccumulation of inorganic Hg and methylmercury may not be fully independent processes. These reactions are currently, however, poorly understood, and more laboratory and field studies are necessary to improve our understanding of these transformations before their significance can be assessed and it can be determined whether their inclusion in bioaccumulation models is needed.

**Reviewer Comment**

Line 411. Where are these regression results presented earlier in the manuscript?

**Author Response**

They are from Table 3 where the model is evaluated, but I can better describe the table outcome in text. I would add the following part to the end of section 3.3. where I discuss the results of the allometric scaling model.

**Suggested edit**

Our base model does agree well with both observed iHg ($R^2$=0.84) and MeHg ($R^2$=0.86) in the Southern North Sea setup, but this is mostly driven by accurate predictions in the lower trophic levels while there is a normalized bias of -0.84 in the Top Predators. This is improved dramatically in the allometric scaling model with the reduction of the normalized bias of top predators to -0.32 which improves the overal $R^2$ of the model to >0.99.

**Reviewer Comment**

Line 436-437. This comment about bivalve communities is speculative.

**Reviewer Comment**

Line 441-442. Consider concluding the paper with a recommendation for future work on model development.

**Author Response**

I will answer these comments together by suggesting a rewrite of the closings remarks. I agree that it is speculative but I do think, especially considering the addition of the further analyses that also points als mollusc as being higher in MeHg that it is logical next step to take into account based on the results of this study. Would refrasing as below, and the expansion of the literature study that point in the same direction be an acceptable form to refrase it?

**Suggested edit**

Filter feeders and molluscs typically accumulate more MeHg than other organisms at similar trophic levels. This pattern is consistent not only in our models but also in available data. This raises a hypothesis that expanding bivalve populations, as seen in mussel or oyster farming, might affect MeHg bioaccumulation in higher trophic levels. This is supported by the observations that fish in lakes invaded by zebra mussels have higher Hg levels than fish in lakes without zebra mussels Blinick et al. (2024). However, such ecological alterations also impact other bioaccumulation factors like biomass distribution and trophic interactions. While our findings support the role of filter feeders and molluscs in MeHg dynamics and higher bioaccumulation in top predators, the complexity of ecological situations requires further specific studies to understand if and when bivalve communities lead to increased MeHg transfer.

Modeling studies can help our understanding of the factors influencing MeHg bioaccumulation, but their predictive capabilities need to be carefully validated. Our findings reveal that filter-feeding mollusks and DOM-utilizing sponges exhibit distinct Hg bioaccumulation patterns compared to other megabenthos. Modeling bivalve aquaculture or DOM-consuming Porifera can help explore their potential role in altering MeHg bioaccumulation. Understanding how functional traits like feeding strategy influence MeHg transfer remains key to improving both predictive models and environmental risk assessments.

**Author Response**

I would also add this section at the end of the model evaluation segment

**Suggested edit**

Evaluation of the GOTM-MERCY-ECOSMO setup reveals performance strengths and weaknesses of the model. There is a strong correlation between the modeled and observed effect of the feeding strategy on the bioaccumulation of MeHg in both the Northern and Southern North Sea setups. However, the Southern North Sea setup outperforms the Northern setup in both pelagic Hg cycling and megabenthos bioaccumulation assessments. In the Northern North Sea setup, the mesozooplankton tHg levels are higher than expected, although this could not be properly validated due to a lack of measurements. What is more notable is our model results show a reduction in MeHg bioaccumulation in the Northern North Sea compared to the Southern North Sea, while this is not present in the observations presented by McClelland et al. (2024). Since the Northern North Sea does have a good agreement with observations in the correlation between the feeding strategy and the bioaccumulation of MeHg, but the overall model performance is lower due to lower than expected MeHg concentrations in all benthic groups, it is likely that the model underestimates the flux of MeHg from the pelagic to the benthic system. In the Southern North Sea setup, macrobenthos can feed directly on the phytoplankton bloom which results in a major exchange of organic material and MeHg between the benthic and pelagic food webs. In the Northern North Sea this is not possible, and the consumption of detritus, either by filtering it from the water or feeding on it from the sediment, is the main food source for megabenthos. The higher performance of the model in the Southern North Sea compared to the Northern North Sea indicates that this model could be further improved by enhancing the MeHg dynamics in deeper water.

**4 Appendix**

Table 6: Data used for the literature study. TL = trophic level, FS = feeding strategy. Values for tHg and MeHg are in ng Hg mg$^{-1}$ d.w.. Samples that where marked as large and small are marked with (l) and (s) in the table respetively.

| Species | TL | FS | THg | MeHg | Location | Phylum | Reference |
|---|---|---|---|---|---|---|---|
| Stronglyocentrotus droebachiensis | 187 | Deposit | 24 | 2 | Svalbard | Echinodermata | Korejwo et al., 2022 |
| Ophiopholis aculeata | 270 | Filter | 47 | 2 | Svalbard | Echinodermata | Korejwo et al., 2022 |
| Baccinum glaciale | 371 | Deposit | 49 | 12 | Svalbard | Mollusca | Korejwo et al., 2022 |
| Henricia sp | 317 | Predator | 348 | 19 | Svalbard | Echinodermata | Korejwo et al., 2022 |
| Astarte borealis | 290 | Filter | 44 | 10 | Chukchi Sea | Mollusca | Fox et al., 2013 |
| Ampelisca macrocephala | 270 | Deposit | 70 | 32 | Chukchi Sea | Arthropoda | Fox et al., 2013 |
| Chinoecetes opilio | 410 | Predator | 131 | 102 | Chukchi Sea | Mollusca | Fox et al., 2013 |
| Neptunea heros | 430 | Predator | 195 | 171 | Chukchi Sea | Mollusca | Fox et al., 2013 |
| Buccinum spp | 410 | Predator | 269 | 171 | Chukchi Sea | Arthropoda | Fox et al., 2013 |
| Gammarellus sp. | 237 | Predator | 39 | 15 | Gulf of St. Lawrence | Mollusca | Lavoie et al., 2010 |
| Littorina littorea | 229 | Grazer | 51 | 13 | Gulf of St. Lawrence | Mollusca | Lavoie et al., 2010 |
| Buccinum undatum | 283 | Predator | 127 | 85 | Gulf of St. Lawrence | Mollusca | Lavoie et al., 2010 |
| Tectura testudinalis | 200 | Grazer | 51 | 9 | Gulf of St. Lawrence | Mollusca | Lavoie et al., 2010 |
| Strongylocentrotus droebachiensis | 169 | Deposit | 42 | 5 | Gulf of St. Lawrence | Echinodermata | Lavoie et al., 2010 |
| Hippoglossoides platessoides | 422 | Benthic fish | 146 | 77 | Gulf of St. Lawrence | Chordata | Lavoie et al., 2010 |
| Glyptocephalus cynoglossus | 422 | Benthic fish | 179 | 100 | Gulf of St. Lawrence | Chordata | Lavoie et al., 2010 |
| Somateria mollissima | 347 | Seabird | 640 | 565 | Gulf of St. Lawrence | Chordata | Lavoie et al., 2010 |
| Anguilla anguilla | | Benthic fish | 1161 | 895 | Baltic Sea | Chordata | Polak-Juszczak, 2018 |
| Gadus morhua | | Benthic fish | 346 | 269 | Baltic Sea | Chordata | Polak-Juszczak, 2018 |
| Platichtys flesus | | Benthic fish | 77 | | Baltic Sea | Chordata | Polak-Juszczak, 2014 |
| Platichtys flesus | | Benthic fish | 58 | | Baltic Sea | Chordata | Polak-Juszczak, 2014 |
| Pleuronectes platessa | | Benthic fish | 51 | | Baltic Sea | Chordata | Polak-Juszczak, 2014 |
| Pleuronectes platessa | | Benthic fish | 40 | | Baltic Sea | Chordata | Polak-Juszczak, 2014 |
| Scophthalmus maximus | | Benthic fish | 114 | | Baltic Sea | Chordata | Polak-Juszczak, 2014 |
| Scophthalmus maximus | | Benthic fish | 85 | | Baltic Sea | Chordata | Polak-Juszczak, 2014 |
| Macoma balthica | | Deposit | 53 | | Baltic Sea | Mollusca | Polak-Juszczak, 2014 |
| Macoma balthica | | Deposit | 25 | | Baltic Sea | Mollusca | Polak-Juszczak, 2014 |
| Saduria entomon | | Predator | 21 | | Baltic Sea | Arthropoda | Polak-Juszczak, 2014 |
| Saduria entomon | | Predator | 14 | | Baltic Sea | Arthropoda | Polak-Juszczak, 2014 |
| Acanthella acuta | | Suspension | 115 | 6 | Mediterranean Sea | Porifera | Orani et al., 2020 |
| Acanthella acuta | | Suspension | 66 | 7 | Mediterranean Sea | Porifera | Orani et al., 2020 |
| Acanthella acuta | | Suspension | 107 | 11 | Mediterranean Sea | Porifera | Orani et al., 2020 |
| Acanthella acuta | | Suspension | 95 | 9 | Mediterranean Sea | Porifera | Orani et al., 2020 |
| Axinella damicornis | | Suspension | 148 | 4 | Mediterranean Sea | Porifera | Orani et al., 2020 |
| Axinella damicornis | | Suspension | 97 | 2 | Mediterranean Sea | Porifera | Orani et al., 2020 |
| Axinella damicornis | | Suspension | 212 | 9 | Mediterranean Sea | Porifera | Orani et al., 2020 |
| Axinella damicornis | | Suspension | 252 | 7 | Mediterranean Sea | Porifera | Orani et al., 2020 |
| Chondrilla nucula | | Suspension | 149 | 2 | Mediterranean Sea | Porifera | Orani et al., 2020 |
| Chondrilla nucula | | Suspension | 233 | 1 | Mediterranean Sea | Porifera | Orani et al., 2020 |
| Chondrilla nucula | | Suspension | 519 | 4 | Mediterranean Sea | Porifera | Orani et al., 2020 |
| Chondrilla nucula | | Suspension | 317 | 4 | Mediterranean Sea | Porifera | Orani et al., 2020 |
| Haliclona fulva | | Suspension | 80 | 3 | Mediterranean Sea | Porifera | Orani et al., 2020 |
| Haliclona fulva | | Suspension | 76 | 2 | Mediterranean Sea | Porifera | Orani et al., 2020 |
| Haliclona fulva | | Suspension | 107 | 6 | Mediterranean Sea | Porifera | Orani et al., 2020 |
| Haliclona fulva | | Suspension | 146 | 6 | Mediterranean Sea | Porifera | Orani et al., 2020 |
| Halichondria panicea | | Suspension | 81 | 23 | Killkieran Bay | Porifera | Orani et al., 2020 |
| Halichondria panicea | | Suspension | 122 | 9 | Killkieran Bay | Porifera | Orani et al., 2020 |
| Hymeniacidon perlevis | | Suspension | 107 | 20 | Killkieran Bay | Porifera | Orani et al., 2020 |
| Hymeniacidon perlevis | | Suspension | 170 | 26 | Killkieran Bay | Porifera | Orani et al., 2020 |
| Chlamys nobilis | | Filter | 60 | 19 | Dapeng Bay | Mollusca | Pan and Wang, 2011 |
| Ruditapes philippinarum | | Filter | 47 | 17 | Tolo Harbo | Mollusca | Pan and Wang, 2011 |
| Saccostrea cucullata | | Filter | 70 | 15 | Clear Water Bay | Mollusca | Pan and Wang, 2011 |
| Perna viridis | | Filter | 30 | 9 | Tolo Harbo | Mollusca | Pan and Wang, 2011 |
| Septifer virgatus | | Filter | 92 | 10 | Clear Water Bay | Mollusca | Pan and Wang, 2011 |
| Mytilidae spp | | Filter | 142 | 79 | Eastern U.S. | Mollusca | Chen et al., 2014 |
| Mytilidae spp | | Filter | | 83 | Eastern U.S. | Mollusca | V. F. Taylor et al., 2019 |
| Mytilidae spp | | Filter | 95 | | Narragansett Bay, RI/MA U.S. | Mollusca | D. L. Taylor et al., 2012 |
| Mytilidae spp | | Filter | 173 | | Gulf of St. Lawrence | Mollusca | Cossa and Tabard, 2020 |
| Maldanidae spp. | | Deposit | 101 | 70 | Minas Basin, Bay of Fundy | Annelida | Sizmur et al., 2013 |
| Maldanidae spp. | | Deposit | 20 | | Bay of Fundy, Nova Scotia | Annelida | English et al., 2015 |
| Glyceridae spp. | | Predator | 37 | 9 | Minas Basin, Bay of Fundy | Annelida | Sizmur et al., 2013 |
| Nereidae spp. | | Deposit | 139 | | Narragansett Bay, RI/MA U.S. | Annelida | V. F. Taylor et al., 2019 |
| Nereidae spp. | | Deposit | 10 | | Bay of Fundy, Nova Scotia | Annelida | English et al., 2015 |
| Ilyanassa obsoleta | | Deposit | 177 | | Narragansett Bay, RI/MA U.S. | Mollusca | V. F. Taylor et al., 2019 |
| Ilyanassa obsoleta | | Deposit | 60 | | Gulf of Maine | Mollusca | Chen et al., 2009 |
| Ilyanassa obsoleta | | Deposit | 40 | | Bay of Fundy, Nova Scotia | Mollusca | English et al., 2015 |
| Litorrina littorea | | Grazer | 90 | | Narragansett Bay, RI/MA U.S. | Mollusca | V. F. Taylor et al., 2019 |
| Litorrina littorea | | Grazer | | 30 | Eastern U.S. | Mollusca | V. F. Taylor et al., 2019 |
| Litorrina littorea | | Grazer | 20 | | Bay of Fundy, Nova Scotia | Mollusca | English et al., 2015 |
| Corophium volutator | | Deposit | 43 | 11 | Minas Basin, Bay of Fundy | Mollusca | Sizmur et al., 2013 |
| Corophium volutator | | Deposit | 10 | | Bay of Fundy, Nova Scotia | Mollusca | English et al., 2015 |
| Amphipod spp. | | Deposit | 93 | | Narragansett Bay, RI/MA U.S. | Arthropoda | V. F. Taylor et al., 2019 |
| Amphipod spp. | | Deposit | 12 | | Gulf of Maine | Arthropoda | Chen et al., 2009 |
| Carcinus maenas | | Predator | 57 | 42 | Eastern U.S. | Arthropoda | Chen et al., 2014 |
| Carcinus maenas | | Predator | 126 | | Narragansett Bay, RI/MA U.S. | Arthropoda | V. F. Taylor et al., 2019 |
| Carcinus maenas | | Predator | | 80 | Eastern U.S. | Arthropoda | V. F. Taylor et al., 2019 |
| Macoma balthica | | Deposit | 10 | | Bay of Fundy, Nova Scotia | Mollusca | English et al., 2015 |
| Balanus balanus | 232 | Filter | 10 | 3 | Minas Basin, Bay of Fundy | Arthropoda | Bradford et al., 2023 |
| Carcinus maenas | 291 | Predator | 32 | 23 | Minas Basin, Bay of Fundy | Arthropoda | Bradford et al., 2023 |
| Corophium volutator | 200 | Deposit | 22 | 11 | Minas Basin, Bay of Fundy | Arthropoda | Bradford et al., 2023 |
| Glyceridae spp. | 415 | Predator | 29 | 18 | Minas Basin, Bay of Fundy | Annelida | Bradford et al., 2023 |
| Goniadidae spp. | 418 | Predator | 69 | 37 | Minas Basin, Bay of Fundy | Annelida | Bradford et al., 2023 |
| Ilyanassa obsoleta | 259 | Deposit | 139 | 26 | Minas Basin, Bay of Fundy | Mollusca | Bradford et al., 2023 |
| Lineidae spp. | 299 | Deposit | 31 | 13 | Minas Basin, Bay of Fundy | Annelida | Bradford et al., 2023 |
| Litorrina littorea | 234 | Grazer | 60 | 8 | Minas Basin, Bay of Fundy | Mollusca | Bradford et al., 2023 |
| Macoma balthica | 232 | Deposit | 76 | 11 | Minas Basin, Bay of Fundy | Mollusca | Bradford et al., 2023 |
| Maldanidae spp. | 237 | Deposit | 45 | 23 | Minas Basin, Bay of Fundy | Annelida | Bradford et al., 2023 |
| Mytilus edulis | 210 | Filter | 59 | 10 | Minas Basin, Bay of Fundy | Mollusca | Bradford et al., 2023 |
| Nereidae spp. | 221 | Deposit | 80 | 6 | Minas Basin, Bay of Fundy | Annelida | Bradford et al., 2023 |
| Pagurus acadiarnus | 290 | Predator | 17 | 7 | Minas Basin, Bay of Fundy | Arthropoda | Bradford et al., 2023 |

| Species | TL | FS | THg | MeHg | Location | Phylum | Reference |
|---|---|---|---|---|---|---|---|
| Phyllodocidae spp. | 294 | Predator | 67 | 18 | Minas Basin, Bay of Fundy | Annelida | Bradford et al., 2023 |
| *Balanus balanus* | | Filter | 7.92 | 1.87 | Minas Basin, Bay of Fundy | Arthropoda | Bradford et al., 2024 |
| *Corophium volutator* | | Deposit | 25.28 | 12.68 | Minas Basin, Bay of Fundy | Arthropoda | Bradford et al., 2024 |
| *Ilyanassa obsoleta* | | Deposit | 112.43 | 36.2 | Minas Basin, Bay of Fundy | Mollusca | Bradford et al., 2024 |
| *Littorina littorea* | | Grazer | 56.58 | 10.82 | Minas Basin, Bay of Fundy | Mollusca | Bradford et al., 2024 |
| Nereidae spp. | | Deposit | 15.16 | 4.29 | Minas Basin, Bay of Fundy | Annelida | Bradford et al., 2024 |
| *Balanus balanus* | | Filter | 15.87 | 2.47 | Minas Basin, Bay of Fundy | Arthropoda | Bradford et al., 2024 |
| *Ilyanassa obsoleta* | | Deposit | 195.39 | 8.76 | Minas Basin, Bay of Fundy | Mollusca | Bradford et al., 2024 |
| *Littorina littorea* | | Grazer | 57.58 | 5.74 | Minas Basin, Bay of Fundy | Mollusca | Bradford et al., 2024 |
| *Mytilus edulis* | | Filter | 62.07 | 13.97 | Minas Basin, Bay of Fundy | Mollusca | Bradford et al., 2024 |
| *Pagurus acadianus* | | Predator | 14.68 | 4.23 | Minas Basin, Bay of Fundy | Arthropoda | Bradford et al., 2024 |
| Corophium volutator | | Deposit | 23.95 | 8.68 | Minas Basin, Bay of Fundy | Arthropoda | Bradford et al., 2024 |
| Ilyanassa obsoleta | | Deposit | 161.23 | 45 | Minas Basin, Bay of Fundy | Mollusca | Bradford et al., 2024 |
| Littorina littorea | | Grazer | 81.97 | 3.23 | Minas Basin, Bay of Fundy | Mollusca | Bradford et al., 2024 |
| Nereidae spp. | | Deposit | 62.8 | 9.61 | Minas Basin, Bay of Fundy | Annelida | Bradford et al., 2024 |
| *Pagurus acadianus* | | Predator | 18.37 | 8.59 | Minas Basin, Bay of Fundy | Arthropoda | Bradford et al., 2024 |
| *Balanus balanus* | | Filter | 6.73 | 3.86 | Minas Basin, Bay of Fundy | Arthropoda | Bradford et al., 2024 |
| Corophium volutator | | Deposit | 19.59 | 9.94 | Minas Basin, Bay of Fundy | Arthropoda | Bradford et al., 2024 |
| Ilyanassa obsoleta | | Deposit | 116.3 | 17.6 | Minas Basin, Bay of Fundy | Mollusca | Bradford et al., 2024 |
| *Littorina littorea* | | Grazer | 50.88 | 9.53 | Minas Basin, Bay of Fundy | Mollusca | Bradford et al., 2024 |
| *Mytilus edulis* | | Filter | 56.3 | 7.33 | Minas Basin, Bay of Fundy | Mollusca | Bradford et al., 2024 |
| *Pagurus acadianus* | | Predator | 14.3 | 4.75 | Minas Basin, Bay of Fundy | Arthropoda | Bradford et al., 2024 |
| Corophium volutator | | Deposit | 21.24 | 12 | Minas Basin, Bay of Fundy | Arthropoda | Bradford et al., 2024 |
| Nereidae spp. | | Deposit | 95.66 | 4.75 | Minas Basin, Bay of Fundy | Annelida | Bradford et al., 2024 |
| Acanthostepheia behringiensis (l) | 350 | Deposit | | 80 | Cape Bathurst (CB) | Arthropoda | McClelland et al., 2024 |
| Acanthostepheia behringiensis (l) | 310 | Deposit | | 62 | Mackenzie Trough (MT) | Arthropoda | McClelland et al., 2024 |
| Acanthostepheia behringiensis (s) | 220 | Deposit | | 9 | Cape Bathurst (CB) | Arthropoda | McClelland et al., 2024 |
| Ampelisca spp. | 210 | Deposit | | 4 | Cape Bathurst (CB) | Arthropoda | McClelland et al., 2024 |
| Anonyx nugax | 370 | Deposit | | 231 | Mackenzie Trough (MT) | Arthropoda | McClelland et al., 2024 |
| Arctolembos arcticus | 220 | Deposit | | 7 | Cape Bathurst (CB) | Arthropoda | McClelland et al., 2024 |
| Paramphithoe sp. | 330 | Deposit | | 127 | Mackenzie Trough (MT) | Arthropoda | McClelland et al., 2024 |
| *Pleustes panoplus* | 270 | Deposit | | 21 | Cape Bathurst (CB) | Arthropoda | McClelland et al., 2024 |
| *Rhachotropis aculeata* | 310 | Deposit | | 55 | Cape Bathurst (CB) | Arthropoda | McClelland et al., 2024 |
| Unidentified amphipod sp. | 340 | Deposit | | 79 | Mackenzie Trough (MT) | Arthropoda | McClelland et al., 2024 |
| Diastylidae sp. | 200 | Deposit | | 8 | Cape Bathurst (CB) | Arthropoda | McClelland et al., 2024 |
| Diastylidae sp. | 190 | Deposit | | 12 | Mackenzie Trough (MT) | Arthropoda | McClelland et al., 2024 |
| *Munnopsis typica* | 270 | Deposit | | 29 | Cape Bathurst (CB) | Arthropoda | McClelland et al., 2024 |
| *Saduria entomon* | 310 | Deposit | | 53 | Mackenzie Trough (MT) | Arthropoda | McClelland et al., 2024 |
| *Saduria sabini* | 350 | Deposit | | 39 | Cape Bathurst (CB) | Arthropoda | McClelland et al., 2024 |
| *Saduria sabini* | 280 | Deposit | | 47 | Mackenzie Trough (MT) | Arthropoda | McClelland et al., 2024 |
| *Synidotea bicuspida* | 310 | Deposit | | 19 | Cape Bathurst (CB) | Arthropoda | McClelland et al., 2024 |
| *Synidotea bicuspida* | 290 | Deposit | | 28 | Mackenzie Trough (MT) | Arthropoda | McClelland et al., 2024 |
| *Eualus gaimardii* | 350 | Generalist | | 52 | Cape Bathurst (CB) | Arthropoda | McClelland et al., 2024 |
| *Eualus gaimardii* | 340 | Generalist | | 106 | Mackenzie Trough (MT) | Arthropoda | McClelland et al., 2024 |
| *Sabinea septemcarinata* | 330 | Generalist | | 35 | Cape Bathurst (CB) | Arthropoda | McClelland et al., 2024 |
| *Sabinea septemcarinata* | 350 | Generalist | | 93 | Mackenzie Trough (MT) | Arthropoda | McClelland et al., 2024 |
| Spirontocaris sp. | 300 | Generalist | | 134 | Mackenzie Trough (MT) | Arthropoda | McClelland et al., 2024 |
| Nymphonidae sp. | 340 | Predator | | 32 | Cape Bathurst (CB) | Arthropoda | McClelland et al., 2024 |
| Nymphonidae sp. | 390 | Predator | | 299 | Mackenzie Trough (MT) | Arthropoda | McClelland et al., 2024 |
| | 330 | Filter feeder | | 29 | Cape Bathurst (CB) | Cnidaria | McClelland et al., 2024 |
| | 330 | Filter feeder | | 38 | Mackenzie Trough (MT) | Cnidaria | McClelland et al., 2024 |
| *Gersemia rubiformis* | 260 | Filter feeder | | 17 | Mackenzie Trough (MT) | Cnidaria | McClelland et al., 2024 |
| *Crossaster papposus* | 350 | Predator | | 367 | Mackenzie Trough (MT) | Echinodermata | McClelland et al., 2024 |
| *Leptasterias littoralis* | 270 | Predator | | 66 | Cape Bathurst (CB) | Echinodermata | McClelland et al., 2024 |
| Leptasterias spp. | 290 | Filter feeder | | 244 | Mackenzie Trough (MT) | Echinodermata | McClelland et al., 2024 |
| *Ophiocten sericeum* | 240 | Deposit | | 8 | Cape Bathurst (CB) | Echinodermata | McClelland et al., 2024 |
| *Ophiocten sericeum* | 260 | Deposit | | 14 | Mackenzie Trough (MT) | Echinodermata | McClelland et al., 2024 |
| *Ophiacantha bidentata* | 340 | Deposit | | 42 | Mackenzie Trough (MT) | Echinodermata | McClelland et al., 2024 |
| *Stegophiura nodosa* | 280 | Deposit | | 13 | Cape Bathurst (CB) | Echinodermata | McClelland et al., 2024 |
| Psolus sp. | 220 | Filter | | 17 | Mackenzie Trough (MT) | Echinodermata | McClelland et al., 2024 |
| Amphiporus sp. | 280 | Deposit | | 34 | Cape Bathurst (CB) | Nemertea | McClelland et al., 2024 |
| Unidentified sipunculid sp. A | **110** | Deposit | | 1.5 | Cape Bathurst (CB) | Annelida | McClelland et al., 2024 |
| Unidentified sipunculid sp. A | **130** | Deposit | | 1.5 | Mackenzie Trough (MT) | Annelida | McClelland et al., 2024 |
| Bylgides sp. | 340 | Predator | | 24 | Cape Bathurst (CB) | Annelida | McClelland et al., 2024 |
| Bylgides sarsi | 300 | Predator | | 54 | Mackenzie Trough (MT) | Annelida | McClelland et al., 2024 |
| *Gattyana cirrhosa* | *320* | Predator | | 69 | Cape Bathurst (CB) | Annelida | McClelland et al., 2024 |
| *Harmothoe imbricata* | *310* | Predator | | 40 | Cape Bathurst (CB) | Annelida | McClelland et al., 2024 |
| Nephtys sp. | ***290*** | Predator | | 149 | Mackenzie Trough (MT) | Annelida | McClelland et al., 2024 |
| *Nereis zonata* | *230* | Deposit | | 20 | Mackenzie Trough (MT) | Annelida | McClelland et al., 2024 |
| *Pectinaria hyperborea* | *190* | Deposit | | 48 | Cape Bathurst (CB) | Annelida | McClelland et al., 2024 |
| *Phyllodoce groenlandica* | *210* | Predator | | 41 | Cape Bathurst (CB) | Annelida | McClelland et al., 2024 |
| Unidentified polychaete sp.* | 220 | Predator | | 22 | Mackenzie Trough (MT) | Annelida | McClelland et al., 2024 |
| Astarte borealis | **250** | Filter | | 32 | Cape Bathurst (CB) | Mollusca | McClelland et al., 2024 |
| Astarte spp | 270 | Filter | | 36 | Cape Bathurst (CB) | Mollusca | McClelland et al., 2024 |
| Astarte spp | 250 | Filter | | 59 | Mackenzie Trough (MT) | Mollusca | McClelland et al., 2024 |
| *Clinocardium ciliatum* | 250 | Filter | | 71 | Cape Bathurst (CB) | Mollusca | McClelland et al., 2024 |
| *Clinocardium ciliatum* | 260 | Filter | | 173 | Mackenzie Trough (MT) | Mollusca | McClelland et al., 2024 |
| Cyclocardia sp. | *180* | Filter | | 89 | Mackenzie Trough (MT) | Mollusca | McClelland et al., 2024 |
| *Nuculana pernula* | **240** | Deposit | | 47 | Mackenzie Trough (MT) | Mollusca | McClelland et al., 2024 |
| *Similipecten greenlandicus* | *290* | Deposit | | 76 | Cape Bathurst (CB) | Mollusca | McClelland et al., 2024 |
| *Similipecten greenlandicus* | *290* | Deposit | | 146 | Mackenzie Trough (MT) | Mollusca | McClelland et al., 2024 |
| *Yoldia hyperborea* | *220* | Deposit | | 15 | Cape Bathurst (CB) | Mollusca | McClelland et al., 2024 |
| *Boreotrophon truncatus* | 260 | Predator | | 10 | Mackenzie Trough (MT) | Mollusca | McClelland et al., 2024 |
| Buccinum polare | 300 | Predator | | 172 | Mackenzie Trough (MT) | Mollusca | McClelland et al., 2024 |
| Buccinum spp. | 260 | Predator | | 89 | Mackenzie Trough (MT) | Mollusca | McClelland et al., 2024 |
| Cryptonatica sp. | 280 | Predator | | 141 | Mackenzie Trough (MT) | Mollusca | McClelland et al., 2024 |
| Cylichna sp. | 220 | Predator | | 26 | Mackenzie Trough (MT) | Mollusca | McClelland et al., 2024 |
| *Margarites costalis* | 290 | Deposit | | 61 | Mackenzie Trough (MT) | Mollusca | McClelland et al., 2024 |
| Neptunea sp. | 280 | Predator | | 268 | Mackenzie Trough (MT) | Mollusca | McClelland et al., 2024 |
| Tachyrhynchus spp. | 140 | Predator | | 14 | Mackenzie Trough (MT) | Mollusca | McClelland et al., 2024 |

**References**

Allison, J. D., Allison, T. L., & Ambrose, R. B. (2005). *ALLISON, J. D. AND T. L. ALLISON. PARTITION COEFFICIENTS FOR METALS IN SURFACE WATER, SOIL, AND WASTE. U.S. Environmental Protection Agency, Washington, DC* (tech. rep.).

Amptmeijer, D. J., Bieser, J., Mikheeva, E., Daewel, U., & Schrum, C. (2025). Bioaccumulation as a driver of high MeHg in coastal Seas. *EGUsphere [preprint]*.

Bieser, J., Amptmeijer, D., Daewel, U., Kuss, J., Soerenson, A. L., & Schrum, C. (2023). The 3D biogeochemical marine mercury cycling model MERCY v2.0; linking atmospheric Hg to methyl mercury in fish. *Geoscientific Model Development Discussions*, 1–59.

Blinick, N. S., Link, D., Ahrenstorff, T. D., Bethke, B. J., Fleishman, A. B., Janssen, S. E., Krabbenhoft, D. P., Nelson, J. K. R., Rantala, H. M., Rude, C. L., & Hansen, G. J. A. (2024). Increased mercury concentrations in walleye and yellow perch in lakes invaded by zebra mussels.

Bradford, M. A., Mallory, M. L., & O'Driscoll, N. J. (2023). Mercury bioaccumulation and speciation in coastal invertebrates: Implications for trophic magnification in a marine food web. *Marine Pollution Bulletin*, *188*, 114647.

Bradford, M. A., Mallory, M. L., & O'Driscoll, N. J. (2024). Ecology and environmental characteristics influence methylmercury bioaccumulation in coastal invertebrates. *Chemosphere*, *346*, 140502.

Buckman, K. L., Seelen, E. A., Mason, R. P., Balcom, P., Taylor, V. F., Ward, J. E., & Chen, C. Y. (2019). Sediment organic carbon and temperature effects on methylmercury concentration: a mesocosm experiment. *The Science of the total environment*, *666*, 1316.

Chen, C. Y., Borsuk, M. E., Bugge, D. M., Hollweg, T., Balcom, P. H., Ward, D. M., Williams, J., & Mason, R. P. (2014). Benthic and Pelagic Pathways of Methylmercury Bioaccumulation in Estuarine Food Webs of the Northeast United States. *PLOS ONE*, *9*(2), e89305.

Chen, C. Y., Dionne, M., Mayes, B. M., Ward, M., Darron, Sturup Stefan, & Brian, J. P. (2009). Mercury Bioavailability and Bioaccumulation in Estuarine Food Webs in the Gulf of Maine. *Environ. Sci. Technol.*, *43*, 1804–1810.

Coquery, M., & Cossa, D. (1995). Mercury speciation in surface waters of the north sea. *Netherlands Journal of Sea Research*, *34*(4), 245–257.

Cossa, D., & Tabard, A.-M. (2020). Mercury in Marine Mussels from the St. Lawrence Estuary and Gulf (Canada): A Mussel Watch Survey Revisited after 40 Years. *Applied science*, *10*, 7556.

Dutton, J., & Fisher, N. S. (2012). Bioavailability of sediment-bound and algal metalsto killifish Fundulus heteroclitus. *Aquatic biology*, *16*, 85–96.

Eklöf, J., Austin, Å., Bergström, U., Donadi, S., Eriksson, B. D., Hansen, J., & Sundblad, G. (2017). Size matters: Relationships between body size and body mass of common coastal, aquatic invertebrates in the Baltic Sea. *PeerJ*, *2017*(1), e2906.

English, M. D., Robertson, G. J., & Mallory, M. L. (2015). Trace element and stable isotope analysis of fourteen species of marine invertebrates from the Bay of Fundy, Canada. *Marine Pollution Bulletin*, *101*(1), 466–472.

Evers, D. C., Egan Keane, S., Basu, N., & Buck, D. (2016). Evaluating the effectiveness of the Minamata Convention on Mercury: Principles and recommendations for next steps.

Fox, A. L., Hughes, E. A., Trocine, R. P., Trefry, J. H., Schonberg, S. V., Mctigue, N. D., Lasorsa, B. K., Konar, B., & Cooper, L. W. (2013). Mercury in the northeastern Chukchi Sea: Distribution patterns in seawater and sediments and biomagnification in the benthic food web.

Gente, S., Minet, A., Lopes, C., Tessier, E., Gassie, C., Guyoneaud, R., Swarzenski, P. W., Bustamante, P., Metian, M., Amouroux, D., & Lacoue-Labarthe, T. (2023). In Vivo

Mercury (De)Methylation Metabolism in Cephalopods under Different pCO 2 Scenarios. *Cite This: Environ. Sci. Technol*, *57*, 5770.

Geyman, B. M., Streets, D. G., Olson, C. I., Thackray, C. P., Olson, C. L., Schaefer, K., Krabbenhoft, D. P., & Sunderland, E. M. (2025). Cumulative Anthropogenic Impacts of Past and Future Emissions and Releases on the Global Mercury Cycle. *Environmental Science and Technology*, *59*(17), 8578–8590.

Gorokhova, E., & Hansson, S. (2000). Elemental composition of Mysis mixta (Crustacea, Mysidacea) and energy costs of reproduction and embryogenesis under laboratory conditions. *Journal of Experimental Marine Biology and Ecology*, *246*(1), 103–123.

Gorokhova, E., Soerensen, A. L., & Motwani, N. H. (2020). Mercury-methylating bacteria are associated with copepods: A proof-of-principle survey in the Baltic Sea. *PLoS ONE*, *15*(3).

Jurkiewicz-Karnkowska, E. (2005). Some Aspects of Nitrogen, Carbon and Calcium Accumulation in Molluscs from the Zegrzyński Reservoir Ecosystem. *Polish Journal of Environmental Studies*, *14*(2), 173–177.

Korejwo, E., Saniewska, D., Bełdowski, J., Balazy, P., & Saniewski, M. (2022). Mercury concentration and speciation in benthic organisms from Isfjorden, Svalbard. *Marine Pollution Bulletin*, *184*, 114115.

Lavoie, R. A., Hebert, C. E., Rail, J.-F., Braune, B. M., Yumvihoze, E., Hill, L. G., & Lean, D. R. S. (2010). Trophic structure and mercury distribution in a Gulf of St. Lawrence (Canada) food web using stable isotope analysis.

McClelland, C., Chételat, J., Conlan, K., Aitken, A., Forbes, M. R., & Majewski, A. (2024). Methylmercury dietary pathways and bioaccumulation in Arctic benthic invertebrates of the Beaufort Sea. *Arctic Science*, *10*(2), 305–320.

Nfon, E., Cousins, I. T., Järvinen, O., Mukherjee, A. B., Verta, M., & Broman, D. (2009). Trophodynamics of mercury and other trace elements in a pelagic food chain from the Baltic Sea.

Orani, A. M., Vassileva, E., Azemard, S., & Thomas, O. P. (2020). Comparative study on Hg bioaccumulation and biotransformation in Mediterranean and Atlantic sponge species. *Chemosphere*, *260*, 127515.

Pan, K., & Wang, W. X. (2011). Mercury accumulation in marine bivalves: Influences of biodynamics and feeding niche. *Environmental Pollution*, *159*(10), 2500–2506.

Pickhardt, P. C., Stepanova, M., & Fisher, N. S. (2006). Contrasting uptake routes and tissue distributions of inorganic and methylmercury in mosquitofish (Gambusia affinis) and redear sunfish (Lepomis microlophus). *Environmental Toxicology and Chemistry*, *25*(8), 2132–2142.

Polak-Juszczak, L. (2014). Selenium and mercury molar ratios in commercial fish from the Baltic Sea: Additional risk assessment criterion for mercury exposure.

Polak-Juszczak, L. (2018). Distribution of organic and inorganic mercury in the tissues and organs of fish from the southern Baltic Sea. *Environmental Science and Pollution Research*, *25*, 34181–34189.

Silberberger, M. J., Renaud, P. E., Kröncke, I., & Reiss, H. (2018). Food-web structure in four locations along the European shelf indicates spatial differences in ecosystem functioning. *Frontiers in Marine Science*, *5*(APR), 300569.

Sizmur, T., Canário, J., Gerwing, T. G., Mallory, M. L., & O'Driscoll, N. J. (2013). Mercury and methylmercury bioaccumulation by polychaete worms is governed by both feeding ecology and mercury bioavailability in coastal mudflats. *Environmental Pollution*, *176*, 18–25.

Taylor, D. L., Linehan, J. C., Murray, D. W., & Prell, W. L. (2012). Indicators of sediment and biotic mercury contamination in a southern New England estuary. *Marine Pollution Bulletin*, *64*(4), 807–819.

Taylor, V. F., Buckman, K. L., Seelen, E. A., Mazrui, N. M., Balcom, P. H., Mason, R. P., & Chen, C. Y. (2019). Organic carbon content drives methylmercury levels in the water column and in estuarine food webs across latitudes in the Northeast United States. *Environmental Pollution*, *246*, 639–649.

Tesán Onrubia, J. A., Petrova, M. V., Puigcorbé, V., Black, E. E., Valk, O., Dufour, A., Hamelin, B., Buesseler, K. O., Masqué, P., Le Moigne, F. A., Sonke, J. E., Rutgers Van Der Loeff, M., & Heimbürger-Boavida, L. E. (2020). Mercury Export Flux in the Arctic Ocean Estimated from 234Th/238U Disequilibria. *ACS Earth and Space Chemistry*, *4*(5), 795–801.

Tsui, M. T., & Wang, W. X. (2004). Uptake and Elimination Routes of Inorganic Mercury and Methylmercury in Daphnia magna. *Environmental Science and Technology*, *38*(3), 808–816.

Wang, W. X., & Tan, Q. G. (2019). Applications of dynamic models in predicting the bioaccumulation, transport and toxicity of trace metals in aquatic organisms. *Environmental Pollution*, *252*, 1561–1573.

Wang, W., & Wong, R. (2003). Bioaccumulation kinetics and exposure pathways of inorganic mercury and methylmercury in a marine fish, the sweetlips Plectorhinchus gibbosus. *Marine Ecology Progress Series*, *261*.

Weil, J., Trudel, M., Tucker, S., Brodeur, R. D., & Juanes, F. (2019). Percent ash-free dry weight as a robust method to estimate energy density across taxa. *Ecology and Evolution*, *9*(23), 13244–13254.

---

## Author Comment (AC3)

**1 Answers to the technical corrections and suggestions**

**Reviewer Comment**

On line 18: Add reference for the threefold increase in environmental Hg and specify whether this data is a global average and the environmental medium that it comes from (ie. sediment and peat archives?)

**Author Response**

I will expand on this sentence and add the reference as below:

**Suggested edit**

Anthropogenic emissions have significantly raised environmental Hg levels, with 78%, 85%, and 50% of atmospheric, upper ocean, and deep ocean Hg, respectively, originating from anthropogenic emissions (Geyman et al., 2025).

**Reviewer Comment**

Line 30: Volume concentration factor (6.4E6) – specify units if applicable.

**Suggested edit**

The bioconcentration process can result in high concentrations in aquatic organisms. This process is commonly quantified using the Volume Concentration Factor (VCF), a unitless ratio between the Hg concentration in phytoplankton and that in the surrounding water:

$$\text{VCF} = \frac{C_{\text{phytoplankton}}}{C_{\text{water}}} \tag{1}$$

where both $C_{\text{phytoplankton}}$ and $C_{\text{water}}$ have the same units, for example, ng Hg $\mu\text{m}^{-3}$. For MeHg, very high volume concentration factors of up to $6.4 \times 10^6$ have been reported in the literature (Lee & Fisher, 2016; Schartup et al., 2018).

**Reviewer Comment**

Line 31: Sentence is overly casual – recommend removing or revising.

**Author Response**

I would make revise the sentence as below:

**Suggested edit**

MeHg concentrations that are elevated due to bioconcentration can be further increased by biomagnification along the aquatic food web.

**Reviewer Comment**

Line 34: Revise to: "Consumption of MeHg-contaminated seafood is the primary pathway of mercury exposure in humans, with elevated risk among coastal and seafood-reliant populations (Zhang et al. 2021)." This revised version better emphasizes exposure pathways while remaining sensitive to the context of seafood-dependent communities. If you

choose to expand on health effects, a brief mention of methylmercury's neurotoxicity could provide a natural transition to your discussion of Minamata Bay. If you do retain the sentences in lines 34 – 37, also consider briefly clarifying Minamata Bay's specific contamination source, to not create a false sense of fear that these pollution levels are common.
Reference: Zhang, et al. (2021) Global health effects of future atmospheric mercury emissions. Nat. Commun. https://doi.org/10.1038/s41467-021-23391-7

**Suggested edit**

Consumption of MeHg-contaminated seafood is the primary pathway of mercury exposure in humans, with elevated risk among coastal and seafood-reliant populations (Sheehan et al., 2014; Zhang et al., 2021). MeHg is a neurotoxin whose overconsumption can decrease IQ points (Trasande et al., 2006), and raise the risk of heart attacks (Genchi et al., 2017). The risk associated with consuming seafood contaminated with MeHg gained significant attention after over 1000 fatalities occurred in Japan in 1956 due to the consumption of contaminated seafood from Minamata Bay(Harada, 1995). Although this MeHg outbreak was a unique event linked to industrial waste disposal containing Hg, it highlighted the dangers of MeHg exposure. To mitigate the risk of acute MeHg contamination incidents, like those in Minamata, and to minimize long-term MeHg exposure, the Minamata Convention on Mercury was established (UNEP, 2013).

**Reviewer Comment**

Line 39: Rephrase to avoid starting with a number or acronym (e.g., "A total of 151 countries...").

**Author Response**

I will update that to:

**Suggested edit**

A total of 151 countries have pledged to reduce their Hg emissions in support of the Minamata Convention and 128 countries have signed and ratified the convention. The global state of Hg as a pollutant and the effect of the Minamata Convention is periodically reviewed in the Minamata Convention Effectiveness Evaluation (Outridge et al., 2018).

**Reviewer Comment**

Line 95: Add references for the coupled models (ie. GOTM, ECOSMO E2E and Mercy v2.0). Alternatively, the subheadings 2.3, 2.4, and 2.5 could be changed to 2.2.1, 2.2.2, and 2.2.3 respectively as they all fall under the "2.2 The models" subheading.

**Author Response**

I will update the subheadings to a lower level as you suggest untill model development, so the structure is:
2.2 The models
2.2.1 The hydrodynamical model
2.2.2 The MERCY v2.0 model
2.2.3 ECOSMO E2E

2.3 Model Development

Additionally I will add the references in addition to their respective paragraph the the introductory sentence on line 95 as fellows:

**Suggested edit**

We used a fully coupled 1D water column model that is run in 2 setups that resemble typical hydrological regimes found in coastal oceans. We coupled the Generalized Ocean Turbulence Model (GOTM) (Burchard et al., 1999) with the ECOSMO E2E ecosystem model (Daewel et al., 2019) and the Mercy v2.0 Hg speciation and bioaccumulation model (Bieser et al., 2023).

**Reviewer Comment**

Figure 1: Uses URL links within the figure caption which is generally not recommended. One possibility for rewording the caption is: "Several sub-images were used to create this figure. Image sources (used under Creative Commons licenses or in the public domain) are as follows: Filter feeder: Sabella spallanzanii (image by Wikipedia contributors, CC BY-SA 3.0, via Wikipedia)."

**Author Response**

The images do not all have the same license, sometimes it is CC BY-SA 4.0, CC BY-SA 2.0 or public domain. I would reword it as below to remove the URLS but do specify the license, owner and source for all images.

**Suggested edit**

Filter feeder: *Sabella spallanzanii* (photo by Diego Delso, CC BY-SA 4.0, via Wikipedia), Suspension feeder: *Aplysina fistularis* (photo by Twilight Zone Expedition Team 2007, NOAA-OE, CC BY 2.0, via Flickr), Generalist feeder: *Crangon crangon* (photo by Etrusko25, Public Domain, via Wikipedia), Deposit feeder: *Buccinum undatum* (photo by Oscar Bos / Ecomare, CC BY 4.0, via Wikipedia), Benthic predator: *Hommarus gammarus* (photo by Bart Braun, Public Domain, via Wikipedia), Top predator: *Sepia officinalis* (photo by Nick Hobgood, CC BY-SA 3.0, via Wikipedia).

**Reviewer Comment**

Line 148: Provide justification or reference for the bprotected value used.

**Suggested edit**

The macrobenthos in the North Sea are estimated to have beween 1.1 and 35.5 gC m$^{-2}$ (Daan & Mulder, 2001; Heip et al., 1992). The value for $B_{Protected}$ is chosen as 1 gC m$^{-2}$ for all macrobenthos except for the benthic predator where $B_{Protected}$ is 0.5 gC m$^{-2}$. These values are chosen to protect macrobenthos functional groups from extinction due to predation when there values are below the expected range.

**Reviewer Comment**

Line 155: Maintain consistent MeHg/iHg order throughout the sentence for clarity.

**Suggested edit**

An assimilation efficiency of 0.31 for iHg and 0.95 for MeHg is chosen for everything except deposit feeding, which has a lower feeding efficiency of 0.07 for iHg and 0.12 for MeHg based on Dutton and Fisher (2012).

**Reviewer Comment**

Line 176: Unsure of what units d-1 refers to.

**Author Response**

I add $d^{-1}$ (per day) the first time this abbreviation is uased. I also rewrote the sentence so that it does not start with a number.

**Suggested edit**

When organic carbon (detritus, labile DOM, and semi-labile DOM) is produced, 5% is allocated to semi-labile DOM. Additionally, detritus breaks down into semi-labile DOM at a rate of 0.001 $d^{-1}$ (per day).

**Reviewer Comment**

Line 210: Include a reference for B10 value interpretation and the Jeffreys–Zellner–Siow prior assumption.

**Author Response**

I will this as below and add a refrence for the Monte Carlo analyses and add the reference for the Jeffrew Zellner Prior

**Suggested edit**

We first calculated the normalized bias as $(modeled - observed)/modeled$ for the average modeled and observed values. After this, we quantified the probability that the modeled mean is of the same distribution as the observations by performing a Bayes factor analysis. The Bayes factor value is estimated by first estimating the likelihood of the modeled mean under the H0 hypothesis, which assumes that the modeled and observed data share the same distribution, and the H1 hypothesis, which assumes that they do not share a distribution. The likelihood of the H1 hypothesis over the H0 hypotheses is the BF10 value. The BF10 factor is estimated using a Jeffreys–Zellner–Siow prior assumption so we assume no prior knowledge (Zellner & Siow, 1980).

**Reviewer Comment**

Line 216: Rephrase for clarity. For example: "A BF10 factor below 1 supports the H1 hypothesis, while BF10 values $< 0.1$ and $< 0.01$ are considered strong and very strong evidence, respectively, in favor of the H0 hypothesis."

**Author Response**

I will this as below, and include a reference:

**Suggested edit**

A BF10 factor below 1 indicates that the modeled distribution is more likely the same as the observed distribution, and a BF10 <0.1 can be considered strong evidence and a BF10 <0.01 as very strong evidence in favor of the H0 hypotheses (Doll & Jacquemin, 2019).

**Reviewer Comment**

Figure 2 caption: Final sentence seems to have been cut down short. Recommend: "This contrasts the iHg concentration (<100 ng g-1 d.w.) for all animals, except starfish, eel, and sponges." The caption should also clarify that the data shown came from a literature review. If each point comes from a separate study, consider citing sources directly in the figure legend.

**Author Response**

I will move this figure to the supplement information. And update the legend. As it has all the data from the dataset I will clearly specify that.

**Suggested edit**

The effect of different feeding strategies on the measured MeHg and iHg in several benthic functional groups and groups of animals feeding on the benthos. The figure is the combination of all literature data presented in Table 5. The seabird is the common eider which feeds on benthos. Bioaccumulated MeHg is below 50 ng g$^{-1}$ d.w. for all functional groups that are not predatorial (predators, benthic fish, and seabirds), but can reach up to 171, 565, and 895 ng g$^{-1}$ d.w. for predators, seabirds, and benthic fish respectively. This contrasts the iHg concentration below 100 ng g$^{-1}$ d.w. for every animal. Except for starfish, Eel, and sponges. The tHg shows that the Hg can even be higher in suspension feeders (in this case sponges) than in fish.

**Reviewer Comment**

Line 256: Typo. Should read: "...followed by deposit feeders with up to 5 g C m-2."

**Author Response**

I will correct that as follows:

**Suggested edit**

Filter feeders have the highest biomass, which is up to 10 g C m$^{-2}$ followed by deposit feeders with up to 5 g C m$^{-2}$, generalist feeders with up to 3 g C m$^{-2}$, and suspension feeders with up to 1 g C m$^{-2}$.

**Reviewer Comment**

Figure 7: Recommend removing plot titles and reformatting to look more like Figure 3. The Hg species should be identified in the y-axis label, and the order of the Hg species should match Figure 3 (MeHg, iHg, tHg). Will also need to be repeated for figure 8.

**Author Response**

I will redo the plots with the updated layout script as shown in Fig. 1. I will also combine Fig. 7 and Fig. 8 so that the differences between the base model and allometric scaling model are easier to evaluate.

[Figure]

Figure 1: The inluence of trophic level on the bioaccumulation of MeHg, iHg, and tHg in both the AS (panels a, c, e) and the base model (panels b, d, f). In the AS model, the relationship with trophic level is stronger, where ln(MeHg)=1.24TL-0.03, compared to the base model, which is ln(MeHg)=0.64TL+1.42. TL represents trophic level, and MeHg is expressed in ng Hg g$^{-1}$ d.w. For iHg, the bioaccumulation patterns are nearly identical, with ln(MeHg)=-0.19TL+5.11 for the AS model and ln(MeHg)=-0.18TL+5.11 for the base model, both showing a weak inverse correlation with trophic level, largely due to higher iHg levels in low trophic level feeders. In terms of tHg, there is a higher increase in bioaccumulation in the AS model (ln(MeHg)=0.43TL+3.76) compared to the base model (ln(MeHg)=0.04TL+4.175), driven by the stronger association between MeHg and trophic level in the AS model.

> **Reviewer Comment**
>
> Table 2 and 3 captions: Define AS as allometric scaling in the caption only

> **Author Response**
>
> Updated caption Table 2

> **Suggested edit**
>
> Comparison of modeled and observed Hg and MeHg bioaccumulation in different feeding strategies for the Southern North Sea (SNS), Northern North Sea (NNS), and field observations. Values are presented as ranges with means in parentheses. Units are ng Hg g d.w. for iHg and MeHg, and% for MeHg percentage. The bottom two rows are the predator and top predator frm the allometric scaling model (AS).

> **Author Response**
>
> Updated caption Table 3

> **Suggested edit**
>
> Statistical analysis of model performance for iHg and MeHg levels by feeding strategy for Southern North Sea (SNS) and Northern North Sea (NNS). The predator and top predator of both the default setup and Allometric Scaling (AS) model is shown.

**References**

Bieser, J., Amptmeijer, D., Daewel, U., Kuss, J., Soerenson, A. L., & Schrum, C. (2023). The 3D biogeochemical marine mercury cycling model MERCY v2.0; linking atmospheric Hg to methyl mercury in fish. *Geoscientific Model Development Discussions*, 1–59.

Burchard, H., Bolding, K., & Villarreal, M. R. (1999). *GOTM, a General Ocean Turbulence Model. Theory, implementation and test cases* (tech. rep.).

Daan, R., & Mulder, M. (2001). The macrobenthic fauna in the Dutch sector of the North Sea in 2003 and a comparison with previous data. *NIOZ-RAPPORT*, *2001-2*, 97.

Daewel, U., Schrum, C., & MacDonald, J. I. (2019). Towards end-to-end (E2E) modelling in a consistent NPZD-F modelling framework (ECOSMO E2E-v1.0): Application to the North Sea and Baltic Sea. *Geoscientific Model Development*, *12*(5), 1765–1789.

Doll, J. C., & Jacquemin, S. J. (2019). Bayesian Model Selection in Fisheries Management and Ecology. *Journal of Fish and Wildlife Management*, *10*(2), 691–707.

Dutton, J., & Fisher, N. S. (2012). Bioavailability of sediment-bound and algal metalsto killifish Fundulus heteroclitus. *Aquatic biology*, *16*, 85–96.

Genchi, G., Sinicropi, M. S., Carocci, A., Lauria, G., & Catalano, A. (2017). Mercury Exposure and Heart Diseases. *International Journal of Environmental Research and Public Health*, *14*(1), 74.

Geyman, B. M., Streets, D. G., Olson, C. I., Thackray, C. P., Olson, C. L., Schaefer, K., Krabbenhoft, D. P., & Sunderland, E. M. (2025). Cumulative Anthropogenic Impacts of Past and Future Emissions and Releases on the Global Mercury Cycle. *Environmental Science and Technology*, *59*(17), 8578–8590.

Harada, M. (1995). Minamata Disease: Methylmercury Poisoning in Japan Caused by Environmental Pollution. *Critical Reviews in Toxicology*, *25*(1), 1–24.

Heip, C., Basford, D., Craeymeersch, J. A., Dewarumez, J.-m., Dorjes, J., de Wilde, P., Duineveld, G., Eleftheriou, A., J Herman, P. M., Niermann, U., Kingston, P., Kiinitzer, A., Rachor, E., Rumohr, H., Soetaert, K., Soltwedel Heip, T., Wilde, d., Heip A Craeymeersch, C. J., Soetaert, a., ... Kiinitzer, S. A. (1992). Trends in biomass, density and diversity of North Sea macrofauna. *ICESJ. mar. Sci*, *49*, 13–22.

Lee, C. S., & Fisher, N. S. (2016). Methylmercury uptake by diverse marine phytoplankton. *Limnology and Oceanography*, *61*(5), 1626–1639.

Outridge, P. M., Mason, R. P., Wang, F., Guerrero, S., & Heimbürger-Boavida, L. E. (2018, October). Updated Global and Oceanic Mercury Budgets for the United Nations Global Mercury Assessment 2018.

Schartup, A. T., Qureshi, A., Dassuncao, C., Thackray, C. P., Harding, G., Sunderland, E. M., Harvard, †., & Paulson, J. A. (2018). A Model for Methylmercury Uptake and Trophic Transfer by Marine Plankton. *Environ. Sci. Technol*, *52*, 18.

Sheehan, M. C., Burke, T. A., Navas-Acien, A., Breysse, P. N., Mcgready, J., & Fox, M. A. (2014). Systematic reviews Global methylmercury exposure from seafood consumption and risk of developmental neurotoxicity: a systematic review. *Bull World Health Organ*, *92*, 254–269.

Trasande, L., Schechter, C., Haynes, K. A., Landrigan, P. J., & Acad Sci, A. N. (2006). Applying Cost Analyses to Drive Policy That Protects Children Mercury as a Case Study.

UNEP. (2013). Global Mercury Assessment 2013: Sources, Emissions, Releases and Environmental Transport. *UNEP Chemicals Branch, Geneva, Switzerland*.

Zellner, A., & Siow, A. (1980). Posterior Odds Ratios for Selected Regression Hypotheses.

Zhang, Y., Song, Z., Huang, S., Zhang, P., Peng, Y., Wu, P., Gu, J., Dutkiewicz, S., Zhang, H., Wu, S., Wang, F., Chen, L., Wang, S., & Li, P. (2021). Global health effects of future atmospheric mercury emissions. *Nature Communications 2021 12:1*, *12*(1), 1–10.

---

## Author Response (AR1)

**1 Introduction**

**Author Response**

Dear Reviewers,

Thank you for your careful reading of our manuscript and for the constructive feedback. This is our second response to the reviewer comments, and in some cases our responses have been updated compared to the initial version. One significant change we made in this revision is the simplification of the model evaluation statistics for the global dataset: we have removed the Bayes factor likelihood analyses. With the additional data provided by McClelland et al. (2024), using only normalized bias, RMSE, NRMSE, and  $R^2$  (Pearson and residual) makes the results clearer and easier to interpret. We would be happy to reinstate the previous analyses if preferred. Below, we provide a point-by-point response, indicating the changes made and the corresponding line numbers in the revised manuscript.

**2 Answer to reviewer 1**

**2.1 Answers to the technical corrections and suggestions**

**Reviewer Comment**

On line 18: Add reference for the threefold increase in environmental Hg and specify whether this data is a global average and the environmental medium that it comes from (ie. sediment and peat archives?)

**Edit**

Line 18

These anthropogenic emissions have significantly raised environmental Hg levels, with 78%, 85%, and 50% of atmospheric, upper ocean, and deep ocean Hg, respectively, originating from anthropogenic emissions Geyman 2025.

**Reviewer Comment**

Line 30: Volume concentration factor (6.4E6) – specify units if applicable.

**Edit**

Line 30

The bioconcentration process can result in high concentrations in aquatic organisms. This process is commonly quantified using the Volume Concentration Factor (VCF), a unitless ratio between the Hg concentration in phytoplankton and that in the surrounding water:

$$VCF = \frac{C_{\text{phytoplankton}}}{C_{\text{water}}} \tag{1}$$

where both  $C_{\rm phytoplankton}$  and  $C_{\rm water}$  have the same units, for example, ng Hg  $\mu$ m-3. For MeHg, very high volume concentration factors of up to  $6.4 \times 10^6$  have been reported in the literature (Lee & Fisher, 2016; Schartup et al., 2018).

**Reviewer Comment**

Line 31: Sentence is overly casual – recommend removing or revising.

**Edit**

Line 37

MeHg concentrations that are elevated due to bioconcentration can be further increased by biomagnification along the aquatic food web.

**Reviewer Comment**

Line 34: Revise to: "Consumption of MeHg-contaminated seafood is the primary pathway of mercury exposure in humans, with elevated risk among coastal and seafood-reliant populations (Zhang et al. 2021)." This revised version better emphasizes exposure pathways while remaining sensitive to the context of seafood-dependent communities. If you choose to expand on health effects, a brief mention of methylmercury's neurotoxicity could provide a natural transition to your discussion of Minamata Bay. If you do retain the sentences in lines 34-37, also consider briefly clarifying Minamata Bay's specific contamination source, to not create a false sense of fear that these pollution levels are common.

Reference: Zhang, et al. (2021) Global health effects of future atmospheric mercury emissions. Nat. Commun. https://doi.org/10.1038/s41467-021-23391-7

**Edit**

Line 40

MeHg is a neurotoxin whose overconsumption can decrease IQ points and raise the risk of heart attacks, and consumption of MeHg-contaminated seafood is the primary pathway of Hg exposure in humans, with elevated risk among coastal and seafood-reliant populations Sheehan 2014, Zhang 2021, Giuseppe 2017, Trasande 2006.

The risk associated with consuming seafood contaminated with MeHg gained significant attention after over 1000 fatalities occurred in Japan in 1956 due to the consumption of contaminated seafood from Minamata Bay Harada1995. Although this MeHg outbreak was a unique event linked to industrial waste disposal containing Hg, it highlighted the dangers of MeHg exposure. In order to reduce the risk of further outbreaks of MeHg intoxications, the Minamata Convention on Mercury was founded. A total of 151 countries have pledged to reduce their Hg emissions in support of the Minamata Convention and 128 countries have signed and ratified the convention UNEP2013. The global state of Hg as a pollutant and the effect of the Minamata Convention is periodically reviewed in the Minamata Convention Effectiveness Evaluation Outridge2018Updated2018.

**Reviewer Comment**

Line 39: Rephrase to avoid starting with a number or acronym (e.g., "A total of 151 countries...").

**Edit**

Line 48

A total of 151 countries have pledged to reduce their Hg emissions in support of the Minamata Convention and 128 countries have signed and ratified the convention. The global state of Hg as a pollutant and the effect of the Minamata Convention is periodically

reviewed in the Minamata Convention Effectiveness Evaluation (Outridge et al., 2018).

**Reviewer Comment**

Line 95: Add references for the coupled models (ie. GOTM, ECOSMO E2E and Mercy v2.0). Alternatively, the subheadings 2.3, 2.4, and 2.5 could be changed to 2.2.1, 2.2.2, and 2.2.3 respectively as they all fall under the "2.2 The models" subheading.

**Author Response**

I will update the subheadings to a lower level as you suggest until model development, so the structure is:

- 2.1 The models
- 2.1.1 The hydrodynamical model
- 2.1.2 The physical setups
- 2.1.3 The MERCY v2.0 model
- 2.2.4 ECOSMO E2E
- 2.2 Model Development

Additionally I will add the references in addition to their respective paragraph the the introductary sentence on line 95 as fellows:

**Edit**

Line 96

We used a fully coupled 1D water column model that is run in 2 setups that resemble typical hydrological regimes found in coastal oceans. We coupled the Generalized Ocean Turbulence Model (GOTM) GOTM1999 with the ECOSMO E2E ecosystem model Daewel2019 and the MERCY v2.0 Hg speciation and bioaccumulation model Bieser2022.

**Reviewer Comment**

Figure 1: Uses URL links within the figure caption which is generally not recommended. One possibility for rewording the caption is: "Several sub-images were used to create this figure. Image sources (used under Creative Commons licenses or in the public domain) are as follows: Filter feeder: Sabella spallanzanii (image by Wikipedia contributors, CC BY-SA 3.0, via Wikipedia)."

**Edit**

Caption Fig. 1

Filter feeder: Sabella spallanzanii (photo by Diego Delso, CC BY-SA 4.0, via Wikipedia), Suspension feeder: Aplysina fistularis (photo by Twilight Zone Expedition Team 2007, NOAA-OE, CC BY 2.0, via Flickr), Generalist feeder: Crangon crangon (photo by Etrusko25, Public Domain, via Wikipedia), Deposit feeder: Buccinum undatum (photo by Oscar Bos / Ecomare, CC BY 4.0, via Wikipedia), Benthic predator: Hommarus gammarus (photo by Bart Braun, Public Domain, via Wikipedia), Top predator: Sepia officinalis (photo by Nick Hobgood, CC BY-SA 3.0, via Wikipedia).

**Reviewer Comment**

Line 148: Provide justification or reference for the byrotected value used.

**Edit**

Line 169

The macrobenthos in the North Sea are estimated to have between 1.1 and 35.5 gC m-2 Heip1992, Daan2001. The value for BProtected is chosen as 1 gC m-2 for all macrobenthos except for the benthic predator where BProtected is 0.5 gC m-2.

**Reviewer Comment**

Line 155: Maintain consistent MeHg/iHg order throughout the sentence for clarity.

**Edit**

Line 182

An assimilation efficiency of 0.95 for MeHg and 0.31 for iHg is chosen for everything except deposit feeding, which has a lower feeding efficiency of 0.07 for iHg and 0.43 for MeHg according to Dutton and Fisher, 2012.

**Reviewer Comment**

Line 176: Unsure of what units d-1 refers to.

**Edit**

Line 190

When organic carbon (detritus, labile DOM, and semi-labile DOM) is produced, 5% is allocated to semi-labile DOM. Additionally, detritus breaks down into semi-labile DOM at a rate of 0.001 d-1 (per day).

**Reviewer Comment**

Line 210: Include a reference for B10 value interpretation and the Jeffreys–Zellner–Siow prior assumption.

**Author Response**

As mentioned at the beginning of the document I would suggest to remove the BF10 values from the paper as I believe they introduce more uncertainty than that they clarify.

**Reviewer Comment**

Line 216: Rephrase for clarity. For example: "A BF10 factor below 1 supports the H1 hypothesis, while BF10 values

Figure 1: The inluence of trophic level on the bioaccumulation of MeHg, iHg, and tHg in both the AS (panels a, c, e) and the base model (panels b, d, f). In the AS model, the relationship with trophic level is stronger, where ln(MeHg)=1.24TL-0.03, compared to the base model, which is ln(MeHg)=0.64TL+1.42. TL represents trophic level, and MeHg is expressed in ng Hg g-1 d.w. For iHg, the bioaccumulation patterns are nearly identical, with ln(MeHg)=-0.19TL+5.11 for the AS model and ln(MeHg)=-0.18TL+5.11 for the base model, both showing a weak inverse correlation with trophic level, largely due to higher iHg levels in low trophic level feeders. In terms of tHg, there is a higher increase in bioaccumulation in the AS model (ln(MeHg)=0.43TL+3.76) compared to the base model (ln(MeHg)=0.04TL+4.175), driven by the stronger association between MeHg and trophic level in the AS model.

**Reviewer Comment**

Table 2 and 3 captions: Define AS as allometric scaling in the caption only

**Edit**

Updated caption Table 2

Comparison of modeled and observed Hg and MeHg bioaccumulation in different feeding strategies for the Southern North Sea (SNS), Northern North Sea (NNS), and field observations. Values are presented as ranges with means in parentheses. Units are ng Hg g d.w. for iHg and MeHg, and% for MeHg percentage. The bottom two rows are the predator and top predator frm the allometric scaling model (AS).

**Edit**

Updated caption Table 3

Statistical analysis of model performance for iHg and MeHg levels by feeding strategy for Southern North Sea (SNS) and Northern North Sea (NNS). The predator and top predator of both the default setup and Allometric Scaling (AS) model is shown.

**3 Answers to reviewer 2**

**Author Response**

Dear reviewer,

Thank you for your comments and taking the time to review the paper. Before I get to answering your specific comments, I want to respond to what I agree is a very general criticism of the paper that should be properly addressed, which is related to using data from different geographic areas. The issue is that there is very little data on MeHg bioaccumulation at the base of the food web, and almost nothing in the North Sea. This complicates model validation and is the reason why the decision was made to simply try to use all data available. However, 2 recent papers came out with a lot of data, of which one paper, by McClelland et al., 2024 samples 476 benthic animals from two locations in the Canadian Arctic. While this is a different location, it does allow us to evaluate if certain patterns between feeding strategies are consistent in samples from the same geographic location. Because of this, we expanded the paper by verifying if the observed patterns from the model and global dataset analyses are also present in a single dataset.

This is introduced in the methods section in line 258 in the section:

2.3.3 Evaluation of the model using a single dataset

The results are presented in line 434 in section:

3.5 The role of the feeding strategy on MeHg bioaccumulation in a single case stud

**3.1 Direct answers on reviewer comments**

**Reviewer Comment**

Since the primary objective of the study is to model Hg bioaccumulation, I recommend that the model evaluation be presented as part of the Results and Discussion rather than the Methods. This change would strengthen the narrative and reduce redundancy—many of the points currently discussed in Section 3 could be streamlined. I suggest restructuring Section 3.1 to serve as the model evaluation, followed by subsequent sections explaining key discrepancies between model output and observations (currently in Section 4).

**Author Response**

The manuscript is overhauled in detail. Currently the paper has the following structure:

- 3 Results
  - 3.1 Model evaluation
    - \* 3.1.1 Evaluation of the Hg cycling and pelagic bioaccumulation
    - \* 3.1.2 Megabenthic biomass
  - 3.2 Bioaccumulation in the model
  - 3.3 The allometric scaling law in high trophic level animals
  - 3.4 Bioaccumulation in the global dataset
  - 3.5 Comparing model and observations
    - \* 3.4.1 The effect of feeding strategy on bioaccumulation
    - \* 3.4.2 The statistical performance of the model
    - \* 3.4.3 The effect of water column mixing
    - \* 3.4.4 Deposit feeders
  - $-\ 3.5$  The role of the feeding strategy on MeHg bioaccumulation in a single case study
- 4 Discussion
  - 4.1 The role of feeding strategy on the bioaccumulation of MeHg
  - 4.2 The AS model
  - 4.3 Bioconcentration of iHg
  - 4.4 Model structural limitations
  - 4.5 Data-related limitations
  - 4.6 Potential improvements
- 5 Summary and conclusion

This allows way the paper has the model evaluate as part of the results to streamline the results section.

**Author Response**

We will also update Fig. 6 to show the bioaccumulation in the AS model, as shown in Fig. 5. This because this is the better performing version of the model and we can therefore

Figure 2: Modeled bioconcentration and biomagnification of iHg and MeHg. Partitioning to detritus and DOM is colored as bioconcentration. The y-axis is cut to show the high and low values. Notably is the high iHg to mgC ratio of detritus and DOM, leading to elevated iHg in suspension feeders. Additionally, higher trophic level animals have higher biomagnified MeHg.

better anlyse this version.

**Author Response**

I would also add this section ad the end of the model evaluation segment

**Edit**

Line 500

The model has the same rates for all megabenthos groups. This allows us to isolate the effect of the feeding strategy, but it should be taken into account that this also means that the model is limited in its ability to predict bioaccumulation of iHg or MeHg in specific animals. Our model is run in the North Sea, while most of the field observations are from different regions. This means that this study should be seen as a hypothesis-generating work that identifies the role of feeding strategies on the bioaccumulation of iHg and MeHg as a potential direction for further empirical studies, rather than a complete classification. Based on this work, however, it appears that the inclusion of megabenthos with different feeding strategies could improve the performance of MeHg bioaccumulation models. At the same time, our analyses demonstrate the underperformance of the model in simulating the deep water bentho-pelagic coupling, which indicates that the performance of the ECOSMO E2E-MERCY-GOTM coupled system should be critically evaluated before it can be used for predictive bioaccumulation modelling in deeper water.

**Author Response**

I will add this to the beginning of the model evaluation section to show the Hg cycling and bioaccumulation in the pelagic is in line with observations.

**Edit**

**3.2 Evaluation of the Hg cycling and pelagic bioaccumulation**

Line 266

The marine cycling and speciation of Hg, in addition to the bioaccumulation in phytoplankton and zooplankton, is an essential driver of the bioaccumulation of iHg and MeHg in the benthic food web. Observed and modelled dissolved tHg concentration, the percentage of tHg that is MeHg, and the Hg content of phytoplankton and zooplankton is shown in Table 2. The concentration of dissolved tHg and the percentage of MeHg of dissolved tHg are compared to observations by Coquery and Cossa (1995), while the bioaccumulation of tHg in phytoplankton and zooplankton is compared to observations by Nfon et al. (2009). It must be noted that the observations by Nfon et al. (2009) are not from the North Sea itself, but from the better-studied nearby Baltic Sea. The average dissolved tHg concentration is 1.7 and 2.1,pM in the Northern and Southern North Sea, respectively. This is well within 1 standard deviation of the 1.7±0.7 pM observed by Coquery and Cossa (1995). The MeHg concentration was observed to be between 0.5 and 4.3% of tHg, with an average of 3% in the North Sea. The percentage MeHg in our model is 2.3% and 2.0% on average, which falls well within that range.

For bioaccumulation, we could not find separate reliable measurements of MeHg and iHg in phytoplankton and zooplankton in the North Sea, and we therefore evaluated the tHg content. This was measured in diatoms to be  $10\pm5$  ng Hg mg-1. This means that the mean bioaccumulation in our model in diatoms is lower, with 5.8 ng Hg mg-1 and 9.0 ng Hg mg-1 in the Northern and Southern North Sea, respectively, but still within 1 standard deviation of the measurements. Observations labeled as zooplankton and mysis were compared to our modeled microzooplankton and mesozooplankton, respectively. All modeled values fall within 1 standard deviation of the observed tHg concentration, with one exception: mesozooplankton in the Northern North Sea, which is 13.5% more than 1 standard deviation above the observations. This is mostly driven by a high iHg content, as the MeHg content is similar in microzooplankton and mesozooplankton.

This similarity in MeHg is caused because, even though mesozooplankton have a higher trophic level, they prefer to feed on larger diatoms which have less MeHg than smaller flagellates, which are preferred by microzooplankton. The high iHg content, especially in the Northern North Sea, is caused by the consumption of detritus by zooplankton in the model. While there is a shortage of data on bioaccumulation at the base of the food web, especially in the North Sea, which complicates model evaluation, the dissolved tHg concentration, the percentage of MeHg, and the tHg content of phytoplankton and zooplankton agree well with observations. With the exception of the 13.5% elevated tHg content in Northern North Sea mesozooplankton, all modeled values fall within 1 standard deviation of the observations. Because of this, we conclude that the model replicates marine Hg cycling and bioaccumulation at the base of the food web in line with observations, with the caveat that we do not have measurements of zooplankton in the Northern North Sea to verify or reject the elevated levels in that setup.

Table 1: Dissolved total Hg (pM), MeHg (% of tHg), and Hg concentrations in biota (ng Hg mg-1 d.w.) across North Sea regions.

|                                                 | Observed        | NNS             | SNS             |
|-------------------------------------------------|-----------------|-----------------|-----------------|
| tHg dissolved (pM)                   | $1.7\pm0.7$     | $1.7 \pm 0.26$  | $2.0 \pm 0.28$  |
| MeHg (% of tHg)                                 | 3 (0.5-4.3)     | $2.3 \pm 0.23$  | $2.0 \pm 0.31$  |
| Diatoms (ng Hg $mg^{-1}$ )                      | $10 \pm 5$      | $7.0\pm1.1$     | $8.3\pm1.6$     |
| Flagellates (ng Hg $mg^{-1}$ )                  |                 | $13.9 \pm 3.0$  | $14.3 \pm 3.0$  |
| Microzooplankton tHg (ng Hg mg -1 )  | $37.5 \pm 31.3$ | $67.4 \pm 29.3$ | $40.3 \pm 11.4$ |
| Microzooplankton MeHg (ng Hg mg -1 ) |                 | $7.1 \pm 2.1$   | $10.5 \pm 2.7$  |
| Mesozooplankton tHg (ng Hg mg -1 )   | $62.5 \pm 12.5$ | $86.7 \pm 15.1$ | $72.3 \pm 19.6$ |
| Mesozooplankton MeHg (ng Hg mg -1 )  |                 | $6.9 \pm 2.6$   | $10.5 \pm 1.7$  |

**Author Response**

I would suggest to add the below expansion of the discussion about what drives the role in feeding strategy.

**Edit**

**Line 488**

Combining the results of the model and the literature studies is difficult due to the high uncertainty in most drivers, including the organic material content of dry weight, and the result should be viewed with skepticism. For example, the data analyses by McClelland et al., 2024 were prepared to mimic consumption by predators: for small arthropods, their skin was not removed, but for gastropods and bivalves, the shell was not taken into account for the weight as predators would typically not eat this. The concentration of MeHg per unit energy is arguably the key measure in bioaccumulation. Predators need to ingest a specific energy amount, so if a prey is composed of half organic material and half non-organic components, such as shell, its MeHg content per dry weight is halved. However, predators would consume double the dry weight to obtain the energy, and thus the same MeHg. In general, the energy appears to be consistent with Ash Free Dry Weight (AFDW), as such ideally we would normalize all measurements of MeHg bioaccumulation per AFDW Weil2019.

Unfortunately, doing this conversion reliably on published data is not possible as AFDW varies with the age and body size of animals, which information is not always registered and made available Eklof2017.

**Author Response**

Additionally I would add the following component to the concluding remarks

**Edit**

Line 536

Filter feeders and molluscs typically accumulate more MeHg than other organisms at similar trophic levels. This pattern is consistent not only in our models but also in available data. This raises a hypothesis that expanding bivalve populations, as seen in mussel or oyster farming, might affect MeHg bioaccumulation in higher trophic levels. This is supported by the observations that fish in lakes invaded by zebra mussels have higher Hg levels than fish in lakes without zebra mussels Blinick et al. (2024). However, such

ecological alterations also impact other bioaccumulation factors like biomass distribution and trophic interactions. While our findings support the role of filter feeders and molluscs in MeHg dynamics and higher bioaccumulation in top predators, the complexity of ecological situations requires further case specific studies to understand if and when bivalve communities lead to increased MeHg transfer.

Modeling studies can help our understanding of the factors influencing MeHg bioaccumulation, but ability to accurately predicht MeHg bioaccumulations needs to be carefully validated. Our findings reveal that filter-feeding molluses and DOM-utilizing suspension feeders have different Hg bioaccumulation patterns compared to other megabenthos. Modeling bivalve aquaculture or DOM-consuming suspension feeders can help explore their potential role in altering MeHg bioaccumulation. Understanding how functional traits like feeding strategy influence MeHg transfer remains key to improving both predictive models and environmental risk assessments.

**Reviewer Comment**

The purpose of Figure 3 is unclear. It is not evident why the authors chose to use Hg data from different ecosystems and plot them against trophic level (referred to as feeding strategy in the figure). Since ecosystems differ in baseline inorganic Hg and MeHg concentrations, the MeHg-trophic level relationship should be examined within each ecosystem independently.

**Author Response**

We removed the image, instead we performed the ecosystem specific analyses based on the McClelland et al. (2024). The original purpose of Fig. 3 was to have a slope for MeHg bioaccumulation to compare our model too, but this is now done by Fig. 6 on line 462 was is based on a single geographical location and therefore does not have the same concerns.

**Reviewer Comment**

For model evaluation, I strongly suggest plotting modeled versus observed concentrations of speciated Hg (inorganic and MeHg) for each modeled feeding strategy. This would provide a clearer and more direct assessment of model performance.

**Author Response**

We will do this by making a barplot for both MeHg and iHg with a broken axis for MeHg so we can show both higher and lower trophic levels. I would also suggest to only do this for the MeHg so it can show both lower and higher trophic levels. We will show Fig. 3 for MeHg and Fig. 4 for iHg..

Figure 3: Mean bioaccumulation of MeHg in both model and observations is shown, with error bars representing 1 SE. The model's predictions do not consistently match observations within 1 SE, yet they display a comparable trend: Top Predators have the highest MeHg levels, followed by predators, with generalists and filter feeders showing similar MeHg, which are higher than those found in deposit feeders. Both model and observations show that suspension feeders have the lowest MeHg levels among the feeding strategies.

Figure 4: Mean bioaccumulation of iHg in both the model and observations. Error bars represent 1 SE. The model accurately shows increased iHg levels in suspension feeders. However, for all other feeding categories, except top predators, the iHg content is overestimated. In top predators, observed iHg levels are higher and not reflected in the model, though these elevated iHg levels have a very high SE.

**Reviewer Comment**

Section 3.2.3, which addresses the effect of feeding strategy on bioaccumulation, is central to the manuscript's aims, yet it is not discussed in sufficient depth. In contrast, the manuscript devotes substantial space to explaining Hg vs. trophic level patterns (Section 3.2.4), which are already well-established in the literature. I recommend condensing the discussion in 3.2.4 and focusing more on how feeding strategies influence MeHg and inorganic Hg transfer, particularly in benthic food webs.

**Author Response**

To address this we removed the segment about trophic level as it is indeed very established. To compensate for this we expanded by validating the the identified role of the feeding strategy is also present in the single dataset published by McClelland et al. (2024). Additionally we expanded the discussion in segments

**Reviewer Comment**

Figure 4 is difficult to interpret. It is unclear whether the data are empirical or simulated. A more straightforward approach might be to present Hg concentrations across feeding strategies as a bar chart with error bars. If the intent is to show correlations between feeding strategies, a correlation coefficient would be more appropriate.

**Author Response**

Figure 4 is removed from the manuscript. Rather we added 2 bar charts comparing iHg and MeHg bioaccumulation in the model and observations in line 372.

**Reviewer Comment**

Section 3.3, on allometric scaling, should appear earlier in the manuscript. When reading Sections 3.2.3 and 3.2.4, I repeatedly found myself wondering about the effects of allometric scaling on the results. Figures 7 and 8 could be consolidated to allow readers to compare model performance with and without allometric scaling more clearly.

**Author Response**

We moved the segment about the allometric scaling model forward to section 3.3 so it is discussed before the main comparisons between the model and observations are made. Additionally we concolodated Fig. 7 and 8 until the new Fig.3 Line 347.

**Reviewer Comment**

Lines 340–350: This content would be better integrated into the allometric scaling section.

**Author Response**

The majority of this was added to the section about the AS model in the discussion section (Line 475).

**Reviewer Comment**

As the authors note, the model is implemented for the North Sea, yet many of the empirical datasets used for comparison originate from other regions. This mismatch raises concerns about the validity of the model evaluation. Comparing model output to observations from ecologically distinct systems—each with different baseline Hg and MeHg levels, food web structures, and biogeochemical conditions—complicates interpretation and undermines the credibility of the evaluation. I strongly recommend either (1) limiting the model evaluation to observed data from the North Sea, or (2) running separate models parameterized for the specific ecosystems from which the empirical data are drawn.

**Author Response**

I agree with this concern of data being drawn from different locations. Unfortunately, there is not enough data from the North Sea to evaluate the model purely on data from the North Sea. I think the biggest concern is that there might be a cocorrelation between certain areas with high MeHg and the feeding strategies that are commen there. I hope that involving the evaluation component purely based on the McClelland et al. (2024) data adresses this concern to some degree. The model is not purely designed to be a predictive model for different megabenthos species in the North Sea, rather it is aimed to analyse if feeding strategy is a significant driver of MeHg bioaccumulation. The aim is to show that running a model in an idealized 1D setup resembling coastal conditions results in differences in the bioaccumulation caused by feeding strategies. Notably, our model shows higher MeHg in filter feeders compared to deposit feeders and extremely elevated levels of iHg in suspension feeders consuming DOM. We find it convincing the same pattern occurs in our model, the global dataset and the McClelland et al. (2024) dataset, but I fully agree that better measurements are necessary to validate models before they can be used in a predictive capacity. Our hope is that this manuscripts helps the message that studies to measurements of biaoccumulation in the benthic food web can be relevant to increase our understandig of bioaccumulation in higher trophic levels.

**4 Answer to reviewer 3**

**4.1 General answer**

**Author Response**

Thank you for your great comments. Before I go into the individual comments in detail, I want to acknowledge a limitation in the paper that you flagged in several of your comments. This is related to the evaluation of the model output and the problem that the combining of data from various locations brings. This was done because, although data on Hg bioaccumulation can be found more often, studies measuring both MeHg and Trophic Level are less common. That being said, two big studies came out in 2024 that after I did the original data analyzes and I can include these papers. Although 2 papers sounds small, they have a lot of samples and drastically improve the sample size. Especially the paper by McClelland et al. (2024) can really supplement the paper. They sampled 476 benthic animals from two locations in the canadian Arctic. We added a third component to the paper where we verify that the patterns observed in our model and the global dataset are also present in the single dataset published by McClelland et al. (2024).

This is introduced in the methods section in line 258 in the section:

2.3.3 Evaluation of the model using a single dataset

The results are presented in line 434 in section:

3.5 The role of the feeding strategy on MeHg bioaccumulation in a single case stud

**4.2 Specific Comments**

**Reviewer Comment**

Data on mercury concentrations in marine megabenthos were compiled and examined for differences in bioaccumulation by feeding strategy. It appears a relatively small number of studies were used (n = 12, Table 5) compared to available published data on mercury in marine megabenthos. What criteria were used for the literature review and selection of papers?

**Author Response**

The papers were selected with a focus on having both trophic level and MeHg concentrations estimated in marine megabenthos. These studies are indeed less common than studies sampling only metal concentration, including total Hg. While verifying new literature, I did find two new papers that were published last year that I would now include in the analyses Bradford et al., 2024; McClelland et al., 2024. The study by McClelland et al. (2024) has a very substantial dataset that allows us to verify our hypothesis purely based on this data. Which has become a major component of the paper. I hope this addresses your concern.

**Reviewer Comment**

More information on the measurement of mercury burdens in the megabenthos studies seems important to include for interpretation and standardization. Sometimes megabenthos tissues cannot be sampled consistently due to differences in exoskeleton and body form. What tissue types were measured for mercury? (e.g., whole body [including exoskeleton], internal viscera, muscle). How was inorganic mercury concentration determined? (i.e., the studies in Table 5 do not include inorganic mercury). Are the modelled concentrations for whole body of megabenthos (e.g., Figure 6)?

**Author Response**

The bioaccumulation in the model and the output in Fig. 6 is shown to be in ng Hg per mg carbon. In the model we tried to isolate the effect of the feeding strategy on bioaccumulation, so the conversion to dry weight assumes a 1:2 ratio. Most measurements indeed express bioaccumulation in dry weight and the carbon content of biota is seldom measured alongside Hg. The conversion from carbon to dry weight does indeed introduce uncertainty, but it is within reasonable levels. For example, the soft part of molluscs are for example found to have between 36% and 48% carbon per dry weight while the carbon content of artrophods (mysis mixta) was found to be 51.4% (Gorokhova & Hansson, 2000; Jurkiewicz-Karnkowska, 2005). The complications is that the study aims to look at the feeding strategy, which does very between phyla. Filter feeders can, for example, be either arthropods in the form of barnacles, molluscs in the form of bivalves, annelids in the form of of fan worms, or echinoderms in the form of brittle stars, because of this a standard 1:2 conversion ratio between carbon and dry weight was kept consistent over all feeding strategies.

We added these citations to the manuscript at the end of the model development section (line 152)

**Edit**

Line 176 Our model is resolved in carbon content, while measurements are often in dry weight. The carbon fraction of dry weight generally ranges from 0.4 to 0.6, but can vary between different taxa (Gorokhova & Hansson, 2000; Jurkiewicz-Karnkowska, 2005). To ensure consistency across different functional groups with diverse feeding strategies, we maintain a 1:2 conversion ratio for carbon to dry weight for all megabenthos functional groups.

**Reviewer Comment**

The empirical mercury data for megabenthos were pooled across geographic locations where environmental mercury exposure may have differed. How were potential confounding effects of geographic variation and feeding strategy resolved? Were the findings of feeding strategy influence on mercury burdens consistent with individual studies from specific geographic areas?

**Author Response**

I agree that this is an issue. We tried to validate our model findings using data, but data is not sufficiently available for all geographic locations to robust individual analyses. In order to address your concerns, we added the analyses solely based on the data presented (McClelland et al., 2024). Based on this, we can see that the results of our studies are consistent between the model, by pooling all geopgraphical locations and by analysing two individual locations.

**Reviewer Comment**

Organism bioaccumulation is described as involving two key processes: bioconcentration and biomagnification (lines 26-34). A more nuanced discussion is suggested here on exposure pathways and also clarification on the mechanistic processes that were modelled. Uptake of aqueous inorganic mercury and methylmercury into the food web occurs via bioconcentration in primary producers. However, consumers are typically exposed to mercury primarily through their diet and not via bioconcentration from water.

**Edit**

**Line 134**

Bioaccumulation is implemented to account for bioconcentration in all trophic levels and biomagnification in all consumers. Phytoplankton have a size-dependent uptake and release rate for the uptake and release of iHg. Based on observations by Pickhardt et al. (2006) which found higher MeHg in smaller phytoplankton but consistent iHg levels, phytoplankton have a size-dependent uptake rate and constant release rates. This means that diatoms and flagellates bioaccumulate similar amounts of iHg, while the smaller flagellates accumulate more MeHg. The uptake and release rates of iHg and MeHg in zooplankton are based on Tsui and Wang (2004) and on W. Wang and Wong (2003) for fish. An essential component of the ecosystem that interacts with bioaccumulation in megabenthos that was not overhauled for this study are the interactions between detritus and DOM and iHg and MeHg. The only  $\mathrm{Hg}^{2+}$  and  $\mathrm{MMHg}^{+}$  are assumed to partition to detritus and DOM, and this partitioning is assumed to be an equilibrium that is instantaneous and is reestimated on every time step. The equilibrium is based on the  $\mathrm{K}_{\mathrm{ow}}$  values which are based on Allison et al. (2005) and Tesán Onrubia et al. (2020). This value is  $\mathrm{log}10(6.4)$  and  $\mathrm{log}10(6.6)$  for the partitioning of  $\mathrm{Hg}^{2+}$  and  $\mathrm{log}10(5.9)$  and  $\mathrm{log}10(6.0)$  for the binding

of MMHg+ to detritus and DOM respectively. This is the same approach that is used and evaluated in Bieser et al. (2023) and Amptmeijer et al. (2025).

**Reviewer Comment**

Figure 6 shows modelled concentrations in biota, where bioconcentrated and biomagnified mercury are differentiated. These model results are not consistent with known trophic transfer processes of mercury. In higher trophic level biota, very little of the total mercury burden is inorganic mercury (e.g., in contrast with modelled result for a top predator) and most mercury is obtained from diet rather than water (bioconcentration). In Figure 6, much of the bioaccumulated mercury is attributed to bioconcentration. E.g., see Wang and Wang, Environmental Pollution 2019, Volume 252, Part B, September 2019, Pages 1561-1573, and other studies cited therein.

**Author Response**

This image stems from an admittedly awkward design choice in the paper. As stated in the paper, we aimed to enhance the model's performance over its previous version by modifying how iHg and MeHg are released during respiration. While some species have relatively high iHg concentrations, it is typically much higher and is better depicted in the Allometric Scaling (AS) model. The image presented is of the base model, which shows MeHg levels that, while in the observed range, are below average observations. Additionally, while some experimental studies such as presented by W. Wang and Wong (2003) find that bioconcentration can be a major exposure route in fish, I agree that it is typically expected to be smaller, as is the case in the AS model.

To improve this in the manuscript moved the segment about the AS model forward, and by default used graphs of the Southern North Sea AS model, as it is the best agreement with observations. In the AS model, top predators have 80-90% of tHg from MeHg from biomagnification, which is more in line with the observations by W. X. Wang and Tan (2019). Bioconcentration remains a significant route in the model, but based on W. Wang and Wong (2003) this is in line with observations. The new image is shown on line 334 and shown below for clarity.

**Reviewer Comment**

The focus of this study is on megabenthos, i.e. consumer organisms. However, a key process that warrants more modelling attention is the process of methylmercury entry in the food web via primary producers. Figure 6 shows modelling results for diatoms and dinoflagellates. How do those mercury concentrations compare with empirical data for phytoplankton? How can the inorganic mercury and methylmercury in primary producers be a result of biomagnification, as indicated in the figure?

**Author Response**

The labeling in Fig. 6, my apology that is indeed mislabeled. I will correct this as shwon in Fig. 5. I think the question surrounding the comparison with phytoplankton aligns with your question about the model validation in the technical comments section "Line 195. Provide more detail on how the model performance was evaluated.". Since the model is quite extensively evaluated for the bioaccumulation in megabenthos, I will add the following paragraph about the evaluation of the model performance in the pelagic before the evaluation of the bioaccumulation in benthos.

Figure 5: Modeled bioconcentration and biomagnification of iHg and MeHg. Partitioning to detritus and DOM is colored as bioconcentration. The y-axis is cut to show the high and low values. Notably is the high iHg to mgC ratio of detritus and DOM, leading to elevated iHg in suspension feeders. Additionally, higher trophic level animals have higher biomagnified MeHg.

**Edit**

Line 267

**4.2.1 Evaluation of the Hg cycling and pelagic bioaccumulation**

The marine cycling and speciation of Hg, in addition to the bioaccumulation in phytoplankton and zooplankton, is an essential driver of the bioaccumulation of iHg and MeHg in the benthic food web. Observed and modelled dissolved tHg concentration, the percentage of tHg that is MeHg, and the Hg content of phytoplankton and zooplankton is shown in Table 2. The concentration of dissolved tHg and the percentage of MeHg of dissolved tHg are compared to observations by Coquery and Cossa, 1995, while the bioaccumulation of tHg in phytoplankton and zooplankton is compared to observations by Nfon et al., 2009. It must be noted that the observations by Nfon et al., 2009 are not from the North Sea itself, but from the better-studied nearby Baltic Sea. The average dissolved tHg concentration is 1.7 and 2.1, pM in the Northern and Southern North Sea, respectively. This is well within 1 standard deviation of the  $1.7\pm0.7$  pM observed by Coquery and Cossa, 1995. The MeHg concentration was observed to be between 0.5 and 4.3% of tHg, with an average of 3% in the North Sea. The percentage MeHg in our model is 2.3% and 2.0% on average, which falls well within that range.

For bioaccumulation, we could not find separate reliable measurements of MeHg and iHg in phytoplankton and zooplankton in the North Sea, and we therefore evaluated the tHg content. This was measured in diatoms to be  $10\pm 5$  ng Hg mg-1. This means that the mean bioaccumulation in our model in diatoms is lower, with 5.8 ng Hg mg-1 and 9.0 ng Hg mg-1 in the Northern and Southern North Sea, respectively, but still within 1 standard deviation of the measurements. Observations labeled as zooplankton and mysis were compared to our modeled microzooplankton and mesozooplankton, respectively. All modeled values fall within 1 standard deviation of the observed tHg concentration, with

one exception: mesozooplankton in the Northern North Sea, which is 13.5% more than 1 standard deviation above the observations. This is mostly driven by a high iHg content, as the MeHg content is similar in microzooplankton and mesozooplankton.

This similarity in the MeHg content of microzooplankton and mesozooplankton in our model is caused because, even though mesozooplankton have a higher trophic level, they prefer to feed on larger diatoms which have a lower MeHg bioconcentration rate than smaller flagellates, which are preferred by microzooplankton. The high iHg content, especially in the Northern North Sea, is caused by the consumption of detritus by zooplankton in the model. While there is a shortage of data on bioaccumulation at the base of the food web, especially in the North Sea, which complicates model evaluation, the dissolved tHg concentration, the percentage of MeHg, and the tHg content of phytoplankton and zooplankton agree well with observations. With the exception of the 13.5% elevated tHg content in Northern North Sea mesozooplankton, all modeled values fall within 1 standard deviation of the observations. Because of this, we conclude that the model replicates marine Hg cycling and bioaccumulation at the base of the food web in line with observations, with the caveat that we do not have measurements of zooplankton in the Northern North Sea to verify or reject the elevated levels in that setup.

Table 2: Dissolved tHg (pM), MeHg (% of tHg), and tHg concentrations in biota (ng Hg mg-1 d.w.) across North Sea regions.

|                                                 | Observed        | NNS             | SNS             |
|-------------------------------------------------|-----------------|-----------------|-----------------|
| tHg dissolved (pM)                   | $1.7\pm0.7$     | $1.7 \pm 0.26$  | $2.0 \pm 0.28$  |
| MeHg (% of tHg)                                 | 3 (0.5-4.3)     | $2.3 \pm 0.23$  | $2.0 \pm 0.31$  |
| Diatoms tHg (ng Hg $mg^{-1}$ )                  | $10 \pm 5$      | $7.0\pm1.1$     | $8.3\pm1.6$     |
| Flagellates tHg (ng Hg mg -1 )       |                 | $13.9 \pm 3.0$  | $14.3 \pm 3.0$  |
| ${ m Microzooplankton~tHg~(ng~Hg~mg^{-1})}$     | $37.5 \pm 31.3$ | $67.4 \pm 29.3$ | $40.3 \pm 11.4$ |
| Microzooplankton MeHg (ng Hg mg -1 ) |                 | $7.1 \pm 2.1$   | $10.5 \pm 2.7$  |
| $Mesozooplankton tHg (ng Hg mg^{-1})$           | $62.5 \pm 12.5$ | $86.7 \pm 15.1$ | $72.3 \pm 19.6$ |
| Mesozooplankton MeHg (ng Hg mg -1 )  |                 | $6.9 \pm 2.6$   | $10.5 \pm 1.7$  |

**5 Technical Corrections**

**Reviewer Comment**

Line 8. Is the inorganic mercury in filter feeders elevated, or more specifically, is it found as a higher proportion of total mercury compared to other megabenthos?

**Author Response**

No, filter feeders do not have elevated iHg levels. The iHg levels in filter feeders are large part of tHg, but filter feeders have similar iHg as most other macrobenthos but elevated MeHg levels, thus a reduced proportion of iHg of tHg. The main megabenthos group that has notably increase in iHg are the suspension feeders. They are defined by their ability to eat DOM (resembling sponges), where filter feeders cannot filter out dissolved particles. Suspension feeders on the other hand, have both elevated iHg levels and an elavated iHg to tHg ratio. This is true both in the model and in the observation in Meditereanean Sea sponges by Orani et al. (2020).

**Reviewer Comment**

Line 18. Cite a reference for this statement.

**Edit**

Line 18

Anthropogenic emissions have significantly raised environmental Hg levels, with 78%, 85%, and 50% of atmospheric, upper ocean, and deep ocean Hg, respectively, originating from anthropogenic emissions (Geyman et al., 2025).

**Reviewer Comment**

Line 26-27. Does bioconcentration only occur in a polluted environment? Is the model then only relevant for polluted environments?

**Edit**

Line 27

There are two key processes involved in bioaccumulation: bioconcentration and biomagnification. When animals absorb Hg directly from their environment; this is called bioconcentration.

**Reviewer Comment**

Line 45. The first effectiveness evaluation of the Minamata Convention has not been completed. Rephrase this text.

**Edit**

Line 51

While there is considerable understanding of MeHg bioaccumulation in high trophic levels, less is known about the bioaccumulation drivers at the base of the food web where Hg concentrations tend to be lower, resulting in reduced risk to humans. As such, these organisms are not prioritized in the current monitoring strategies under the ongoing effectiveness evaluation of the Minamata Convention, which focuses primarily on fish, humans, and predatory wildlife Evers2016EvaluatingSteps. Additionally, the evaluation to date has shown that Hg and MeHg concentrations in water and sediment do not correlate well with levels in biota, leading to greater emphasis on biological monitoring over abiotic compartments.

**Reviewer Comment**

Line 51-52. Some megabenthos are not lower trophic level biota (e.g., are secondary consumers) and thus are not necessarily representative of processes at the base of the food web.

**Edit**

Line 60

The benthic food web is highly complex, making it challenging to improve our understanding of bioaccumulation within it (Silberberger et al., 2018).

**Reviewer Comment**

Line 74. Perhaps change "in silico" to "a modelling experiment"

**Edit**

Line 83

Afterward, we conducted a modeling experiment in which megabenthos with various feeding strategies compete under physical drivers in idealized scenarios that are typical of megabenthos-rich coastal oceans.

**Reviewer Comment**

Line 86-89 and elsewhere. Use past tense to describe work that was completed.

**Author Response**

I will change that anywhere in the manuscript. For example in line 80 it is refrased as:

**Edit**

To compare the findings with the literature, we collected field studies measuring Hg in megabenthos. The studies we used are shown in Table S1. We categorized the megabenthos into the same feeding categories, "deposit feeder", "filter feeder", "suspension feeder", "grazer", and "predator". To better look at the effect of the trophic level, we also added "primary producers" as the base of the food web, and "seabird" and "benthic fish" as top predators. We analyzed whether trophic level and feeding strategy influence megabenthos iHg, MeHg, and/or tHg content. The observations were analyzed by visualizing the data, performing linear regression, and plotting a correlation matrix of the differences in bioaccumulation between different feeding strategies. The total and partial R2 of the linear regression of the trophic level and the feeding strategy were compared to analyze the effect of both drivers on bioaccumulated iHg, MeHg, and tHg.

**Reviewer Comment**

Figure 1. What is the black line that connects biota to detritus, DOM and sediment organic carbon?

**Author Response**

I will update the caption of the figure as stated in Fig. 6. I also added a dotted black line to show mortality from phytoplankton and zooplankton.

**Reviewer Comment**

Line 186. Rephrase "samples are sampled"

**Edit**

Line 116

Because of this, most samples are from shallow, well-mixed coastal areas, and we used this setup to evaluate the performance of the models.

**Reviewer Comment**

Section 3.1. How do the results of this analysis compare with published findings reported in the literature?

**Author Response**

In the literature there is relatively little attention to direct role of feeding strategy or phyla on bioaccumulation. Several individual papers remark things such as that mussels have higher values than crustaceans but I could not find our main connclusion supported in the literature. That being said, I hope the expansian of the paper by doing the further analyses with the data presented by McClelland et al. (2024) is convincing that while this interaction itself is not supported as a conclusion in literature studies, published data does support our conclusions.

**Reviewer Comment**

Line 255. Unclear meaning – "validate that they survive in the model"

**Edit**

Line 296

While our megabenthos groups only vary in their feeding strategies and lack a direct real-world counterpart, it is important to ensure that all functional groups have consistent biomass in the model and thus the results originate from the modeled interactions, and are not altered due to unrealistically high or low biomass in the model.

**Reviewer Comment**

Figure 6. How do mercury concentrations per unit carbon (reported in Figure 6) compare to literature reported values that do not take carbon content into account?

Figure 6: The overview of the modeled megabenthos functional groups and how they interact with each other and functional groups in the ECOSMO E2E model. There are 5 macrobenthic functional groups. The filter feeder feeds on pelagic detritus, zooplankton, and phytoplankton. The suspension feeders feed on pelagic detritus, phytoplankton, zooplankton, and DOM. The generalist feeds on phytoplankton, zooplankton, pelagic detritus, and sediment organic carbon. The deposit feeder feeds on sediment organic carbon. The benthic predator feeds on the other 4 megabenthos functional groups and the top predator solely feeds on the benthic predator. The arrows indicate trophic interactions where the arrow goes from the prey to the predator and the arrows have the same colour as the prey. The black lines represent loss of organic material due to mortality. When megabenthos die, their organic carbon is transferred to pelagic DOM and detritus, as well as the sediment, shown by the solid black arrow. In contrast, when pelagic organisms die, their organic carbon is transferred to DOM and detritus, indicated by the dotted black arrow.

**Author Response**

As described in above, the conversion factor of carbon to dry weight is assumed to be a 1:2 ratio for all functional groups. This conversion is indeed a source of uncertainty. I expanded on this in the suggested expansion of the paper at the beginning of the revieuw, Additionally we added a segment on line 516 where we discuss data related uncertainty.

**Reviewer Comment**

Line 310-313. Are there published empirical studies that support this model prediction regarding filter feeders?

**Author Response**

There is published emperical data, as described at the beginning of these answer, that supports our conclusion. But no studies directly comparing filter feeders in the way we did in this study with other feeding strategies. So our conclusion comes from the combination of having measurements that show elevated MeHg levels in filter feeders combined with a modeling explanation as why this might be because they are filter feeders.

**Reviewer Comment**

Line 361. Does the bioaccumulated inorganic mercury originate from water or dietary exposure?

**Author Response**

The majority of this in all catagories except suspension feeders is via bioconcentration, thus uptake from the water. We added the following to clarify this:

**Edit**

Line 489

In Fig. 6 we can see that the vast majority of iHg in filter, deposit, and generalist feeders originates from bioconcentration, thus direct uptake from the water is the dominant pathway of iHg bioaccumulation in our model in these feeding strategies.

**Reviewer Comment**

Line 394. Explain further what is meant by "in vivo mercury speciation".

**Author Response**

We removed the segment about in vivo mercury speciation. It is not the focus on this paper, and either we would need to long to explain it properly or it does not really contribute to this dicussion as it is not relevant for most of the animals we looked at.

**Reviewer Comment**

Line 411. Where are these regression results presented earlier in the manuscript?

**Author Response**

They are from Table 3 where the model is evaluated, but I can better describe the table outcome in text. I would add the following part to the end of section 3.3. where I discuss the results of the allometric scaling model.

**Edit**

Line 335

Our base model does agree well with both observed iHg ( $R^2$ =0.84) and MeHg ( $R^2$ =0.86) in the Southern North Sea setup, but this is mostly driven by accurate predictions in the lower trophic levels while there is a normalized bias of -0.84 in the Top Predators. This is improved dramatically in the allometric scaling model with the reduction of the normalized bias of top predators to -0.32 which improves the overal  $R^2$  of the model to >0.99.

**Reviewer Comment**

Line 436-437. This comment about bivalve communities is speculative.

**Reviewer Comment**

Line 441-442. Consider concluding the paper with a recommendation for future work on model development.

**Author Response**

I will answer these comments together by suggesting a rewrite of the closing remarks. I agree that it is speculative, but I do think, especially considering the addition of the further analyses that also point to molluscs as being higher in MeHg, that it is a logical next step to take into account based on the results of this study. We rephrased it as follows to make it clear that our work should be seen as hypothesis-generating modeling work that suggests investigating the potential relevance of benthic community for MeHg bioaccumulation might be relevant.

**Edit**

Our findings suggest that fish from food webs dominated by filter feeders would have the highest MeHg content, since filter feeders have the highest MeHg content in both our model and observations. It also creates an indication that the introduction of bivalve communities in the form of mussel or oyster farming could increase MeHg levels in higher food chains. However, such changes in the ecosystem would inevitably change other factors in the ecosystem, including biomass and trophic interactions that are also essential drivers for MeHg bioaccumulation. While our model should be seen as a hypothesis-generating work that requires empirical validation, it does suggest that case-by-case studies are needed to fully understand how changes in the base of the food web will affect the concentration of MeHg in high trophic level fish.

We strongly recommend targeted field studies that systematically measure iHg, MeHg, and trophic levels in diverse marine communities to assess how the structure of the food web influences the bioaccumulation of MeHg in seafood.

**References**

- Allison, J. D., Allison, T. L., & Ambrose, R. B. (2005). ALLISON, J. D. AND T. L. ALLISON.

  PARTITION COEFFICIENTS FOR METALS IN SURFACE WATER, SOIL, AND
  WASTE. U.S. Environmental Protection Agency, Washington, DC (tech. rep.).
- Amptmeijer, D. J., Bieser, J., Mikheeva, E., Daewel, U., & Schrum, C. (2025). Bioaccumulation as a driver of high MeHg in coastal Seas. *EGUsphere* [preprint].
- Bieser, J., Amptmeijer, D., Daewel, U., Kuss, J., Soerenson, A. L., & Schrum, C. (2023). The 3D biogeochemical marine mercury cycling model MERCY v2.0; linking atmospheric Hg to methyl mercury in fish. *Geoscientific Model Development Discussions*, 1–59.
- Blinick, N. S., Link, D., Ahrenstorff, T. D., Bethke, B. J., Fleishman, A. B., Janssen, S. E., Krabbenhoft, D. P., Nelson, J. K. R., Rantala, H. M., Rude, C. L., & Hansen, G. J. A. (2024). Increased mercury concentrations in walleye and yellow perch in lakes invaded by zebra mussels.
- Bradford, M. A., Mallory, M. L., & O'Driscoll, N. J. (2024). Ecology and environmental characteristics influence methylmercury bioaccumulation in coastal invertebrates. *Chemosphere*, 346, 140502.
- Coquery, M., & Cossa, D. (1995). Mercury speciation in surface waters of the north sea. *Netherlands Journal of Sea Research*, 34 (4), 245–257.
- Dutton, J., & Fisher, N. S. (2012). Bioavailability of sediment-bound and algal metalsto killifish Fundulus heteroclitus. *Aquatic biology*, 16, 85–96.
- Geyman, B. M., Streets, D. G., Olson, C. I., Thackray, C. P., Olson, C. L., Schaefer, K., Krabbenhoft, D. P., & Sunderland, E. M. (2025). Cumulative Anthropogenic Impacts of Past and Future Emissions and Releases on the Global Mercury Cycle. *Environmental Science and Technology*, 59(17), 8578–8590.
- Gorokhova, E., & Hansson, S. (2000). Elemental composition of Mysis mixta (Crustacea, Mysidacea) and energy costs of reproduction and embryogenesis under laboratory conditions.

  Journal of Experimental Marine Biology and Ecology, 246(1), 103–123.
- Jurkiewicz-Karnkowska, E. (2005). Some Aspects of Nitrogen, Carbon and Calcium Accumulation in Molluscs from the Zegrzyński Reservoir Ecosystem. *Polish Journal of Environmental Studies*, 14(2), 173–177.
- Lee, C. S., & Fisher, N. S. (2016). Methylmercury uptake by diverse marine phytoplankton. Limnology and Oceanography, 61(5), 1626–1639.
- McClelland, C., Chételat, J., Conlan, K., Aitken, A., Forbes, M. R., & Majewski, A. (2024). Methylmercury dietary pathways and bioaccumulation in Arctic benthic invertebrates of the Beaufort Sea. *Arctic Science*, 10(2), 305–320.
- Nfon, E., Cousins, I. T., Järvinen, O., Mukherjee, A. B., Verta, M., & Broman, D. (2009). Trophodynamics of mercury and other trace elements in a pelagic food chain from the Baltic Sea.
- Orani, A. M., Vassileva, E., Azemard, S., & Thomas, O. P. (2020). Comparative study on Hg bioaccumulation and biotransformation in Mediterranean and Atlantic sponge species. *Chemosphere*, 260, 127515.
- Outridge, P. M., Mason, R. P., Wang, F., Guerrero, S., & Heimbürger-Boavida, L. E. (2018, October). Updated Global and Oceanic Mercury Budgets for the United Nations Global Mercury Assessment 2018.
- Pickhardt, P. C., Stepanova, M., & Fisher, N. S. (2006). Contrasting uptake routes and tissue distributions of inorganic and methylmercury in mosquitofish (Gambusia affinis) and redear sunfish (Lepomis microlophus). *Environmental Toxicology and Chemistry*, 25(8), 2132–2142.
- Schartup, A. T., Qureshi, A., Dassuncao, C., Thackray, C. P., Harding, G., Sunderland, E. M., Harvard, †., & Paulson, J. A. (2018). A Model for Methylmercury Uptake and Trophic Transfer by Marine Plankton. *Environ. Sci. Technol*, 52, 18.

- Silberberger, M. J., Renaud, P. E., Kröncke, I., & Reiss, H. (2018). Food-web structure in four locations along the European shelf indicates spatial differences in ecosystem functioning. *Frontiers in Marine Science*, 5(APR), 300569.
- Tesán Onrubia, J. A., Petrova, M. V., Puigcorbé, V., Black, E. E., Valk, O., Dufour, A., Hamelin, B., Buesseler, K. O., Masqué, P., Le Moigne, F. A., Sonke, J. E., Rutgers Van Der Loeff, M., & Heimbürger-Boavida, L. E. (2020). Mercury Export Flux in the Arctic Ocean Estimated from 234Th/238U Disequilibria. ACS Earth and Space Chemistry, 4(5), 795–801
- Tsui, M. T., & Wang, W. X. (2004). Uptake and Elimination Routes of Inorganic Mercury and Methylmercury in Daphnia magna. *Environmental Science and Technology*, 38(3), 808–816
- Wang, W. X., & Tan, Q. G. (2019). Applications of dynamic models in predicting the bioaccumulation, transport and toxicity of trace metals in aquatic organisms. *Environmental Pollution*, 252, 1561–1573.
- Wang, W., & Wong, R. (2003). Bioaccumulation kinetics and exposure pathways of inorganic mercury and methylmercury in a marine fish, the sweetlips Plectorhinchus gibbosus.

  Marine Ecology Progress Series, 261.

---

## Author Response (AR2)

**1 Answer to the Associate editor decision**

Thank you for taking the time to give me suggestions to improve the writing. Below I will comment on how I addressed these specific concerns. In line with your suggestions, I gave the whole document a thorough proofread and, as such, I have attached a tracked change version below this message. I hope this makes it easy to scan all potential issues that have been addressed. The only other change I made in addition to proofreading is that I added the code availability section. The code is currently being published, so it should be downloadable via that DOI in the next days. I hope that the proofreading makes the manuscript in line with your expectations.

**Associate editor Comment**

Line 304: should be a semicolon after "Fig. 3"

**Edit**

Line 304

Figure 3 shows the modeled bioaccumulation in the AS model in the Southern North Sea; note that the values are expressed in ng Hg mg C-1, as this is the best proxy in our model to show the dietary uptake of Hg per unit of energy and nutrients consumed.

**Associate editor Comment**

Line 312: Should be a comma after "In Fig. 4a, c, and e".

**Edit**

Line 315

In Fig. 4a, c, and e, we show the relationship between the trophic level and the bioaccumulation of iHg, MeHg, and tHg in megabenthos.

**Associate editor Comment**

Line 314: Should read "trophic level" (singular) and not "trophic levels" (plural). Similarly, the verb should be in the singular form and written as "...that is not present" instead of "...that are not present".

**Edit**

Line 315

There is an increase in the MeHg content with trophic level, which is not present for iHg

**Associate editor Comment**

Line 314: Please specify what parameters the anti-correlation are between to clarify the meaning.

**Edit**

Line 315

There is a weak anti-correlation ( $R^2 = 0.20$ ) between the bioaccumulation of iHg and the trophic level.

**Associate editor Comment**

Figure 3 caption" should read "notable" instead of "notably". Instead of "of detritus and DOM," I believe it should read "in detritus and DOM," to clarify the meaning. In addition, there should be a period at the end of the caption.

**Edit**

Caption Fig. 3

Notable is the high iHg to mgC ratio associated with detritus and DOM, leading to elevated iHg in suspension feeders.

**Associate editor Comment**

Line 319: Would improve the flow and clarity to add "and" before "taking into account".

**Author Response**

I change the sentence structure to active voice to make it smoother as below.

**Edit**

Line 321

In Table 2, we show the results of a linear regression using the global dataset while accounting for both the trophic level and the feeding strategy; each model's relative fit explains Hg bioaccumulation based on both factors

**Associate editor Comment**

Line 321 states "This shows that we can explain the bioaccumulation of ln(MeHg) very well" but it is not clear what "this" is referring to. Please clarify the writing.

**Associate editor Comment**

Line 323: Please clarify what is meant by an average fit.

**Author Response**

I am sorry that was indeed vague and undevined. I updated that as below to intermediate explanatory power  $(R^2=0.46)$

**Edit**

Line 323

These linear regressions show that the bioaccumulation of MeHg can be predicted very well ( $R^2$ =0.72) with a linear model that takes both drivers into account, while iHg is poorly explained ( $R^2$ =0.11) and tHg shows intermediate explanatory power ( $R^2$ =0.46).

**Associate editor Comment**

Line 327: In the sentence "this still shows that the partial...", please clarify in the writing what "this" is referring to.

**Author Response**

That is update to "These linear regressions" for clarity in the setence above

**Associate editor Comment**

Line 331: In the sentence "This shows that if we...", please clarify in the writing what "this" is referring to.

**Edit**

Line 333

Table 3 shows that if we take the allometric scaling law into account, the model results for high-trophic-level animals improve considerabl

**Associate editor Comment**

Line 333: Add a comma after "In Fig. 4b, d, and e".

**Edit**

Line 334

In Fig. 4b, d, and e, we show the relationship between the natural logarithm of bioaccumulation and the trophic level of the AS model in the Southern North Sea setup

**Associate editor Comment**

Please note that the above list is meant to provide examples of the types of editing required and is not an exhaustive list. A thorough review and editing of the manuscript to improve clarify and decrease typos is suggested, with focus on the Results and Discussion. I look forward to seeing the revised version of the manuscript, which I believe will be ready for publication after some minor editing.

**Author Response**

I went over the manuscript and there were indeed a number of typos and unclear sentences in the manuscript. I split several sentences into smaller sentences to increase flow and increase clarity by not using unclear reference terms such as this. Below is a tracked changed version of the paper to demonstrate in detail where changes where made.

**Feeding strategy as a key driver of the bioaccumulation of MeHg in megabenthos**

David J. Amptmeijer1, Andrea Padilla2, Sofia Modesti2, Corinna Schrum1, 2, and Johannes Bieser1

Correspondence: David J. Amptmeijer (davidamptmeijer@gmail.com)

Abstract. The bioaccumulation of methylmercury (MeHg) in the marine food chain poses a neurotoxic risk to human health, especially through the consumption of seafood. Although MeHg bioaccumulation at higher trophic levels is relatively well understood, MeHg bioaccumulation at the base of the food web remains underexplored. Given the neurotoxic effects of methylmercury MeHg on human health, it is essential to understand the drivers of bioaccumulation at every level of the food chain. In this study, we incorporate We coupled six megabenthos functional groups into the ECOSMO marine in the ECOSMO end-to-end ecosystem model, coupled to the MERCY marine v2.0 Hg cycling model. We investigated how various feeding strategies influence the bioaccumulation of both inorganic Hg (iHg) and MeHg in marine ecosystems. We show that the feeding strategy significantly influences bioaccumulation and correlates stronger more strongly with iHg than the trophic leveland that does trophic level. In particular, suspension feeders have elevated iHg levels, while filter feeders have higher MeHg valueselevated MeHg levels compared to other megabenthos. Additionally, we show that feeding strategies alone allow us to accurately model the bioaccumulation of both iHg and MeHg can be accurately modeled solely based on feeding strategies in low trophic-level in low-trophic-level megabenthos. However, when modeling higher trophic levels, incorporating the allometric scaling law dramatically improves the substantially improves model performance. These results demonstrate the need for a holistic approach in which iHg, MeHg, and trophic levels the trophic level of organisms are evaluated at both high and low trophic levels to identify what food web structures drive high MeHg concentrations in seafood.

**1 Introduction**

Mercury (Hg) is a naturally occurring element. In addition to its natural occurrence, it is also emitted through various anthropogenic activities, such as the burning of fossil fuels, small-scale artisanal gold mining, and the production of cement and ferrous metals (Pacyna et al., 2006). These anthropogenic emissions have significantly raised environmental Hg levels, with 78%, 85%, and 50% of atmospheric, upper ocean, and deep ocean Hg, respectively, originating from anthropogenic emissions (Geyman et al., 2025).

When elemental Hg (Hg0Hg0) is emitted, it can undergo long-range atmospheric transport. In this way, it can be transported on a global scale and deposited, allowing global dispersion and subsequent deposition in the oceans, thus increasing Hg levels in the marine environment (Durnford et al., 2010). Marine Hg0 is volatile and can return to the atmosphere or be

<sup>1Matter Transport and Ecosystem Dynamics, Helmholtz-Zentrum Hereon, Geesthacht, Germany

<sup>2Universität Hamburg, Institute for Marine Sciences, Mittelweg 177, 20146 Hamburg, Germany

oxidized into dissolved Hg (Hg2+) (Sommar et al., 2020). This Hg2+ can be reduced back to volatile elemental Hg0, 
[revised manuscript text omitted]
<del>for the</del> uptake and release of iHg. Based, based on observations by Pickhardt et al. (2006)that, which found higher MeHg in smaller phytoplankton but consistent iHg levels, phytoplankton have. As such, different phytoplankton groups are implemented with a size-dependent uptake rate and and release rate for iHg but only size-dependent uptake rates with constant release rates -This means that for MeHg. This results in diatoms and flagellates bioaccumulating similar amounts of iHg, while the smaller flagellates accumulate more MeHg compared to larger diatoms. The uptake and release rates of iHg and MeHg in zooplankton are based on Tsui and Wang (2004) and on Wang and Wong (2003) for fish. An essential component of the ecosystem that interacts with bioaccumulation in megabenthos that was not overhauled for this study is the interactions between detritus and DOM and iHg and MeHg. The only Hg species assumed to partition to DOM an and detritus are Hg2+ and MMHg+, and this partitioning is assumed to be an equilibrium that is instantaneous and is reestimated on every time step. The equilibrium is based on the Kparticle partitioning coefficient for organic matter Kdwd values which are based on Allison et al. (2005) and Tesán Onrubia et al. (2020). This value is  $log_{10}(6.4)$  and  $log_{10}(6.6)$  for the partitioning of  $Hg^{2+}$  and  $log_{10}(5.9)$  and log10(6.0) for the binding of MMHg+ to detritus and DOM respectively. This is the same approach that is used and evaluated in Bieser et al. (2023).

**2.2 Model development**

To use the model to study bioaccumulation in megabenthos, the higher trophic level of the ECOSMO E2E model is altered. We exchanged the functional group replaced the macrobenthos, fish 1, and fish 2 with 6 functional groups with six megabenthos functional groups, as shown in Fig. 1. The megabenthos groups are separated by their feeding strategy: filter feeder, deposit feeder, generalist feeder, suspension feeder, predator, and top predator.

Filter feeders filter suspended particles from the water column. In our model, they can eat phytoplankton, zooplankton, and detritus. Examples of filter feeders are mussels, tubeworms, and barnacles. The second group is **deposit feeders**. These animals consume organic carbon from the sediment; in our model, they exclusively feed on organic carbon deposited in the sediment. This group would include includes gastropods and polychaete worms, such as the lugworm (*Arenicula marina*). The **generalist feeder resembles** includes animals such as brown shrimp (*Crangon crangon*), which can utilize various feeding strategies. In our model, this group feeds on phytoplankton, zooplankton, detritus, and deposited material. We also include a **suspension feeder**. Suspension feeders, such as sponges, can consume detritus and DOM. The consumption of DOM, which is too small to be consumed by filter feeders, differentiates suspension and filter feeders. A common strategy to consume DOM as a food source is the utilization of symbiotic bacteriasuch as mechanism for consuming DOM involves symbiotic bacteria, as seen in chemosymbiotic bivalves from the families Lucinidae, Solemyidae, and Thyasiridae, and microbial biomes of high microbial assemblage sponges (Dufour, 2018; Olinger et al., 2021). Finally, we included 2-two predators. The first predator is referred to as **the predator**, it. The predator feeds on the 4-four benthic groups mentioned above , and it and has an equal preference and grazing rate in all groups, but it will prioritize abundant groups. This preference is caused implemented by making the food available for predation by the predators not linearly related to the abundance of the prey, but calculated as:

$$b_{ ext{available}} = egin{cases} b_{ ext{biomass}}, & ext{if } b_{ ext{biomass}} \geq b_{ ext{protected}}, \ b_{ ext{biomass}} rac{b_{ ext{biomass}}}{b_{ ext{protected}}}, & ext{if } b_{ ext{biomass}} < b_{ ext{protected}}. \end{cases}$$

in which,

175

180

160

- $b_{\text{available}}$ : Portion of prey biomass in g C m-2 accessible to predators.
- $b_{\text{protected}}$ : Level of prey biomass in g C m-2 below which hunting becomes less optimal or energetically inefficient.
- $b_{\text{biomass}}$ : Total prey biomass in g C m-2 in the environment.

The megabenthos in the North Sea are estimated to have between 1.1 and 35.5 gC m-2 (Heip et al., 1992; Daan and Mulder, 2001). The value for  $B_{Protected}$  is chosen as 1 gC m-2 for all megabenthos except for the benthic predator where  $B_{Protected}$  is 0.5 gC m-2. These values are chosen to protect megabenthos functional groups from extinction due to predation when their values are below the expected range. This relationship models 2-two real-world interactions. First, when the concentration of prey is low, the small number of individuals can more likely survive under ideal circumstances and, therefore, may be less exposed to predation (Campanella et al., 2019). Secondly, several predators, such as the shore crab, adapt their behaviors to the density

Figure 1. The overview of the modeled megabenthos functional groups and how they interact with each other and functional groups in the ECOSMO E2E model. There are 5-six megabenthic functional groups. The filter feeder feeds on pelagic detritus, zooplankton, and phytoplankton. The suspension feeders feed on pelagic detritus, phytoplankton, zooplankton, and DOM. The generalist feeds on phytoplankton, zooplankton, pelagic detritus, and sediment organic carbon. The deposit feeder feeds on sediment organic carbon. The benthic predator feeds on the other 4-four megabenthos functional groups and the top predator solely feeds on the benthic predator. The arrows indicate trophic interactions where the arrow goes from the prey to the predator and the arrows have the same colour as the prey. The black lines represent loss of organic material due to mortality. When megabenthos die, their organic carbon is transferred to pelagic DOM and detritus, as well as the sediment, shown by the solid black arrow. In contrast, when pelagic organisms die, their organic carbon is transferred to DOM and detritus, indicated by the dotted black arrow. Several sub-images have been used in this image. Sources of the images: Filter feeder: Sabella spallanzanii (photo by Diego Delso, CC BY-SA 4.0, via Wikipedia), Suspension feeder: Aplysina fistularis (photo by Twilight Zone Expedition Team 2007, NOAA-OE, CC BY 2.0, via Flickr), Generalist feeder: Crangon crangon (photo by Etrusko25, Public Domain, via Wikipedia), Deposit feeder: Buccinum undatum (photo by Oscar Bos / Ecomare, CC BY 4.0, via Wikipedia), Benthic predator: Hommarus gammarus (photo by Bart Braun, Public Domain, via Wikipedia), Top predator: Sepia officinalis (photo by Nick Hobgood, CC BY-SA 3.0, via Wikipedia).

of the prey and learn to be more efficient in the hunting of more common prey (Chakravarti and Cotton, 2014). Our model is resolved in carbon content, while measurements are often in dry weight. The carbon fraction of dry weight generally ranges from 0.4 to 0.6, but can vary between different taxa (Gorokhova and Hansson, 2000; Jurkiewicz-Karnkowska, 2005). To ensure consistency across different functional groups with diverse feeding strategies, we maintain a 1:2 conversion ratio for carbon to dry weight for all megabenthos functional groups.

**2.2.1 Assimilation efficiency of iHg and MeHg**

The assimilation efficiency (AE) of iHg and MeHg is a key parameter in correct biomagnification modeling. AE is based on laboratory experiments that analyze AE in phytoplankton (Metian et al., 2020; Wang and Wong, 2003). An assimilation efficiency of 0.95 for MeHg and 0.31 for iHg is chosen for everything except deposit feeding, which has a lower feeding efficiency of 0.07 for iHg and 0.43 for MeHg according to Dutton and Fisher (2012).

**2.2.2 Semi-labile DOM**

In the ECOSMO E2E model, only labile-DOM is resolved. This means that there is very little DOM. In our model, we want to incorporate a suspension feeder that would utilize DOM as a food source. Because of this, we added a DOM component referred to as semi-labile DOM. This semi-labile DOM has the same bacterial degradation rate as that of the detritus, and it has the same Hg partitioning behavior as labile DOM. When organic carbon (detritus + labile-DOM + semi-labile-DOM) is formed, 5% is formed as semi-labile DOM, and there is a breakdown of the detritus into semi-labile DOM of 0.001 d-1 (per day). Since the categorization of DOM is very complex, these rates are estimated to create a low maximum of 50 mg C m-3. This is lower than the DOM concentrations typically found in the North Sea, but because it is unclear which fraction of DOM can be consumed by suspension feeders, this amount provides suspension feeders a unique food source that they can utilize while not outcompeting other megabenthos (Lønborg et al., 2024).

**2.2.3 Allometric scaling model**

Finally, we run the model while taking into account other ran the model incorporating additional drivers of MeHg bioaccumulation to see whether it improves the model evaluate whether they improved model performance. There are three interactions that we take into account for this second model. First, the allometric scaling law, which states that larger animals have a lower base metabolic rate when normalized to body weight (da Silva et al., 2006). Secondly, we account for the observations that MeHg bioaccumulation in fish increases as the water temperature increases, indicating that increased activity does not increase MeHg excretion while it increases MeHg uptake due to a higher grazing rate (Dijkstra et al., 2013). Finally, we assume that predators need to spend more energy on active metabolism to hunt their prey. Because of this, we assumed that the total relative respiration rate of for predators and top predators is not altered, so both models have the same carbon cycle. However, MeHg is excreted at a lower rate of we assumed a MeHg excretion rate (0.002 d-1, rather than their respiration rate, which is the same base metabolic rate as the fish in the ECOSMO E2E model), 
[revised manuscript text omitted]
 we therefore evaluated the tHg content. This was measured in diatoms to be 10±5 ng Hg mg-1. This means that the instead. The mean bioaccumulation in our model in diatoms is lower, with 5.8 ng Hg mg-1 and 9.0 ng Hg mg-1 in the Northern and Southern North Sea, respectively, but still within tone standard deviation of the measurements. Observations labeled as zooplankton and mysis were compared to our modeled microzooplankton and mesozooplankton, respectively. All modeled values fall within tone standard deviation of the observed tHg concentration, with one exception: mesozooplankton in the Northern North Sea, which is 13.5% more than tone standard deviation above the observations. This is mostly driven by a high iHg content, as the MeHg content is similar in microzooplankton and mesozooplankton.

This similarity in the MeHg content of MeHg content between microzooplankton and mesozooplankton in our model is eaused arises because, even though mesozooplankton have occupy a higher trophic level, they prefer to preferentially feed on larger diatomswhich. These diatoms have a lower MeHg bioconcentration rate than smaller flagellates, which are the smaller flagellates preferred by microzooplankton. The high iHg content, especially in the Northern North Sea, is caused by the consumption of detritus by zooplankton in the model. While there is a shortage of data on bioaccumulation at the base of the food web, especially in the North Sea, which complicates model evaluation, the dissolved tHg concentration, the percentage of MeHg, and the tHg content of phytoplankton and zooplankton agree well with observations. With the exception of the 13.5% elevated tHg content in Northern North Sea mesozooplankton, all modeled values fall within 1 one standard deviation of the observations. Because of this, we conclude that the model replicates marine Hg cycling and bioaccumulation at the base of the food web in line with observations, with the caveat that we do not have measurements of zooplankton in the Northern North Sea to verify or reject the elevated levels in that setup.

**Table 1.** Dissolved tHg (pM), MeHg (% of tHg), and tHg concentrations in biota (ng Hg mg-1 d.w.) across North Sea regions in observations and the modeled setups. The observations of aquatic tHg and % MeHg are from Coquery and Cossa (1995) and the bioaccumulation in biota is compared to observations by Nfon et al. (2009).

|                                                  | Observed        | NNS             | SNS             |
|--------------------------------------------------|-----------------|-----------------|-----------------|
| tHg dissolved (pM)                    | $1.7\pm0.7$     | $1.7 \pm 0.26$  | $2.0 \pm 0.28$  |
| MeHg (% of tHg)                                  | 3 (0.5–4.3)     | $2.3 \pm 0.23$  | $2.0\pm0.31$    |
| Diatoms tHg (ng Hg mg -1 )            | 10 ± 5          | $7.0 \pm 1.1$   | $8.3 \pm 1.6$   |
| Flagellates tHg (ng Hg mg -1 )        |                 | $13.9 \pm 3.0$  | $14.3 \pm 3.0$  |
| Microzooplankton tHg (ng Hg mg -1 )   | $37.5 \pm 31.3$ | $67.4 \pm 29.3$ | $40.3 \pm 11.4$ |
| Microzooplankton MeHg (ng Hg mg -1 )  |                 | $7.1\pm 2.1$    | $10.5 \pm 2.7$  |
| Mesozooplankton tHg (ng Hg mg -1 )    | $62.5 \pm 12.5$ | $86.7 \pm 15.1$ | $72.3 \pm 19.6$ |
| Mesozooplankton MeHg (ng Hg $\mathrm{mg}^{-1}$ ) |                 | $6.9 \pm 2.6$   | $10.5 \pm 1.7$  |

**310 3.1.2 Megabenthic biomass**

While our megabenthos groups only vary differ in their feeding strategies and lack a direct real-world counterparts, it is important to ensure that all functional groups have consistent biomass in the modeland thus. This guarantees that the results originate from the modeled interactions, and are not altered due to rather than being influenced by unrealistically high or low modeled biomass. The biomass. We show the yearly progression of the megabenthos biomass is shown in Fig. 2. Filter feeders have the highest biomass, which is up to 10 g C m-2 followed by deposit feeders with up to 5 g C m-2, generalist feeders with up to 3 g C m-2, and suspension feeders with up to 1 g C m-2. Higher trophic levels have lower biomass, with up to 0.2 g C m-2 for the predator and 0.5 g C m-2 for the top predator. This shows that after a simulation period of 20 years, all megabenthos have a stable population, while biomass is highest at the base of the food web.

**3.2 Bioaccumulation in the model**

The Figure 3 shows the modeled bioaccumulation in the AS model in the Southern North Seais shown in Fig. 3,; note that the values are expressed in ng Hg mg C-1, as this is the best proxy in our model to show the dietary uptake of Hg per unit of energy and nutrients consumed. There is a very high concentration of iHg in the sediment, detritus, and DOM. These values are 0.60, 1.1, and 2.6 ng Hg mg C-1 for iHg and 0.089, 0.0067, and 0.012 ng Hg mg C-1 for MeHg respectively. The high amount of iHg in organic carbon is in line with observations that found values of up to 0.114-1.192 ng Hg mg d.w. in sediment in sediments from the Scheldt estuary and that DOM strongly binds up to 1.0 ng Hg mg-1 (Zaferani and Biester, 2021; Haitzer et al., 2002; Muhaya et al., 1997), which would approximate our modeled 2.6 ng Hg mg C-1 if we assume a carbon to weight carbon-to-weight ratio of 1:2. These high iHg values in DOM lead to high values in suspension feeders in both setups. The bioaccumulation of MeHg is very different from that of iHg and has the highest bioaccumulation in the top predators and

**Figure 2.** Megabenthos biomass in the modeled Southern North Sea, dominated by filter feeders, followed by deposit feeders, generalist feeders, suspension feeders, predators, and top predators. Biomass fluctuates between 10 and 15 gC m-2 and all functional groups have stable populations

predators, followed by deposit feeders and suspension feeders. In Fig. 4a, c, and e, we show the relationship between the trophic level and the bioaccumulation of iHg, MeHg, and tHg in megabenthosin the model is shown. There is an increase in the MeHg content with trophic levels that are level, which is not present for iHg. For iHg, there is There is a weak anti-correlation  $(R^2 = 0.20)$  between the bioaccumulation of iHg and the trophic level, which is mainly caused by the extremely high iHg content of the low-trophic-level suspension feeders. There is no positive relationship between the bioaccumulation of tHg and the trophic level  $(R^2 = 0.02)$ , while this is there is a positive relationship present in the AS model  $(R^2 = 0.50)$ ; this indicates that our base model underestimates the bioaccumulation at higher trophic levels.

**3.3 Bioaccumulation in the gobal global dataset**

330

335

340

345

In Table 2, we show the results of a linear regression using the global dataset taking into account while accounting for both the trophic level and the feeding strategy; the relative fit of each modeleach model's relative fit explains Hg bioaccumulation based on both factors. The trophic level and feeding strategy are adapted to regressions are based on the natural logarithms of iHg, tHg, and MeHg. This shows that we can explain as dependent variables. These linear regressions show that the bioaccumulation of In(MeHg) MeHg can be predicted very well ( $R^2$ =0.72) with a linear model that takes both drivers into account, while the bioaccumulation of iHg is poorly explained ( $R^2$ =0.11) and the bioaccumulation of tHg has an average fit tHg shows intermediate explanatory power ( $R^2$ =0.46). Furthermore, we show the unique contributions of the fit of each driver, the partial  $R^2$ . Note that feeding strategy and trophic level can sometimes co-correlate, especially in the case of high MeHg bioaccumulation in predators, benthic fish, and seabirds, as predators are naturally higher in trophic level than the preythey consumenaturally occupy higher trophic levels than their prey. The feeding strategy has an explanatory power larger greater explanatory power than that of the trophic level for tHg and iHg, while it is similar for MeHg. Despite the limitations mentioned

Figure 3. Modeled bioconcentration and biomagnification of iHg and MeHg. Partitioning to detritus and DOM is colored as bioconcentration. The y-axis is cut to show the high and low values. Notably Notable is the high iHg to mgC ratio of associated with detritus and DOM, leading to elevated iHg in suspension feeders. Additionally, higher trophic level animals have higher biomagnified MeHg.

above, this still shows that the partial R2 for the feeding strategy is double twice that of the trophic level for tHg, demonstrating the importance of the feeding strategy for the bioaccumulation of tHg feeding strategy in Hg bioaccumulation at the base of the food web.

**Table 2.** R-squared and Partial R-squared Results for ln(THg), ln(iHg), and ln(MeHg)

| Model                                | ln(tHg) | ln(iHg) | ln(MeHg) |
|--------------------------------------|---------|---------|----------|
| Full Model R-squared                 | 0.46    | 0.11    | 0.72     |
| Partial R-squared (Feeding Strategy) | 0.22    | 0.089   | 0.32     |
| Partial R-squared (Trophic Level)    | 0.10    | 0.012   | 0.31     |

**3.4 The allometric scaling law in high trophic level high-trophic-level animals**

In Table 3, we show the model performance against the global dataset performance of the base and the AS model. This AS models against the global dataset. Table 3 shows that if we take the allometric scaling law into account, the model results for

high-trophic level animals increase high-trophic-level animals improve considerably. In Fig. 4b, d, and e, we show the relation relationship between the natural logarithm of bioaccumulation and the trophic level of the AS model in the Southern North Sea setup. The normalized bias in the predator and top predators the top predator decreased from -0.37 and -0.82 to -0.26 and -0.24-0.80 to -0.32 and -0.12, respectively. Our base model does agree well with both observed iHg ( $R^2$ =0.840.61) and MeHg ( $R^2$ =0.86) in the Southern North Sea setup, but this is mostly driven by accurate predictions in the lower trophic levels while there is a normalized bias of -0.84 in the Top Predators. This is improved dramatically -0.80 in MeHg bioaccumulation in the top predator. The model performance in MeHg bioaccumulation is improved substantially in the AS model with the reduction of the normalized bias of top predators to -0.32 the top predator to -0.22, which improves the overall  $R^2$  of the model to >0.99.

355

**Table 3.** Statistical analysis of model performance for iHg and MeHg levels by feeding strategy for the Southern North Sea (SNS) and Northern North Sea (NNS). The predator and top predator of both the default setup and the Allometric Scaling (AS) model is are shown. For all individual feeding strategies, we show the normalized biasis shown, and for the full model, the RMSE, NRMSE,  $R^2_{Pearson Pearson}$ , and  $R^2_{Residual}$  is shown.

|                                         | S      | NS    | NNS   |       |  |
|-----------------------------------------|--------|-------|-------|-------|--|
|                                         | iHg    | МеНд  | iHg   | MeHg  |  |
| Suspension                              | 0.18   | 1.09  | -0.18 | 0.24  |  |
| Filter                                  | 1.48   | -0.28 | 1.45  | -0.69 |  |
| Deposit                                 | 1.01   | -0.36 | 0.34  | -0.75 |  |
| Generalist                              | 1.31   | -0.35 | 1.23  | -0.73 |  |
| Predator                                | 0.41   | -0.37 | 0.07  | -0.77 |  |
| Top predator                            | -0.22  | -0.80 | -0.46 | -0.92 |  |
| Predator (AS)                           | 0.41   | -0.31 | 0.07  | -0.75 |  |
| Top predator (AS)                       | -0.22  | -0.12 | -0.46 | -0.67 |  |
| Overall Model Perfo                     | rmance |       |       |       |  |
| RMSE                                    | 40     | 132   | 40    | 146   |  |
| NRMSE                                   | 0.36   | 0.35  | 0.35  | 0.39  |  |
| $R^2_{Pearson}$                         | 0.61   | 0.86  | 0.24  | 0.94  |  |
| R 2 Residual      | <0     | <0    | <0    | <0    |  |
| RMSE (AS)                               | 40     | 22.8  | 40    | 108   |  |
| NRMSE (AS)                              | 0.36   | 0.061 | 0.35  | 0.29  |  |
| R 2 Pearson (AS)  | 0.61   | >0.99 | 0.24  | 0.99  |  |
| R 2 Residual (AS) |

Figure 4. The influence of trophic level on the bioaccumulation of MeHg, iHg, and tHg in both the AS model (panels a, c, e) and the base model (panels b, d, f). In the AS model, the relationship with trophic level is stronger, where  $\frac{\ln(MeHg)=1.24TL-0.03ln(MeHg)=1.24TL-0.03}{\ln(MeHg)=0.64TL+1.42ln(MeHg)=0.64TL+1.42}$ . TL represents trophic level, and MeHg is expressed in ng Hg g-1 d.w. For iHg, the bioaccumulation patterns are nearly identical, with  $\frac{\ln(MeHg)=0.19TL+5.11}{\ln(MeHg)=0.19TL+5.11}$  for the AS model and  $\frac{\ln(MeHg)=0.18TL+5.11}{\ln(MeHg)=0.18TL+5.11}$  for the base model, both showing a weak inverse correlation with trophic level, largely due to higher iHg levels in  $\frac{\ln(MeHg)=0.43TL+3.76}{\ln(MeHg)=0.43TL+3.76}$  compared to the base model ( $\frac{\ln(MeHg)n(MeHg)=0.04TL+4.75}{\ln(MeHg)=0.04TL+4.75}$ ), driven by the stronger association between MeHg and trophic level in the AS model.

**3.5 Comparing model and observations**

365

**3.5.1 The effect of feeding strategy on bioaccumulation**

The mean annual average annual mean and range of the bioaccumulation of modeled bioaccumulated iHg and MeHgin our model and the range and mean of measured iHg and MeHg, along with the observed ranges and means, are shown in Table 4. We additionally visualised in Fig. 5 visualized the modeled values of the AS model in the Southern North Sea compared to the observations —in Fig. 5. In Fig. 5a, the bioaccumulation of MeHg, and in Fig. 5b the bioaccumulation of iHg is visualised are visualized. All values fall within the range of observations, except for the modeled top predator in the base model. In the AS model, the top predator has values for both iHg and MeHg in both the Southern North Sea and the Northern North Sea that are within the range of observations. The most notable observation for iHg bioaccumulation is that, although the variation in measured iHg is considerable, suspension feeders consistently have high iHg values. In both the Southern North Sea setup and the observation observations, the mean MeHg is lowest in suspension feeders (17 and 8 ng Hg g-1 d.w. respectively), while it is very similar for deposit feeders (22 and 35 ng Hg g-1 d.w. respectively), filter feeders (28 and 39 ng Hg g-1 d.w. respectively), and generalist feeders (26 and 40 ng Hg g-1 d.w. respectively). MeHg is notably higher for predators and highest for top predators in the observations with Observed MeHg is higher in predators (77 and ng Hg g-1 d.w.) and highest in top predators (381 ng Hg g-1 d.w. respectively which is close to the—). These values closely match the modeled MeHg values of 54 and 337 in the AS model for predator and top predator than the 49 and 73 ng Hg g-1 in the base model respectively win the AS model in the Southern North Sea.

[revised manuscript text omitted]

In addition to using the global bioaccumulation dataset to evaluate our hypothesis that the feeding strategy is a key driver of bioaccumulation, we also evaluate if our hypothesis holds true assess whether this hypothesis holds when analyzing a comprehensive published dataset from a single study. The fit of the linear model against the natural logarithm of the a linear model to the natural log of bioaccumulated MeHg based on the data published by McClelland et al. (2024) is shown in Fig. 7. The R2 is similar with at 0.43 and 0.45 in the CB and MT, respectively, while the bioaccumulation is a bit bioaccumulation is slightly lower in the CB ( $\frac{\log(\text{MeHg}_{BA})}{\log(\text{MeHg}_{BA})} = 0.137 + 1.14 \text{TL}$ ) compared to that in  $ln(MeHg_{BA}) = 0.137 + 1.14 \text{TL}$ ) compared to the MT ( $\frac{\log(\text{MeHg}_{BA})=0.256+1.39*\text{TL}}{\log(\text{MeHg}_{BA})}$ , where  $\ln(\text{MeHg}_{BA})=0.256+1.39\text{TL}$ ). Here, MeHgBA is the bioaccumulated MeHg in ng Hg mg-1 d.w., and TL is the trophic level. The influence of the feeding strategy on MeHg bioaccumulation based on the results of McClelland et al. (2024) is shown in Table 5. While the only significant effect is that deposit feeders in the MT having have less MeHg than would be expected on given their trophic positions, some other effects are consistent, albeit not significant although not significant, in both locations. The strongest effect is that filter feeders have consistently consistently have higher MeHg (residuals are 0.7 in the CB and 0.8 in the MT), while deposit feeders have lower MeHg (residuals are -0.2 in the CB and -0.5 in the MT). The results of the same analyses for phylar effects of phylum on MeHg bioaccumulation are shown in Table 6. Here we see two consistent significant effects. Molluscs have elevated MeHg levels (residuals are 0.61 in the CB and 0.51 in the MT), while arthropods have reduced MeHg values (residuals are -0.35 in the CB and -0.30 in the MT). The percentage difference in MeHg bioaccumulation per feeding strategy is visualized in Fig. 8 and per phyla phylum in Fig. 9. The average percentage difference between observed values and the expectation based on trophic level is 102% and 128% in the CB and MT<del>respectively, respectively, for filter feeders, while deposit.</del> Deposit feeders have 19% and 37% less MeHg than would be predicted based on trophic level alone in the CB and MT, respectively. In the analysis per phylumphylum-level analysis, we see that molluscs have highly elevated MeHg levels with an increase of 66% (CB) and

**Figure 7.** The linear fitted model between the natural logarithm of the bioaccumulated MeHg in ng Hg mg-1 d.w. and the Trophic Level in the data presented by McClelland et al. (2024). For extra clarity the different Phyla shown with different colors while the different feeding strategies are marked with different symbols. In both the CB and MT setups there positive relationship between trophic level and the bioaccumulation of MeHg, but R2 is only 0.43 and 0.45 in the CB and MT respectively, so it does not explain the full variation in bioaccumulation.

85% respectively in the CB and MT(MT). The largest reduction in observed MeHg compared to the predicted values based on trophic level is in arthropods; here there is a decrease compared to the predicted values of 29% and 26% in the CB and MT, respectively.

The results of the final analyses are shown in Table 7. Despite the lower sample size, which reduces statistical power, the results indicate that filter feeders consistently have higher MeHg levels than predicted based on their trophic position and phyla, while deposit feeders tend to have lower MeHg concentrations. These results are stronger in the MT, with a change of 118% and -40% in filter and deposit feeders respectively than in the CB with a change of respectively. In the CB, the changes are smaller, with 7.2% and -14.8% in filter and deposit feeders, respectively. It must be stated that this final analysis is included to address potential concern between concerns regarding the co-correlation of between phyla and feeding strategy, but the problem of the although the reduced sample size has to be addressed remains a limitation. In the CB, where the increase in MeHg in filter feeders is rather low after correcting for both trophic level and feeding strategy, there are only three filter feeders, which are molluscs, and they. These three make up 3/5 of the mollusc samples in this location, meaning that so these results should be seen with skepticism interpreted with caution, as filter feeders and molluscs have too much overlap overlap strongly. On

460

Figure 8. Percentage difference from trophic level predicted MeHg concentrations by feeding strategy, with error bars showing  $\pm 1$  SE. In both CB and MT regions, filter feeders have elevated MeHg levels relative to trophic level based expectations, while deposit feeders are reduced. Predators display higher MeHg than predicted, though the effect is smaller than in filter feeders; in CB, this increase does not exceed one SE. Generalist feeders have a slight reduction compared to expectations, but this is well within one SE, and were not present in CB for cross-region comparison.

Figure 9. Percentage difference from the predicted MeHg bioaccumulated based on trophic level per phylaphylum, the error bars represent ±1 SE. The notable phyla are Mollusca and Arthropoda, while Mollusca have a notable increase in MeHg bioaccumulation compared to the prediction of 85% and 66% respectively in the CB and MT, there is a reduction of 26% and 29% in Arthropoda in the CB and MT respectively. Annelida are inconsistent with an increase in the CB and decrease in the MT compared to the predictions. Echinodermata have a mean reduction compared to the prediction in both the CB and the MT, but the SE is much larger than the mean effect.

**Table 5.** Mean residuals ( $\pm$ SE) of  $\frac{\log \ln (MeHg)}{\log \log (MeHg)}$  by feeding strategy and region, after trophic level correction. Significant deviations (p < 0.05) are marked with \*.

| Region | Feeding Strategy | n  | Mean Residual $\pm$ SE | p-value |
|--------|------------------|----|------------------------|---------|
| СВ     | Deposit feeder   | 16 | $-0.208 \pm 0.181$     | 0.268   |
| CB     | Filter feeder    | 3  | $0.704\pm0.286$        | 0.133   |
| CB     | Predator         | 6  | $0.203 \pm 0.331$      | 0.568   |
| MT     | Deposit feeder   | 15 | $-0.467 \pm 0.159$     | 0.011*  |
| MT     | Filter feeder    | 5  | $0.824\pm0.395$        | 0.105   |
| MT     | Generalist       | 3  | $-0.143 \pm 0.319$     | 0.698   |
| MT     | Predator         | 12 | $0.277\pm0.226$        | 0.247   |

**Table 6.** Mean residuals ( $\pm$ SE) of  $\frac{\log \ln (MeHg)}{\log (MeHg)}$  by phylum and region, after trophic level correction. Significant deviations (p < 0.05) are marked with \*.

| Region | Phylum        | n  | Mean Residual $\pm$ SE | p-value      |
|--------|---------------|----|------------------------|--------------|
| СВ     | Annelida      | 6  | $0.229 \pm 0.424$      | 0.612        |
| CB     | Arthropoda    | 11 | $-0.349 \pm 0.137$     | 0.0294*      |
| CB     | Echinodermata | 3  | $-0.198 \pm 0.584$     | 0.767        |
| CB     | Mollusca      | 5  | $0.611 \pm 0.211$      | 0.0446*      |
| MT     | Annelida      | 5  | $-0.405 \pm 0.377$     | 0.343        |
| MT     | Arthropoda    | 12 | $-0.304 \pm 0.111$     | 0.0196*      |
| MT     | Echinodermata | 5  | $-0.188 \pm 0.509$     | 0.730        |
| MT     | Mollusca      | 13 | $0.509 \pm 0.231$      | $0.0482^{*}$ |

the other hand, in the MT, there are five filter feeders from multiple phyla (Mollusca and Echinodermata), and the effect is considerably stronger, with filter feeders having 118% more MeHg than would be expected based on their trophic level and phyla.

**4 Discussion**

**4.1 The role of feeding strategy on the bioaccumulation of MeHg**

Overall we find that the feeding strategy plays an important role in the bioaccumulation of MeHg in our model, the global dataset, and the single dataset published by McClelland et al. (2024). Because of this, we find it convincing that the role of the feeding strategy in MeHg bioaccumulation deserves further attention in both modeling and empirical studies.

**Table 7.** The effect of feeding strategy on MeHg bioaccumulation per Region compared to the prediction accounting for both trophic level and feeding strategy. Significant (p < 0.05) is marked with \*. There is still a consistent increase in filter feeders and a consistent decrease in deposit feeders. This is effect is larger in the MT with a relative percentage increase of 118% in filter feeders and a decrease of 40% in deposit feeders.

| Feeding Strategy | % Diff (MT) | p-value (MT) | % Diff (CB) | p-value (CB) |
|------------------|-------------|--------------|-------------|--------------|
| Deposit feeder   | -40.0       | 0.034*       | -14.8       | 0.888        |
| Filter feeder    | 118.0       | 0.034*       | 7.2         | 0.888        |
| Generalist       | -25.9       | 0.563        | _           | _            |
| Predator         | 3.0         | 0.895        | 9.4         | 0.888        |

**4.2 The AS model**

480

485

490

495

Our base model fails to reproduce the high values in the While the base model underestimates MeHg in top predators, but this is improved this improvement is observed in the AS model. The normalized bias is reduced for MeHg bioaccumulation in top predators in the Southern North Sea decreased from -0.80 to -0.22-0.12. In the AS model, we get obtain a linear relationship of 1.24x-0.03 ( $R^2=0.93$ ), which has a similar slope to  $\ln(\text{MeHg}) = 1.24x-0.03$  ( $R^2=0.93$ ). This slope is similar to the linear relationships found in the 1.14x+0.387 and 1.39x+0.256 found in CB and MT station stations of the McClelland et al. (2024) dataset, which are 1.14x+0.387 and 1.39x+0.256, respectively. The improvement in the AS model compared to the base model indicates that the lower MeHg release rates in high-trophic-level animals should be taken into account. We tried to run accounted for when modeling MeHg bioaccumulation in higher trophic levels. We also tested the model with the lower MeHg release rate in applied to all megabenthos, but this resulted in unrealistically high values in both both at the base and top of the food web, so we cannot just use the lower. Therefore, implementing the allometric scaling law is preferable to lowering the MeHg release rate at every trophic level. Because of this, we conclude that We conclude that, in addition to the feeding strategy, the difference in the release rate of MeHg related to differences in MeHg release rates associated with body size, metabolic ratemetabolism, or activity also likely has a significant contribution likely contribute significantly to the high MeHg values in high-trophic-level animals.

**4.3 Bioconcentration of iHg**

The largest bias in our model, which remains uncorrected in the AS model, is the overestimation of iHg in the filter and generalist feeders. Although the modeled iHg values are not out of outside the observed range, the consistently high normalized bias indicates that the model overestimates the bioaccumulation of iHg iHg bioaccumulation. In Fig. 3, we can see that the vast majority of iHg in filter and generalist feeders originates from bioconcentration. The most important driver of bioconcentration is the ratio between uptake and release raterates, or the uptake-release ratio. Our model has uses an uptake-release ratio of 210 l g-1 d.w.This is, derived from Tsui and Wang (2004), as it represents the lowest ratio found reported in the literature. The exact

rate was obtained by withdrawing subtracting the modeled carbon excretion rate and deducting this from the measured iHg release rate to have obtain an iHg-specific release rate; this rate, which was found to be 0.04 d-1, as presented in Amptmeijer et al. (2025). Other studies, such as Pan and Wang (2011), found higher uptake-release ratios between 424 and 781 l g-1 d.w.

The discrepancy between the modeled and observed iHg can be caused by may stem from several factors. First, iHg concentrations in North Sea megabenthos could be higher than those reported in other coastal zones. However, there are currently no empirical data to support or invalidate this conclusion at the moment. Secondly confirm or refute this. Second, translating experimentally obtained uptake and release rates to observations of iHg might depend on the may depend on drivers that are not captured in the model. In either case, it is hard-difficult to verify the root cause of this high normalized bias, as the bioaccumulation of iHg iHg bioaccumulation is comparatively understudied compared to the bioaccumulation of relative to MeHg, both in models and empirical studies.

**4.4 Model structural limitations**

500

505

510

515

520

525

530

The GOTM-MERCY-ECOSMO coupled system captures the influence of feeding strategy on MeHg bioaccumulation, but performance differs between regions. The Southern North Sea setup performs well in pelagic Hg cycling and benthic bioaccumulation, whereas the Northern North Sea setup underestimates MeHg in all benthic groups and shows unexpectedly high mesozooplankton tHg, which cannot be directly validated due to a lack of data. The the model predicts lower MeHg bioaccumulation in deeper water, which is not true for the does not match observations by McClelland et al. (2024). This suggests that MeHg fluxes from the pelagic to the benthic system are underestimated.

In shallow waters, megabenthos can feed directly on the phytoplankton and zooplankton blooms, which leads to a strong bentho-pelagic exchange of organic carbon and Hg. In deeper waters, megabenthos mainly rely on detritus that sinks from the euphotic zone, which, in our model, carries less MeHg. But the The higher performance in shallow conditions combined with the reduced performance in deeper conditions indicates that the model could be improved in areas driving deep water processes controlling deep-water MeHg bioaccumulation, such as sediment Hg chemistry, deep-water Hg speciation, the bentho-pelagic coupling, or the transport of Hg to deeper water due to the sinking of due to sinking organic material.

**4.5 Data-related limitations**

Combining the results of the model and the literature studies is difficult due to the high uncertainty in most drivers, including the organic material content of dry weight, and the result. Therefore, the results should be viewed with skepticism caution. For example, the data analyses by McClelland et al. (2024) were prepared to mimic consumption by predators: for small arthropods, their skin was skins were not removed, but for gastropods and bivalves, the shell was not taken into account for the weightas predators would typically not eat this. shells were not included in the weight, as predators typically would not eat them.

The concentration of MeHg per unit energy is arguably the key measure in bioaccumulation. Predators need to ingest a specific energy amount amount of energy, so if a prey is composed of half organic material and half non-organic components, such as shell, its MeHg eontent concentration per dry weight is halved. However, predators would consume double the dry

weight to obtain the same energy, and thus the same MeHg MeHg intake remains unchanged. In general, the energy appears to be consistent with energy content is best approximated by Ash Free Dry Weight (AFDW), as such ideally we would normalize all measurements of MeHg bioaccumulation per. Ideally, all MeHg bioaccumulation measurements should be normalized to AFDW (Weil et al., 2019). Unfortunately, doing Unfortunately, performing this conversion reliably on published data is not possible, as AFDW varies with the age and body size of animals, which information information that is not always registered and reported or made available (Eklöf et al., 2017).

**4.6 Potential improvements**

535

540

545

550

555

560

The model has uses the same rates for all megabenthos groups. This allows us to isolate the effect of the feeding strategy, but it should be taken into account that this also means that the model limited in its also limits the model's ability to predict bioaccumulation of iHg or MeHg in specific animals. Our model is run in simulations are run for the North Sea, while most of the field observations are whereas most field observations come from different regions. This means that Therefore, this study should be seen as a hypothesis-generating work that identifies, identifying the role of feeding strategies on the bioaccumulation of in iHg and MeHg bioaccumulation as a potential direction for further empirical studies, rather than a complete classification. Based on this work, however, it appears that providing a robust quantification.

Based on our results, the inclusion of megabenthos with different feeding strategies could improve the performance of MeHg bioaccumulation models. At the same time, our analyses demonstrate the underperformance of the model in simulating the deep water deep-water bentho-pelagic coupling, which. This indicates that the performance of the ECOSMO E2E-MERCY-GOTM ECOSMO-MERCY-GOTM coupled system should be critically evaluated before it can be being used for predictive bioaccumulation modelling in deeper watermodeling in deeper waters.

**5 Summary and conclusion**

In this study, we analyze the role of the trophic level and the feeding strategy on the bioaccumulation of iHg and MeHg. We did this by performing a literature study and running a fully coupled 1D model in two idealized setups representing two different hydrodynamic regimes in which megabenthic communities can live. Our study estimates that the trophic level predicts trophic level alone explains up to 32% of the variability of MeHg in MeHg concentrations in the benthic food web. If we include both the feeding strategy and the trophic level, this increases Including both trophic level and feeding strategy increases this explained variability to 72%. We show that several feeding strategies have significant differences, highlighting significant differences between feeding strategies.

We show Additionally, we demonstrate that there are notable differences between feeding strategies. iHg is higher in suspension feeders and MeHg is low in suspension feeders and grazers, while filter feeders have the highest MeHg followed by deposit feeders. Our model expands on this by demonstrating that we can accurately model the bioaccumulation of iHg and MeHg at the base of the food web by only taking the feeding strategy into account.

We find it convincing that both our Our model results, the literature study in which we aggregate all measurements, and the literature study where we take samples from a single study all suggest similar patterns where together with both aggregated and single-study literature analyses, consistently suggest that feeding strategy is an important a key driver of bioaccumulation at the base of the food web, even if these results should be seen with skepticism. While these findings should be interpreted cautiously, due to the large inherent uncertainty in the model. Because, the consistent indication that feeding strategy is a key driver in MeHg bioaccumulation does suggest it is a potential target for further empirical studies. In the Southern North Sea setup, feeding strategy in our base model correlates well with observed iHg ( $R^2$ =0.61) and MeHg ( $R^2$ =0.86) in the Southern North Sea setup, it appears that the feeding strategy, suggesting it is a key driver controlling the bioaccumulation of both iHg and MeHg of bioaccumulation at the base of the food web. However, this strong performance is mostly because 4 out of our 6 This strong performance largely reflects the fact that four of our six megabenthos groups are low trophic level-low-trophic-level non-predators, and our base model starts to underperform considerably in its ability to model. The model underperforms in predicting MeHg bioaccumulation in higher trophic levels. This problem is solved by taking into account higher trophic-level organisms. This is improved upon by accounting for the allometric scaling law and assuming that MeHg removal from the organism is not linked to the total but rather to the base metabolic rate. Because of this, we accept conclude that our hypothesis that the feeding strategy is an essential driver of the bioaccumulation of iHg and MeHg in low-trophic-level animals , but is supported by both our model and observations, but our model shows that other differences in the organisms between highand low-trophic-level animals should also be taken into account when predicting MeHg values in high-trophic-level fish. Our model and observation focus on lower-trophic-level benthic invertebrates, with some high-trophic-level animals added to create context. The importance of this for the bioaccumulation of MeHg in animals of high trophic levels high-trophic-level animals is that all biomagnification is an exponential function starting at the base of the food web. Therefore, a change in MeHg at the base of the food web will correspond to a similar relative increase at the top of the food chain. Because the feeding strategy has such a large impact on the base of the food web, high trophic-level high-trophic-level animals could have considerably different MeHg values depending on the species composition of the base of the food web.

Interestingly, despite the although iHg has a lower biomagnification potential of iHg, its high abundance in certain concentrations in some low-trophic-level animals can lead to higher tHg in low-trophic-level animals result in higher tHg levels in these organisms than in higher-trophic-level animals. This discrepancy can distort risk perception, as safety assessments often rely on tHg measurements that do not distinguish between iHg and MeHg. Animals, such as suspension-feeding bivalves, may have high Hg values while remaining safe for human consumption. Our findings demonstrate the importance of Hg speciation data in marine organisms to help improve food safety guidelines and inform regulatory policies, demonstrating the need to measure different Hg species to not misidentify the toxicity of biota.

**5.1 Societal relevance & future work**

565

570

575

580

585

590

Our study highlights the critical role of benthic diversity in driving MeHg bioaccumulation. Both trophic interactions and the feeding strategy significantly influence MeHg bioaccumulation, which has important implications for seafood safety and fisheries management. Understanding these processes can help explain the spatial and temporal variability in the MeHg content of fish, which is crucial for policymakers to develop effective regulations that safeguard human health and marine ecosystems.

Filter feeders and molluscs typically accumulate more MeHg than other organisms at similar trophic levels. This pattern is consistent not only in our models but also in available data. This raises a hypothesis that expanding bivalve populations, as seen in mussel or oyster farming, might affect MeHg bioaccumulation in higher trophic levels. This is supported by the observations that fish in lakes invaded by zebra mussels have higher Hg levels than fish in lakes without zebra mussels Blinick et al. (2024) (Blinick et al., 2024). However, such ecological alterations also impact other bioaccumulation factors like biomass distribution and trophic interactions. While our findings support the role of filter feeders and molluscs in MeHg dynamics and higher bioaccumulation in top predators, the complexity of ecological situations requires further case-specific studies to understand if and when bivalve communities lead to increased MeHg transfer.

Modeling studies can help our understanding of the factors influencing MeHg bioaccumulation, but the ability to accurately predict MeHg bioaccumulation needs to be carefully validated. Our findings reveal that filter-feeding molluscs and DOM-utilizing suspension feeders have different Hg bioaccumulation patterns compared to other megabenthos. Modeling bivalve aquaculture or DOM-consuming suspension feeders can help explore their potential role in altering MeHg bioaccumulation. Understanding how functional traitslike feeding strategy, such as feeding strategy, influence MeHg transfer remains key to improving both is essential for improving predictive models and environmental risk assessments.

Our findings suggest that fish from food webs dominated by filter feeders would have the highest MeHg content, since filter feeders have the highest MeHg content in both our model and observations. It also creates an indication that the introduction of bivalve communities in the form of mussel or oyster farming could increase MeHg levels in higher food chains. However, such changes in the ecosystem would inevitably change other factors in the ecosystem, including biomass and trophic interactions that are also essential drivers for MeHg bioaccumulation. While our model should be seen as a hypothesis-generating work that requires empirical validation, it does suggest that case-by-case studies are needed to fully understand how changes in the base of the food web will affect the concentration of MeHg in high trophic level fish.

Based on our results, we strongly recommend targeted field studies that systematically measure iHg, MeHg, and trophic levels level in diverse marine communities to assess how the structure of the food web influences the bioaccumulation of MeHg. 
[revised manuscript text omitted]